# Interactions between TTYH2 and APOE facilitate endosomal lipid transfer

Anastasiia Sukalskaia[1], Andreas Karner[2], Anna Pugnetti[1], Florian Weber[2], Birgit Plochberger[2] & Raimund Dutzler[1✉]

The Tweety homologues (TTYHs) constitute a family of eukaryotic membrane proteins that, on the basis of structural features, were recently proposed to contribute to lipid transfer between soluble carriers and cellular membranes[1]. However, in the absence of supporting data, this function was hypothetical. Here through pull-down of endogenous proteins, we identify APOE as the interaction partner of human TTYH2. Subcellular fractionation and immunocytochemistry assays showed that both proteins colocalize in endosomal compartments. Characterization of the specific interaction between APOE and TTYH2 through binding assays and structural studies enabled us to identify an epitope in an extended domain of TTYH2 that faces the endosomal lumen. Structures of complexes with APOE-containing lipoprotein particles revealed a binding mode that places lipids in a suitable position to facilitate their diffusion into the membrane. Moreover, in vitro studies revealed that lipid transfer is accelerated by TTYH2. Collectively, our findings indicate that TTYH2 has a role in the unloading of APOE-containing lipoproteins after they are endocytosed. These results define a new protein class that facilitates the extraction of lipids from and their insertion into cellular membranes. Although ubiquitous, this process could be of particular relevance in the brain, where APOE is involved in the transfer of lipids between astrocytes and neurons.

The hydrophobic nature of membrane lipids and triglycerides requires their transport in the extracellular fluid as soluble lipoprotein complexes. One of the proteins that has evolved for lipid transport is apolipoprotein E (APOE)[2,3]. Although ubiquitously expressed, with the highest levels found in the liver[4], APOE has a particularly important role in the brain where, as the main apolipoprotein, it is essential for the shuttling of phospholipids and cholesterol between astrocytes and neurons[5]. Among the three APOE subtypes (APOE2, APOE3 and APOE4), APOE4 is a major determinant of the genetic predisposition to Alzheimer's disease[6–8]. APOE is a modular, 299-residue-long protein[9]. Its amino-terminal domain folds into an antiparallel four-helix bundle and has two positions that contain either a cysteine or an arginine that define the three subtypes[10–12]. However, the relationship between these amino acids and the susceptibility to developing neurodegenerative diseases is poorly understood. The flexible carboxy-terminal domain is responsible for the engagement of APOE with lipids and for the oligomerization of the protein in their absence[13,14]. After binding to lipids, tetrameric APOE dissociates and rearranges, which enables its C-terminal helices to wrap around the edge of a bilayer to form a soluble, disc-shaped lipoprotein[15,16]. When triglyceride and cholesterol ester levels are high, the complex expands into a spherical particle that is thought to be coated with lipoproteins[6]. For uptake into cells, APOE-containing lipoproteins interact with receptors located at the plasma membrane that bind the lipoprotein to facilitate endocytosis[17]. Such receptors include the low-density lipoprotein receptor, which interacts with an epitope located on the N-terminal domain of APOE[18].

In contrast to the well-known mechanism of how these complexes are recognized on the cell surface, the subsequent intracellular steps of how APOE-containing lipoproteins are processed after endocytosis are unclear.

A class of proteins that have been proposed to play a part in lipid-carrier interactions are the Tweety homologues (TTYHs)[1,19]. TTYHs constitute a family of eukaryotic membrane proteins with poorly defined function[20,21]. Although the three human family members (TTYH1, TTYH2 and TTYH3) were initially classified as anion channels[22,23], this role was later refuted in studies that investigated their structure and function. All three paralogues share similar structural properties as homodimeric proteins, with each subunit containing five transmembrane helices and an extended extracellular domain, which mediates the bulk of subunit interactions[1,24]. A distinct feature of TTYHs is the wide hydrophobic cavity that extends from the membrane towards the extracellular environment, which is filled with lipid-like densities in all structures[1]. This feature has led to the speculation that TTYHs function as facilitators of lipid transfer between the cell membrane and soluble lipid-carrier proteins[1]. However, in the absence of known interaction partners, this assumption was hypothetical. Here we identify APOE as an interaction partner of the paralogue TTYH2 in endosomal compartments. We also characterize the structural basis of the interaction and the ability of TTYH2 to facilitate lipid transfer between APOE and the cell membrane. Overall, our work establishes a previously unknown function of a membrane protein that presumably catalyses the unloading of lipids from lipoprotein complexes after they are endocytosed.

[1]Department of Biochemistry University of Zurich, Zurich, Switzerland. [2]Department of Medical Engineering, University of Applied Sciences Upper Austria, Linz, Austria. ✉e-mail: dutzler@bioc.uzh.ch

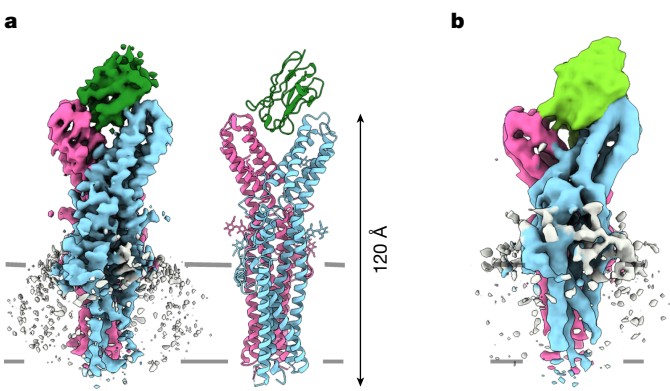

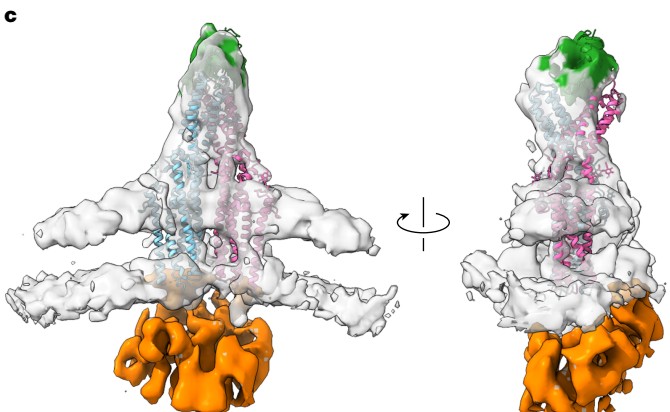

**Fig. 1 | Recognition of TTYH2 by synthetic nanobodies. a**, Cryo-EM density (3.7 Å, left) and ribbon representation (right) of the TTYH2–Sb1 complex. **b**, Cryo-EM density (4.3 Å) of the TTYH2–Sb2 complex. **c**, Cryo-EM density (8.2 Å) of the TTYH2–Sb1 complex in native vesicles isolated from HEK293 cells overexpressing TTYH2 by binding to resin containing immobilized Sb1. The relationship between views is indicated. The density of the mobile C terminus is in orange. In **a**–**c**, TTYH2 subunits are in cyan and magenta, sybodies are in green.

## Sybody selection against TTYH2

We sought to gain insight into the functional properties of TTYHs as potential mediators of lipid transfer by identifying their interaction partners. To facilitate this process for the ubiquitously expressed paralogue TTYH2, we screened nanobodies from synthetic libraries (termed sybodies)[25] against the purified protein. These efforts enabled the isolation of two sybody binders that selectively targeted human TTYH2: Sb1^TTYH2 (short Sb1) and Sb2^TTYH2 (short Sb2) (Extended Data Fig. 1a,b). Analyses of cryogenic electron microscopy (cryo-EM) structures of complexes of TTYH2 with Sb1 or Sb2 revealed clear densities that were attributed to the sybodies. The sybodies occupied equivalent regions on the extended extracellular domain of the dimeric protein in the gap located between the two subunits, which adopted a similar arrangement as initially identified in the apo-structure of TTYH2 (refs. 1,24) (Fig. 1a,b, Extended Data Table 1 and Extended Data Fig. 1c–e). Both structures contained one sybody per TTYH2 dimer, as binding of a second molecule would be prohibited because of steric reasons. For the TTYH2–Sb1 complex, the data facilitated detailed molecular interpretation. The sybody formed contacts with both subunits of the TTYH2 dimer at epitopes located in a region with poor sequence conservation (Extended Data Fig. 1f–h). Next, we investigated whether these binders recognize TTYH2 in a membrane environment, as previous studies have suggested different organizations of TTYHs in a cellular context. That is, the dimers either dissociate[24] or they assemble into larger oligomers[26]. To this end, immobilized Sb1 was used to pull down overexpressed TTYH2 in cell-derived membrane vesicles, and then structural characterization by cryo-EM was performed. Three-dimensional (3D) reconstruction of TTYH2 in the lipid bilayer showed the same characteristic features as observed in a detergent system (Fig. 1c, Extended Data Table 1 and Extended Data Fig. 1i–k). In this reconstruction, discernible density at the intracellular side seemed to correspond to the mobile C-terminal part of TTYH2, which folds up in the membrane environment or in the presence of an unknown interaction partner (Fig. 1c). Together, our results show that TTYH2 in native membranes can be recognized by a binder that was selected against the detergent-solubilized form of TTYH2. Therefore, the dimeric structure of TTYH2 formed in a detergent system is representative of its organization in cells.

## Discovery of APOE as an interaction partner

After selecting for sybodies that specifically recognize native TTYH2 dimers, we used Sb1 to isolate endogenous TTYH2 complexes from HEK293 cells. We then characterized the pulled down proteins by mass spectrometry (Extended Data Fig. 2a). TTYH2 and APOE were among the most abundant co-purified components, which indicated that they interact with each other (Extended Data Fig. 2b). Before characterizing the interaction of both proteins in detail, we were interested in the subcellular localization of TTYH2 and whether it overlaps with APOE to confirm that both proteins are located in suitable proximity in a cellular environment. To this end, we fractionated cellular compartments of untransfected HEK293 cells and a mouse neuroblastoma (N2A) cell line by density-gradient centrifugation and then performed western blot analyses (Fig. 2a and Extended Data Fig. 2c–j). In these experiments, strong bands of APOE and TTYH2 colocalized in endosomal fractions of HEK293 cells, as indicated by organelle-specific markers (Fig. 2a and Extended Data Fig. 2d). Similar localization of TTYH2 was also observed in N2A cells (Extended Data Fig. 2c,e). We also studied the cellular localization of TTYH2 by confocal fluorescence microscopy. A punctate pattern inside cells that overlapped with accumulated endocytosed APOE and the endosomal marker RAB9 was observed, which provided further evidence for the intracellular localization of TTYH2 (Fig. 2b,c and Extended Data Fig. 3a–c). Collectively, these experiments demonstrate that TTYH2 and APOE accumulate in overlapping endosomal compartments and that they interact in a cellular context.

## TTYH2–APOE interactions

To gain further insight into the interaction mechanism, we studied the binding of TTYH2 to APOE3 (referred to as APOE hereafter) in different lipidation states. To assay site specificity, we investigated the interaction of TTYH2 with APOE expressed in *Escherichia coli*. Delipidated APOE partially competed with Sb2 for its binding site in TTYH2 (Fig. 2d and Extended Data Fig. 3d–g). In contrast to full-length APOE, no competition was detected with its N-terminal domain, which suggested that the N terminus of APOE is insufficient to displace the sybody (Fig. 2d and Extended Data Fig. 3h). For experiments carried out with lipoprotein complexes, we reconstituted disc-shaped particles by assembling APOE with detergent-solubilized phospholipids and cholesterol. Similar results were obtained using this system, whereby these complexes were also able to compete with a fraction of the bound Sb2. These results demonstrate that TTYH2 interacts with APOE in different lipidation states at sites that partially overlap with the sybody epitope (Fig. 2d and Extended Data Fig. 3g).

Classical methods such as surface plasmon resonance spectroscopy and microscale thermophoresis were not suitable for characterizing APOE binding to TTYH2. Therefore, we used single-molecule force spectroscopy (SMFS) to directly assay the interaction between both proteins

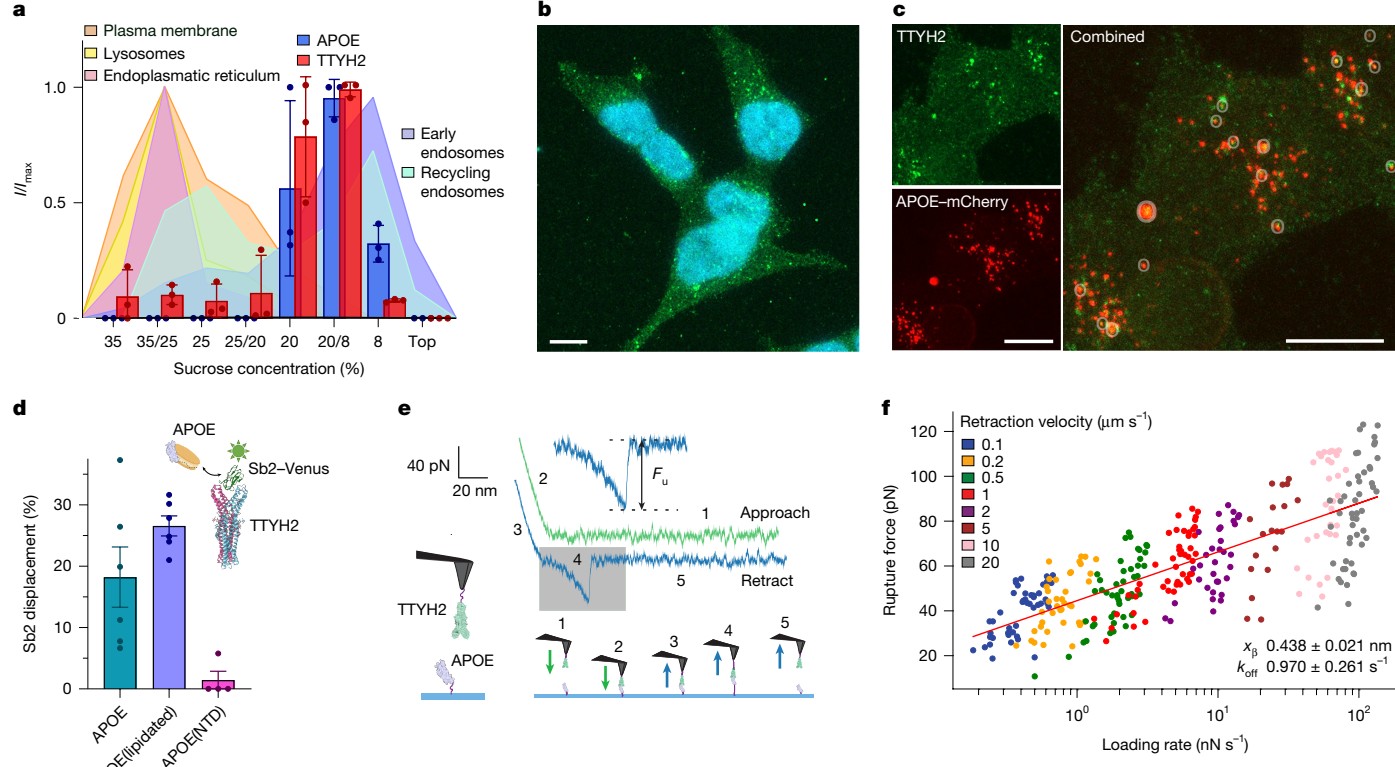

**Fig. 2 | TTYH2–APOE interactions and subcellular distribution. a**, Subcellular fractionation of HEK293 cells on a sucrose gradient was performed, and TTYH2, APOE and specific markers that define the cellular origin of the fractions (shaded background) were identified by western blotting. Plotted are averaged intensities ($I/I_{max}$) from $n = 3$ independent fractionation experiments. 35/25, 25/20 and 20/8 refer to samples from interfaces between the gradient steps. Errors are the s.e.m. **b,c**, Confocal fluorescent microscopy images of HEK293 cells showing the intracellular localization of TTYH2 (green). **b**, TTYH2 staining in relation to the nucleus (cyan). **c**, Colocalization of TTYH2 with endocytosed lipoproteins reconstituted from an APOE–mCherry fusion protein. Shown are individual channels (left) and their combination (right). Overlaps of both fluorescent markers are indicated by circles. **d**, Competition between Venus-labelled Sb2 bound to TTYH2 and APOE constructs indicates partial sybody displacement by lipidated APOE (APOE(lipidated), $n = 6$ replicates from 4 independent experiments) and delipidated APOE (APOE, $n = 6$ replicates from 4 independent experiments) but not its N-terminal domain (APOE(NTD), $n = 4$

replicates from 4 independent experiments). Shown are averages of the indicated number of experiments, errors are the s.e.m. Differences compared with the displacement by APOE(NTD) were analysed in a two-sample two-sided *t*-test and were significant for both lipidated and delipidated APOE (APOE(lipidated) $P = 0.000003$, APOE $P = 0.017$). **e**, Representative force–distance curve displaying the interaction between TTYH2 immobilized on the measurement tip and APOE immobilized on a solid support (see the scheme on the left), as measured by the $F_u$ (magnification of the highlighted area) after retraction of the tip. The steps of the measurement protocol are illustrated below and indicated in the traces. **f**, Analysis of the $F_u$ as a function of the loading rate measured at different retraction velocities of the tip. Shown are individual measurements, the calculated $k_{off}$ constant and the distance to the energy barrier from the equilibrium position ($X_\beta$). Error bars are the s.d. The line displays a fit to a Bell–Evans model. Scale bars, 10 μm (**b,c**). The diagrams in **e** and **d** were created using BioRender (https://www.biorender.com).

by measuring the mechanical dissociation of single APOE–TTYH2 complexes. We covalently attached TTYH2 to the measuring tip of an atomic force microscope and analysed its interactions with delipidated APOE immobilized on a glass surface. During the approach and subsequent retraction of the cantilever, stochastic binding events were observed, which enabled the measurement of an unbinding force ($F_u$), which is directly correlated to the dissociation kinetics of the complex[27–29] (Fig. 2e). Its linear dependence on the logarithm of the loading rate is a hallmark of a specific interaction and can be used to determine the kinetic off-rate ($k_{off}$) constant[30–32], which was measured to be about 1 s$^{-1}$ at neutral and acidic pH (Fig. 2f and Extended Data Fig. 3i). Similar binding behaviour was observed for immobilized Sb2, which was used as the positive control (Extended Data Fig. 3j). By contrast, addition of soluble Sb2 to the surrounding buffer led to binding competition with both immobilized Sb2 and APOE (Extended Data Fig. 3k). This result further highlights the specificity of binding of these two molecules and the overlap of their epitopes.

Collectively, the data from the two orthogonal methods used to characterize the interaction of APOE with TTYH2 suggest that the lipoprotein binds to an epitope that is presumably located at the extracellular domain of the membrane protein.

## TTYH2 in complex with delipidated APOE

To uncover the structural basis of its interaction with TTYH2, we studied complexes assembled with APOE in different lipidation states. We initially focused on complexes with an essentially lipid-free apolipoprotein. To this end, we added APOE that was expressed in *E. coli* and delipidated during the purification steps to TTYH2 purified in detergent and characterized its structure by cryo-EM. To confine its location in the complex, we labelled APOE with a nanogold cluster attached to its N terminus before its assembly with TTYH2 and collected a small dataset. In the resulting map, we observed strong density of the gold complex located in the gap between the extracellular domains of the TTYH2 dimer (facing the lumen of an intracellular compartment) (Fig. 3a). A large dataset of a complex with unlabelled APOE produced a map at 4 Å that was better resolved in the regions corresponding to TTYH2. This dataset also contained additional density at the extracellular domain in a fraction of particles separated by 3D variability analysis[33] (Fig. 3b, Extended Data Table 1 and Extended Data Fig. 4a–e). This additional density was wedged between both subunits, which was in accordance with the location observed in the nanogold-labelled

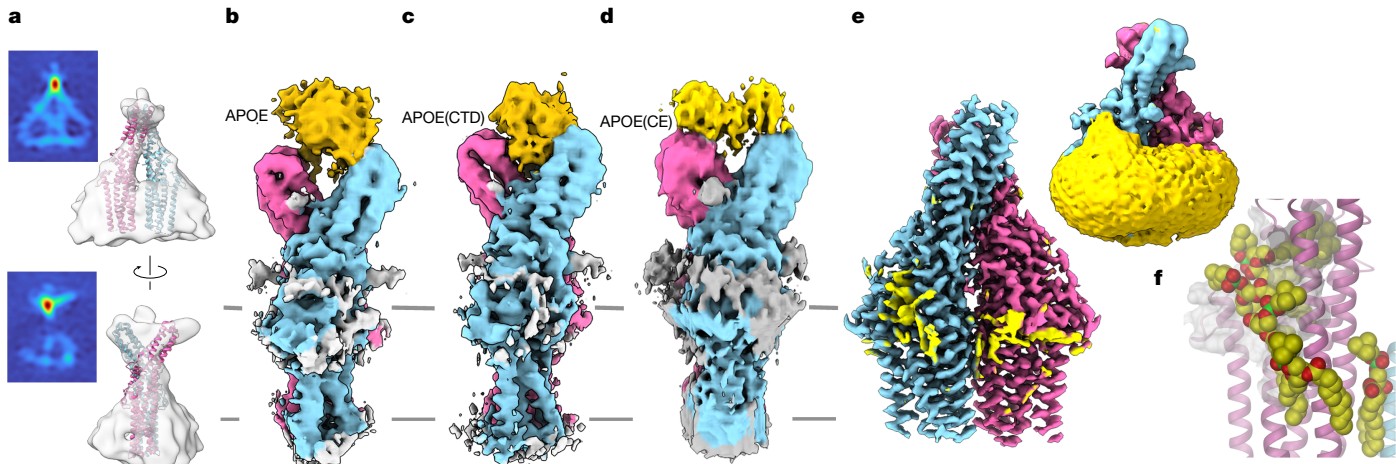

**Fig. 3 | Structures of TTYH2 in complex with delipidated APOE. a**, Low-resolution 3D reconstruction of TTYH2 in complex with APOE containing a nanogold label shown in two orientations. A slice through the density that defines the position of the gold label located between the two extracellular domains of the dimer is shown on the left. **b**, Cryo-EM density (4 Å) of detergent-solubilized TTYH2 in complex with delipidated APOE expressed in *E. coli*. **c**, Cryo-EM density (3.95 Å) of detergent-solubilized TTYH2 in complex with the delipidated C-terminal domain of APOE (APOE(CTD)) expressed in *E. coli*. **d**, Cryo-EM density (4.3 Å) of detergent-solubilized TTYH2 in complex with APOE in an uncontrolled lipidation state obtained after the coexpression of both proteins in HEK293 cells (APOE(CE)). For **b**–**d**, The lower resolution of the

density of APOE (gold) reflects the high local mobility of the bound protein. **e**, High-resolution (2.7 Å) structure of TTYH2 obtained by combining particles that did not contain APOE from all datasets reveals its interaction with lipids (yellow) that remained bound during purification. A low-pass-filtered density (right) illustrates the local distortion of the detergent or lipid environment (yellow) reaching into the lipid-filled cavity. **f**, Magnification of the region surrounding the hydrophobic cavity emerging from the membrane with interacting lipids. The protein is shown as a ribbon. Refined positions of selected lipid molecules are shown as space-filling models. The molecular surface is shown in white. For **b**–**f**, the two subunits of TTYH2 are shown in different colours.

complex (Fig. 3a,b). The heterogeneous binding of APOE to TTYH2 compromised its resolution and consequently prevented a detailed interpretation of its interaction. However, analyses of the maps enabled us to confine the TTYH2 region in contact with APOE to the upper part of the extracellular domain facing the gap between the two subunits. The partial overlap with the epitope recognized by Sb2 illustrates the competitive relationship between these interacting molecules, as observed in displacement assays and SMFS (Figs. 1b and 2d and Extended Data Fig. 3g,k). Although conserved among the TTYH2 orthologues, this region varied among the three human family members, which reflects the supposed specificity of the interaction (Extended Data Fig. 4f). Consequently, we did not find similar density of the apolipoprotein after the addition of gold-labelled APOE to TTYH3 and to a mutant of TTYH2, in which four residues of the presumed binding site were mutated to their equivalent positions in TTYH3 (Extended Data Fig. 4g,h). Likewise, a high-resolution structure of TTYH3 that was incubated with unlabelled APOE did not contain extra density in this region (Extended Data Table 1 and Extended Data Fig. 5a–f). Analyses of structures of TTYH2 with a construct of the C-terminal domain of APOE revealed extra density that resembled the data obtained with the full-length protein. This result suggests that this domain mediates the observed interaction between APOE and TTYH2 (Fig. 3c, Extended Data Table 1 and Extended Data Fig. 5g–j).

A generally similar interaction mechanism for TTYH2 and APOE was obtained from preparations in which both APOE and TTYH2 were coexpressed in HEK293 cells. Complexes were obtained after affinity purification of TTYH2, from which a fraction of the purified particles contained the bound lipoprotein in an uncontrolled but probably poorly lipidated state. Cryo-EM density from three independent datasets, extending to about 3.6–4 Å each, provided a similar result as for the complex with APOE purified from *E. coli*, in which we detected residual density at the tip of the extracellular domain of TTYH2 (Fig. 3d, Extended Data Table 1 and Extended Data Fig. 6). Collectively, we observed consistent interactions of largely delipidated APOE in complex with TTYH2 obtained from two different sample preparation strategies. In both cases, the

apolipoprotein formed contacts with residues specific to TTYH2 in the upper part of the extracellular domain, which suggests that binding is selective for this paralogue (Fig. 3b–d and Extended Data Fig. 4f).

Finally, we combined the large particle population that did not contain density attributed to APOE from all the obtained datasets to generate a structure of TTYH2 at 2.7 Å (Fig. 3e, Extended Data Table 1 and Extended Data Fig. 7a–c). In this high-resolution map, we were able to assign residual density surrounding the membrane-inserted part of TTYH2 and filling the hydrophobic pocket extending towards the extracellular side to phospholipids and either cholesterol or the sterol-like moieties of the detergent (Fig. 3e and Extended Data Fig. 7d). The distorted yet largely continuous lipid belt reaching from the outer leaflet of the bilayer to the hydrophobic pocket emerging from the membrane implicated that there was free diffusion of lipids between both compartments (Fig. 3e and Extended Data Fig. 7d). During this process, the predominantly vertical position of lipids in the bilayer gradually changed into a horizontal orientation found in the cavity (Fig. 3f and Extended Data Fig. 7d–f). This transition is presumably facilitated by the N-terminal helix of TTYH2 preceding the first membrane-spanning segment, which is peripherally attached to the membrane and surrounded by a lipid-like density with a horizontal orientation (Extended Data Fig. 7e). The high-resolution structure illustrates how the architecture of TTYH2 might facilitate the extraction and insertion of lipids into the bilayer by providing a suitable pathway from its outer leaflet towards the extracellular environment.

## TTYH2–lipoprotein complexes

After identifying the region of interaction between TTYH2 and unlipidated APOE, we were interested in whether the protein interacts with APOE-containing lipoprotein particles in a similar manner. To this end, we reconstituted disc-shaped particles using the same protocol as described for the competition experiments. These particles were added to either detergent-solubilized and purified TTYH2 or cell-derived vesicles containing overexpressed TTYH2 before vitrification and cryo-EM data collection. Datasets of two independent samples with detergent-solubilized

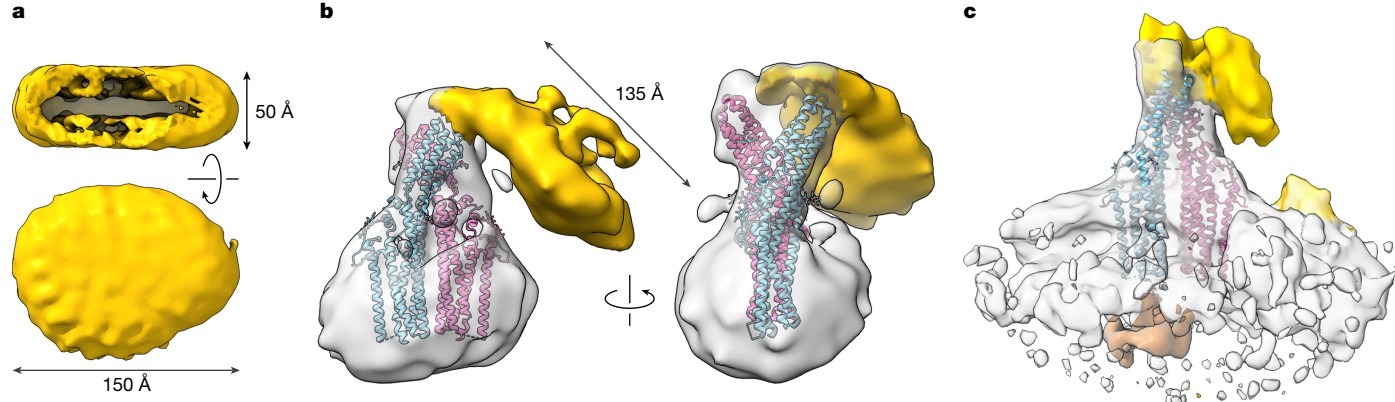

**Fig. 4 | Structure of TTYH2 in complex with disc-shaped APOE lipoproteins.**
**a,b**, Structures from samples containing TTYH2 in detergent and reconstituted APOE lipoproteins. **a**, Representative structure of a lipoprotein disc (150 × 110 × 50 Å). The top view is along the disc and shows the high density of the polar head-group regions of the bilayer. The bottom view is towards the plane of the bilayer. **b**, Low-resolution reconstruction of TTYH2 in complex with a lipoprotein disc at 14 Å viewed from two different directions. The disc sharing the dimensions of isolated lipoproteins binds to the extracellular domain of TTYH2 in vicinity to the binding region of delipidated APOE and is oriented towards the membrane. **c**, Structure of a TTYH2–APOE–lipoprotein complex obtained from cell-derived vesicles at 10.7 Å, with the density corresponding to the cytoplasmic region in orange. In **b** and **c**, TTYH2 is shown as a ribbon. For **a–c**, lipoprotein discs are in yellow.

TTYH2 showed generally similar features, with a heterogeneous particle distribution that was readily apparent in 2D classifications (Extended Data Fig. 8a). In addition to the familiar TTYH2 structure, other populations corresponded to lipoproteins and their complexes with TTYH2 (Extended Data Table 1 and Extended Data Fig. 8). The 3D reconstruction produced three distinct structures of oval-shaped disc-like assemblies, with diameters ranging between 130 and 150 Å, which was expected for assemblies of APOE with cholesterol and phospholipids (Fig. 4a and Extended Data Fig. 8a,d). All showed a similar bimodal distribution of density corresponding to the lipid head-group region at an appropriate distance for a lipid bilayer (Fig. 4a) and resembled previously described structures of APOE-containing lipoprotein discs[16].

For a large subpopulation of 3D classes that contained TTYH2, additional density indicated the binding of lipoproteins (Extended Data Fig. 8a). After further 3D classification, reconstructions at a resolution between 13 Å and 14 Å clearly displayed density of the bound lipoprotein particles with dimensions that resembled the size of free particles (Fig. 4b and Extended Data Fig. 8e,f). In these maps, one part of the disc-shaped lipoprotein was attached to the extracellular domain, in a similar region as observed for the lipid-free APOE, although details of the binding position could not be assigned owing to the limited resolution (Fig. 4b and Extended Data Fig. 8e,f). The remainder of the disc was oriented towards the membrane domain of TTYH2, in proximity to the hydrophobic cavity that is filled with lipid-like density as defined in previously determined structures of TTYHs[1,24] and the high-resolution structure of TTYH2 obtained in this study (Figs. 3e,f and 4b and Extended Data Fig. 7). A generally similar mechanism of the lipoprotein interaction was found in structures of TTYH2 in cell-derived vesicles (Fig. 4c, Extended Data Table 1 and Extended Data Fig. 9a–d), which illustrates that this interaction is preserved in the membrane-embedded protein. In both cases, the arrangement of APOE-containing lipoproteins and TTYH2 would bring lipids in proximity to the hydrophobic extracellular cavity. This result is in line with the proposed role of the TTYH2 in facilitating lipid transfer between the lipoprotein particle and the membrane.

## Lipid-transfer assays

After identifying how TTYH2 interacts with APOE in different lipidation states, we were interested in whether this interaction enhances the exchange of lipids between lipoproteins and the cell membrane. The transfer of fluorescent lipids in in vitro assays is compromised by the large background fluorescence that stems from the nonspecific interactions of APOE-containing lipoprotein particles with liposomes, which masks any additional activity conferred by TTYH2 (ref. 34). We therefore turned to membrane systems based on the lipid dipalmitoylphosphocholine (DPPC). DPPC has two fully saturated fatty acid chains and decreases the fluidity of the bilayer by raising the transition temperature from a lipid-ordered into a lipid-disordered phase. Lipid membranes composed of DPPC show low background binding of APOE[35], which make them suitable systems for the investigation of lipid transfer with TTYH2 reconstituted into liposomes. In our assay, we monitored the fluorescence decay of NBD-labelled phosphatidylethanolamine (PE), initially in lipoprotein particles through its quenching by rhodamine-labelled lipids contained in the liposomes (Fig. 5a). To optimize sensitivity, the DPPC content of liposomes was varied by mixing it with 1-palmitoyl-2-oleoylphosphatidylcholine (POPC). In proteoliposomes with a 85% DPPC content, we detected a substantial (14-fold) acceleration of lipid transfer mediated by TTYH2 (Fig. 5b and Extended Data Fig. 9e–h), which strongly exceeded the effect obtained from the tethering of APOE-containing lipoproteins to the membrane (Extended Data Fig. 9e,f). A much slower rate was observed for liposomes that contained reconstituted TTYH3, a result that further emphasizes the paralogue-specific mechanism of lipid transfer (Extended Data Fig. 9g,h). Together, these experiments provide support for the idea that TTYH2 is a facilitator of lipid exchange between soluble lipoprotein complexes and the membrane. We also revisited a potential function of TTYH2 as a scramblase that catalyses lipid transitions between membrane leaflets. However, we did not find convincing evidence of this function in either reconstituted systems or cellular assays (Extended Data Fig. 9i–l), results that are in line with those obtained from previous studies[1]. Finally, we studied HEK293 cells incubated with APOE-containing lipoproteins with fluorescently labelled PE. There was a distinct intracellular distribution of the labelled lipids and a partial overlap with TTYH2, which indicated that endocytosed phospholipids can be retained in the same compartment (Extended Data Fig. 9m). These results provide evidence for the colocalization of endocytosed lipids with TTYH2, as expected for its presumed role as an endosomal lipid transfer catalyst.

## Discussion

Our study identified APOE as an interaction partner of TTYH2 and provides evidence for the role of the latter as a lipid-transfer catalyst. The original identification of such an interaction in a kidney-derived cell line (Extended Data Fig. 2a) points towards a ubiquitous process. Confocal microscopy and subcellular fractionation data obtained from

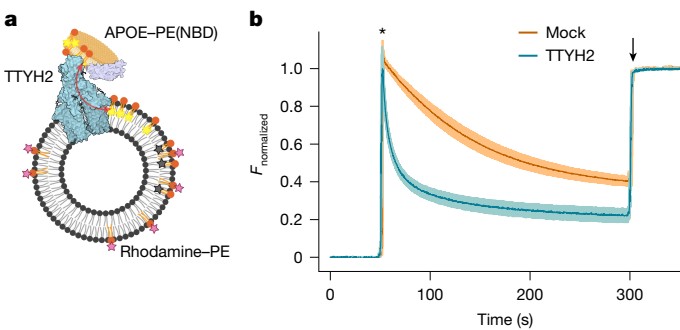

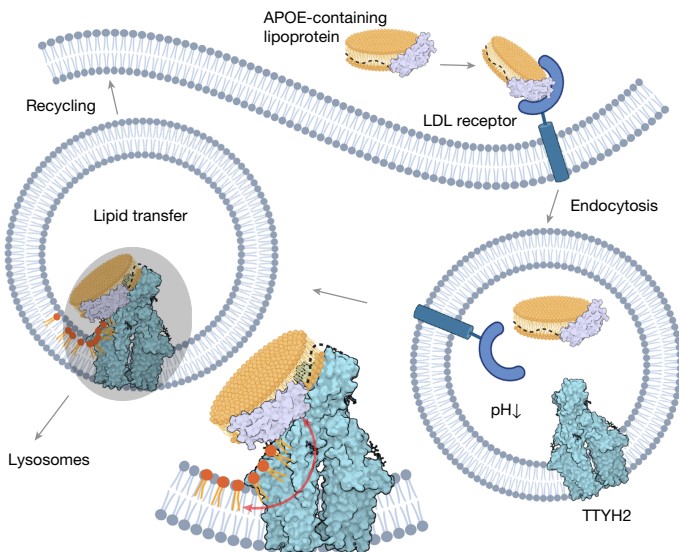

**Fig. 5 | TTYH2-mediated lipid transfer. a**, Scheme of the lipid-transfer assay monitoring the decay of NBD fluorescence from labelled donor lipids reconstituted into APOE-containing lipoprotein particles (APOE–PE(NBD)) quenched by rhodamine-labelled acceptor lipids (rhodamine–PE) in liposomes containing 85% DPPC. **b**, TTYH2-enhanced lipid transfer (cyan) in comparison to the transfer in mock liposomes (orange) as control. Normalized fluorescence ($F_{normalized}$) traces show mean of $n = 5$ (TTYH2) and $n = 4$ (mock) experiments from 2 independent reconstitutions. The asterisk indicates the addition of NBD-labelled lipoprotein discs, and the arrow indicates the addition of Triton X-100, which led to the dissolution of liposomes and lipoproteins. Error bars are the s.e.m. The diagram in **a** was created using BioRender (https://www.biorender.com).

**Fig. 6 | Proposed cellular role of TTYH2–APOE interactions.** Schematic of the processes that lead to the encounter of TTYH2 and APOE-containing lipoprotein particles in endosomes after their receptor-mediated endocytosis and dissociation from the low-density lipoprotein (LDL) receptor after acidification of the endosomal lumen. The interaction of both proteins facilitates the exchange of lipids between the lipoprotein and the endosomal membrane, as illustrated by the magnified diagram of the shaded area (left). The diagram was created using BioRender (https://www.biorender.com).

HEK293 and N2A cells, which showed that both proteins are located in endosomal compartments, provided evidence for their interaction in a cellular environment emphasizing the physiological relevance of the process (Fig. 2a–c and Extended Data Figs. 2c–e and 3a–c). This finding generally agrees with the recently reported localization of a *Drosophila* homologue of TTYH in the endolysosomal system[21], and with other studies in which TTYHs were also found in lysosomes[36–38]. The intracellular expression of these proteins argues against the idea that TTYH2 encounters APOE on the cell surface and instead suggests the involvement of a downstream process that occurs after the recognition of the apolipoprotein by a specific receptor that leads to its endocytosis[17,39]. This mechanism would enable the interaction of both proteins in the confined environment of the endosomal lumen. The lower protein complexity in the lumen compared with the extracellular fluid and the enrichment of the endocytosed target, both loosen the affinity- and specificity-requirements for the interaction.

Our structural and functional data demonstrated the interaction of TTYH2 with APOE in different lipidation states. However, these data are limited by several technical challenges. Binding studies were complicated by the propensity of APOE to interact with hydrophobic surfaces. Structural studies were hampered by the conformational heterogeneity of the apolipoprotein and its lipid complexes, which precluded structural characterization of APOE at high resolution. Finally, lipid-transfer studies were hindered by the nonspecific interactions of lipoproteins with cellular membranes. However, despite these limitations, our work provided insights that were supported by complementary observations. The confinement of the TTYH2-binding site of APOE to its C-terminal domain (Figs. 2d and 3c) distinguishes this interaction from that with the low-density lipoprotein receptor, which is mediated by the N-terminal domain[9,10]. This argues against the likelihood of substantial APOE subtype specificity, as the residues that distinguish its isoforms reside on the N-terminal domain[12]. However, a definitive answer to this question requires further experimental investigation. The location of the binding site of APOE at the extracellular domain of TTYH2 (facing the endosomal lumen) was revealed by cryo-EM maps, which displayed a density of delipidated APOE and its C-terminal domain in the same region (Fig. 3a–d). Although the limited resolution of the density, owing to the heterogeneous binding of APOE, prevented definitive assignment of contact residues, it implicates the involvement of a region that is highly conserved among TTYH2 orthologues. This region is different

among the three human TTYH paralogues, which provides further evidence for the specificity of the interaction and is further supported by the absence of APOE density in the structure of TTYH3 (Extended Data Figs. 4f–h and 5a–f). Finally, our data provided insight into the binding of APOE-containing lipoprotein particles to TTYH2 and its relevance for mediating lipid transfer. Cryo-EM data in detergent and native vesicles revealed the location of disc-shaped APOE-containing lipoprotein particles at the tip of the extracellular domain to bring lipids in proximity of an extended hydrophobic cavity of TTYH2 that emerges from the lipid bilayer (Fig. 4b,c). A role of this cavity for lipid transfer was initially proposed on the basis of the presence of lipid-like densities found in all three TTYH paralogues[1]. Its relevance was further supported by the structure of TTYH2 at high resolution obtained in this study, which revealed continuous lipid distribution emerging from the outer membrane leaflet into the cavity, with the protein facilitating lipid reorientation (Fig. 3e,f and Extended Data Fig. 7d–f). Additional evidence for TTYH2 acting as catalyst of lipid transfer was obtained from different experimental strategies. Cellular studies showed signal overlap in enriched TTYH2 environments with endocytosed PE lipids, which indicated a potential interaction under physiological conditions. Indeed, the presence of cluster-like structures pointed towards the colocalization of TTYH2 and PE lipids delivered to cells by APOE (Extended Data Fig. 9m). Similarly, in vitro studies demonstrated that suitably labelled lipids incorporated in APOE-containing lipoprotein particles encountered lipids located in TTYH2-containing liposomes (Fig. 5). The latter experiments showed a strong acceleration of lipid transfer compared with empty liposomes and proteoliposomes that contained TTYH3, which further illustrated the specificity of the process (Fig. 5b and Extended Data Fig. 9g,h).

Collectively, our results suggest that TTYH2 has a role in the unloading and potential reloading of endocytosed lipoprotein particles (Fig. 6). In our model, the uptake of these particles is initiated when they bind to specific receptors at the cell surface. As a result of clathrin-mediated endocytosis and the subsequent fusion with other endosomal compartments, the lipoprotein particles are brought into proximity with TTYH2. Acidification in the endosomal lumen leads to

the dissociation of the lipoprotein from its receptor, which enables interaction with TTYH2 to facilitate the transfer of lipids to the endosomal membrane (or vice versa) (Fig. 6). Although a net transfer of lipids to the nearby leaflet would lead to an imbalance that ultimately requires relaxation by a lipid scramblase, we did not find evidence that TTYH2 itself would be the protein that mediates this process (Extended Data Fig. 9i–l). Although the described transfer process seems to be ubiquitous, it is likely to be of particular importance in the brain, where the shuttling of lipids from astrocytes to neurons is essential[5]. After unloading or lipid exchange, a fraction of APOE is presumably recycled or transported for lysosomal degradation. In the described process, we do not anticipate strong discrimination between APOE isoforms nor do we have evidence for substantial lipid specificity. However, in light of the importance of APOE as the predominant apolipoprotein in the brain[6], a role of TTYH2 in the unloading of certain brain lipids, such as glycosphingolipids or ether lipids for which transfer would encounter a larger energy barrier, could be of particular relevance. Moreover, in light of the isoform-dependence of APOE in the predisposition to neurodegenerative diseases[7], an involvement of TTYH2 in pathological processes is possible. The mechanism of TTYH2-mediated lipid transfer distantly resembles other systems in which membrane proteins facilitate the exchange of lipids between bilayers and soluble lipid carriers such as the lysosomal NPC sterol transporters[40] or the ATG2–ATG9 system, which channels lipids during autophagosome formation[41,42].

In summary, we showed that TTYH2 is involved in endosomal lipid transfer; however, its role now requires further evidence from cellular studies. Similarly, identification of interaction partners of the paralogues TTYH1 and TTYH3 is needed, as these partners might contribute to related functions. Finally, the role of the C-terminal domain of TTYH2 requires further investigation to better understand the involvement of TTYHs in interactions on the cytoplasmic side. Despite these open questions, our work provides new insight into a previously uncharacterized lipid transport mechanism and a foundation for future investigations.

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

## Methods

### Antibodies

All antibodies were purchased from Thermo Fisher Scientific unless specified otherwise. Primary antibodies for western blotting were used at a 1:4,000 dilution and secondary antibodies at 1:10,000 dilution. The following primary antibodies were used for western blotting: rabbit anti-TTYH2 (PA5-34395) raised against a 14-amino-acid N-terminal peptide; rabbit anti-APOE (16H22L18); rabbit anti-EEA1 (F.43.1); rabbit anti-LAMP1 (107); rabbit anti-RAB11 (20229-1-AP, Proteintech); rabbit anti-Na,K-ATPase (ST0533); rabbit anti-SERCA (JM10-20); rabbit anti-RAB7 (PA5-23138); and rabbit anti-GM130 (ARC0589). A goat anti-rabbit HRP-conjugated antibody (31460) was used as the secondary antibody for western blotting. The primary antibody used for immunocytochemistry was a rabbit anti-human-TTYH2 antibody (1:500 dilution; Antibodies-online.com) raised against the 455–534-amino-acid C-terminal peptide and a mouse anti-human-RAB9 (1:500 dilution; Mab9). The secondary antibodies used for immunohistochemistry was a goat anti-rabbit Alexa 488-conjugated antibody (1:200 dilution, A-11008) and goat anti-mouse Alexa 594-conjugated antibody (1:200 dilution, A-11032). The recognition of TTYH2 was confirmed for both anti-TTYH2 primary antibodies on purified protein (Supplementary Fig. 1a,b).

### Mammalian and bacterial cell culture and strains

Suspension HEK293 GNTI⁻ cells were grown in HyCell TransFx-H (Cytiva) medium supplemented with 1% FBS, 4 mM L-glutamine, 0.4% Poloxamer 188 and 100 U ml⁻¹ penicillin–streptomycin at 37 °C and 5% $CO_2$ while shaking. Proteins were expressed for 60 h unless specified otherwise. Adherent neuroblastoma cells (N2A, American Type Culture Collection) were grown in 10 cm dishes in Eagle's minimum essential medium (Merck) supplemented with 10% FBS, 2 mM L-glutamine, 100 U ml⁻¹ penicillin–streptomycin, 1 mM sodium pyruvate and nonessential amino acids at 37 °C and 5% $CO_2$. Adherent HEK293T cells were grown in DMEM (Gibco) supplemented with 10% FBS and 100 U ml⁻¹ penicillin–streptomycin at 37 °C and 5% $CO_2$. N2A adherent cells were grown to full confluency before transfer to 245 × 245 mm dishes (Corning). The cells were further grown on these dishes for 2–3 days to 70–90% confluency and used for subcellular fractionation. The HEK293 TMEM16F knockout cells (provided by H. Yang) used for the cellular scrambling assays were grown in DMEM (Gibco) supplemented with 10% FBS and 100 U ml⁻¹ penicillin–streptomycin at 37 °C and 5% $CO_2$ in 10 cm dishes. All mammalian cell lines used in this study tested negative for mycoplasma infection.

For DNA preparation and protein expression, bacterial cells were grown in Terrific broth supplemented with 0.6% (v/v) glycerol and selection antibiotics with shaking. MC1061 chemically competent cells were used for the expression of sybodies. Chemically competent BL21 (DE3) cells were used for APOE expression. All DNA preparations were done in MC1061 cells.

### Construct expression and purification

All cloning steps were carried out in suitably modified vectors using FX cloning[43] or QuickChange mutagenesis (Stratagene, Agilent). Genes encoding human TTYH2 and TTYH3 (Genscript) were cloned into a pcDXC3MS vector (Addgene, 49030, suitable for protein expression in mammalian cells) and used for transient transfection of suspension HEK293 GNTI⁻ cells. The TTYH2 in the pcDXC3MS construct was used as a template to generate the TTYH2(G165P/D166E/Q169R/F173R) mutant construct through site-directed mutagenesis[44]. The cells were transfected at a density of 1.5–2 × 10⁶ cells per ml with PEI MAX, and valproic acid (4 mM) was added to stop cell division. After 60 h of expression, cells were collected and washed in PBS. Cell pellets were flash-frozen in liquid nitrogen and stored at −80 °C. On the day of purification, pellets were thawed and resuspended in lysis buffer (20 mM HEPES,

pH 7.4, 200 mM NaCl, 2% (w/w) GDN, DNAse and protease inhibitors) at a ratio of 10 ml lysis buffer per 10 ml cell pellet. The resuspended cells were lysed for 1–2 h at 4 °C while rotating. The lysate was centrifuged at 15,000g for 30 min at 4 °C. The filtered (pore size of 5 μm) supernatant was added to 2 ml of Strep-Tactin resin per 50 ml cell pellet pre-equilibrated with SEC buffer (10 mM HEPES, pH 7.4, 200 mM NaCl and 50 μM GDN) and incubated in batches for 1 h at 4 °C while rotating. The flow-through was discarded and the resin was washed with 50 column volumes (CV) of SEC buffer. The protein was eluted with 5 CV of SEC buffer supplemented with 5 mM desthiobiotin, concentrated and loaded on a Superose 6 10/300 GL column for size-exclusion chromatography (SEC) in SEC buffer. Fractions corresponding to the protein peak were pooled, concentrated and used either fresh for the preparation of cryo-EM grids and reconstitution into liposomes or flash-frozen in liquid nitrogen with the addition of 10% glycerol and stored at −80 °C for later use in sybody-binding or APOE-displacement assays.

TTYH2 purified for sybody selection was concentrated after elution from Strep-Tactin resin and chemically biotinylated using EZ-link NHS-PEG4 biotin. Biotin was added at a 10-fold molar excess over TTYH2 monomer and the mixture was incubated for 1 h at 4 °C. Subsequently, Tris pH 7.4 was added to reach a concentration of 5 mM to quench the reaction and the tag was cleaved by the addition of 3C protease and incubation of the mixture on ice overnight. The next morning, biotinylated and cleaved TTYH2 was purified by SEC on a Superose 6 10/300 GL column to separate free biotin and the 3C protease. The peak fractions were pooled and concentrated, flash-frozen in liquid nitrogen after the addition of 10% glycerol and stored at −80 °C until use for sybody selection. Human ferroportin, used as a negative control in ELISA assays during sybody selections, was expressed, purified and biotinylated in the same way, except that the lysis buffer contained 2% and all other buffers with 0.04% (w/w) DDM instead of GDN.

The gene encoding human APOE3 including its signal peptide (Genscript) was cloned into a custom pcDX vector for mammalian expression fusing a 3C cleavage site and a His₆ tag on the C terminus. This construct was used for the coexpression of both APOE and TTYH2 in HEK293 GNTI⁻ cells. The cells were transfected with plasmids containing APOE and TTYH2 mixed in 1.1:1 molar ratio at a cell density of 1.5–2 × 10⁶ cells per ml with PEI MAX. Valproic acid (4 mM) was added to stop cell division. A typical preparation for the coexpression of both proteins was carried out from a 0.6–0.9 l culture of transfected cells. After 38 h of expression, cells were collected and washed in PBS. Fresh cell pellets were used for the purification of the TTYH2–APOE complex. Cell lysis was performed as described above and cleared lysate was manually loaded onto a 1 ml bed of Strep-Tactin resin. The resin was washed with 15 ml SEC buffer and the protein complex was eluted in 5 ml SEC buffer supplemented with 5 mM desthiobiotin. The eluted complex was concentrated to 200 μl and separated on a Superose 6 Increase 5/150 GL column sequentially through the injection of 50 μl aliquots per run. The small column size was selected to shorten the elution time and to reduce the dissociation of the TTYH2–APOE complex. The peak fractions were concentrated to 1 mg ml⁻¹ and used for the freezing of cryo-EM grids.

For its expression in *E. coli*, *APOE3* was subcloned into p7XNH3 vector (Addgene, 47064) with the signal peptide (residues 1–18) removed. For lipid-transfer experiments in liposomes containing DGS-NTA(Ni) lipid, APOE3 in the p7XNH3 construct was modified by introducing a GSGSGSGSG linker between the 3C recognition site and the *APOE3* gene. APOE3 was expressed in 3 l of BL21 cells for 3.5 h after induction. The cells were collected and cell pellets were flash-frozen in liquid nitrogen and stored at −80 °C. APOE was purified in its delipidated state using the strategy described for MSP purification[45] but with minor adjustments. Frozen pellets were thawed and resuspended in 50 ml lysis buffer (50 mM Tris, pH 8, 500 mM NaCl, 1% Triton X-100, DNAse and protease inhibitors). The resuspended cells were lysed with a HPL6 high-pressure homogenizer (Maximator) and centrifuged

at 15,000$g$ for 30 min at 4 °C. The cleared lysate was loaded on 4 ml Ni-NTA bed resin pre-equilibrated in buffer containing 50 mM Tris, pH 8, 500 mM NaCl and 1% Triton X-100 by batch binding under rotation for 1 h at 4 °C. The flow-through was discarded and the resin was first washed with 50 ml of the equilibration buffer, followed by subsequent washing steps with 50 ml equilibration buffer containing 50 mM sodium cholate instead of Triton X-100 and with 50 ml equilibration buffer without detergent supplemented with 30 mM imidazole. The protein was eluted in 15 ml buffer containing 50 mM Tris, pH 8, 500 mM NaCl and 500 mM imidazole and the His$_{10}$-tag was cleaved by the addition of 3C protease and incubation for 15 min at 4 °C. The cleaved protein was concentrated and loaded on a Superose 6 10/300 GL column equilibrated in 10 mM HEPES, pH 7.4 and 200 mM NaCl buffer. The peak fractions corresponding to the tetrameric delipidated APOE were pooled, concentrated and used for reconstitution into lipoprotein particles or flash-frozen in liquid nitrogen and stored at −80 °C to be used in displacement assays or for cryo-EM preparations. N-terminal (containing residues 19–209) and C-terminal (containing residues 191–317) fragments of APOE were subcloned into p7XNH3 vector, expressed and purified in the same way as described for the full-length construct.

For labelling with nanogold, APOE3 was purified as described above but without cleaving the His-tag. Next, 1.8 nm Ni-NTA-Nanogold (Nanoprobes) was mixed with APOE at a nanogold to APOE molar ratio of 1:5 and incubated for 15 min at room temperature. A low nanogold concentration was chosen to reduce nonspecific binding. Unbound nanogold was separated from the APOE–nanogold complex using a Sephadex G50 column. The nanogold-labelled APOE was mixed with purified TTYH2, TTYH3 or the TTYH2(G165P/D166E/Q169R/F173R) mutant at a 1:1 molar ratio and incubated on ice for 30 min. The complex was purified by SEC using a Superose 6 Increase 5/150 GL column. Peak fractions corresponding to the respective TTYH dimer were pooled, concentrated and used for the preparation of cryo-EM grids.

For lipid-transfer experiments in liposomes containing DGS-NTA(Ni) lipid, APOE was expressed in the modified p7XNH3 vector with the extended linker between the 3C recognition site and the *APOE3* gene. The protein was purified as described above but without cleaving the His-tag. Sybodies were purified as previously described[25]. In brief, sybody constructs in a pSbinit (Addgene, 110100) vector containing a pelB signal sequence for periplasmic expression attached to their N terminus were expressed in 1 l of MC1061 *E. coli* culture for 14 h. The cells were collected and flash-frozen or used directly for purification. Cell pellets were resuspended in 50 ml TBS and lysed using a HPL6 high-pressure homogenizer (Maximator). The lysate was centrifuged at 15,000$g$ for 30 min at 4 °C. The cleared lysate was used for batch binding on Ni-NTA resin with 4 ml bed volume pre-equilibrated in TBS under rotation for 1 h at 4 °C. The flow-through was discarded and the resin was washed with 50 ml TBS containing 30 mM imidazole. Sybodies were eluted in 15 ml TBS containing 500 mM imidazole, concentrated and purified by SEC on a Superdex 200 10/300 GL column. The peak fractions were pooled and concentrated to final protein concentration of 0.5–1 mM, flash-frozen and stored at −80 °C. For the isolation of endogenous TTYH2 and for the purification of TTYH2-containing cell-derived vesicles, Syb1 cloned in a pSbinit vector construct was modified for purification on Strep-Tactin resin. A Strep-Tactin-binding protein sequence was added at the C terminus following the His-tag sequence. Syb2 was subcloned into the pcDXC3VMS vector for mammalian expression containing a Venus fluorescent tag sequence on the C terminus following the 3C cleavage site. The 3C cleavage site was deleted for an uncleavable fusion of the sybody to Venus. The sybody was expressed in HEK293 GNTI$^-$ cells for 60 h. The cells were collected, washed in PBS and cell pellets were either flash-frozen in liquid nitrogen and stored at −80 °C or used directly for purification. The sybody was purified as described above.

## Selection of synthetic nanobodies against TTYH2

TTYH2 in a pcDXC3MS vector was expressed in 3 l HEK293 GNTI$^-$ cells, purified and biotinylated as described above. The selection was carried out using mRNA libraries and vectors provided by M. Seeger as previously described[25]. In brief, chemically biotinylated TTYH2 (with 50% efficiency) was used in one round of ribosome display with concave, loop and convex synthetic libraries encoding synthetic nanobodies (termed sybodies), which primarily differ in the length of the CDR3 region. Each library contained around 10$^{12}$ binders at the onset of the selection. The ribosome display output from the three libraries containing the DNA of captured sybodies was recloned into a vector for phage production and used for two rounds of phage display. The phage display output containing DNA of captured sybodies was subsequently subcloned into a pSbinit vector for sybody expression, and an initial pool of selected binders was identified by ELISA with TTYH2 as the target protein and ferroportin as the negative control. Clones with the highest signal over background were sequenced. Sybodies that showed promising biochemical properties were tested for their binding to TTYH2 by SEC (Supplementary Fig. 1b–d). TTYH2 supplemented with a 1.6 molar excess of the respective sybody was loaded on a Superose 6 Increase 5/150 GL column and the presence of the sybody in fractions containing TTYH2 was detected by SDS–PAGE. In this way, it was possible to isolate TTYH2 binders from all three libraries, two of which were used in this study, namely Sb1 from the concave library (with a short CDR3) and Sb2 from the loop library (with a medium CDR3).

## Isolation of endogenous TTYH2 complexes

To isolate endogenously expressed TTYH2 in complex with potential interaction partners, 1.8 l HEK293 GNTI$^-$ cells at a density of $4 \times 10^6$ cells per ml was used. The cells were collected, washed in PBS and resuspended in 80 ml lysis buffer (20 mM HEPES pH 7.4, 200 mM NaCl and 2% (w/w) GDN). The lysate was incubated at 4 °C under rotation for 1 h. Next, 2.2 mg Sb1 was immobilized on Strep-Tactin resin with 0.5 ml bed volume by batch binding for 1 h at 4 °C under rotation. The resin was drained and the excess sybody was removed with 5 ml SEC buffer. The cell lysate was centrifuged at 15,000$g$ for 30 min at 4 °C and manually loaded onto the resin containing immobilized Sb1. The resin was washed with 50 ml SEC buffer. The sybody-bound TTYH2 complexes were eluted with 2.5 ml SEC buffer supplemented with 5 mM desthiobiotin. Eluted complexes were concentrated using a centrifugal filter with 3 kDa MW cut-off and analysed by liquid chromatography coupled to mass spectrometry for identification of potential interaction partners (carried out by the Functional Genomics Center Zurich).

## Subcellular fractionation

To investigate the cellular localization of TTYH2, we fractionated subcellular compartments by density centrifugation on a step sucrose gradient. To this end, 600 ml HEK293 GNTI$^-$ cells at a density of $2 \times 10^6$ cells per ml or neuroblastoma cells grown on two $245 \times 245$ cm plates to confluency were used per experiment. The cells were collected, washed in PBS and resuspended at a 1:1 volume ratio in buffer containing 8.25% (w/w) sucrose, 10 mM HEPES pH 7.4, 1.5 mM MgCl$_2$, 30 µM cycloheximide and protease inhibitors. The cytoplasmic content was released by dounce homogenization with 15 strokes on ice. The homogenized cells were centrifuged twice at 2,000$g$ for 10 min at 4 °C to separate the nuclear fraction. The post-nuclear supernatant (PNS) was applied on top of a step sucrose gradient. The gradient was created by layering 1 ml aliquots of buffer (10 mM HEPES pH 7.4, 1.5 mM MgCl$_2$ and 30 µM cycloheximide) with decreasing sucrose content as follows: 35%, 25%, 20%, PNS (8.25%) for HEK293 cells and 25%, 20%, 15%, PNS (8.25%) for N2A cells. It was necessary to introduce a step with 15% sucrose for N2A cells to better separate the plasma membrane fraction. The sample was centrifuged at 210,000$g$ for 3.5 h at 4 °C using 4.2 ml tubes and a SW 60 Ti swinging-bucket rotor (Beckman). The fractions were collected

and analysed by western blotting with antibodies against TTYH2, APOE and organelle markers (Supplementary Figs. 2–6).

## Reconstitution of lipidated APOE

All lipids in this study were purchased from Avanti Polar Lipids. Lipids were used as a chloroform solution and prepared by evaporating chloroform under a nitrogen stream and washing with diethyl ether. Excess solvent was evaporated by desiccation for 1 h and the lipid film was resuspended to 10 mg ml$^{-1}$ in buffer containing 20 mM HEPES, pH 7.4, 100 mM KCl and 2 mM CaCl$_2$ by sonication. Lipids were flash-frozen and stored at −80 °C. For cryo-EM preparations with lipidated APOE, a lipid mix containing POPE, POPG, egg PC and cholesterol at a 3:1:1:0.5 w/w ratio was used. For APOE used in FRET-based lipid-transfer assays, the lipids consisted of soy polar extract (Avanti, 541602) with additional 5% (w/w) NBD-PE (tail labelled, Avanti, 810156P). For APOE used in immunocytochemistry experiments, the lipids consisted of soy polar extract with additional 15% (w/w) rhodamine–PE (head labelled, Avanti, 810150). For APOE–mCherry used in immunocytochemistry experiments, the lipids consisted of POPE, POPG, egg PC and cholesterol at a 3:1:1:0.5 w/w ratio. APOE was expressed in bacteria and purified as described above. Lipoproteins were prepared using the cholate dialysis method[46]. Lipids were solubilized in sodium cholate at a 1:1 molar ratio, and purified APOE concentrated to 4 mg ml$^{-1}$ was mixed with solubilized lipids at a 1:100 APOE to lipid molar ratio in a reaction volume of 0.3–0.7 ml. The mix was incubated for 16 h at 4 °C under rotation. The detergent was removed by dialysis in buffer (added at 5,000× higher volume) containing 10 mM Tris, pH 8 and 150 mM NaCl in two steps over the course of 2 days. The final lipoprotein complexes were separated from aggregates and degradation products by SEC on a Superose 6 10/300 GL column equilibrated with 10 mM HEPES, pH 7.4 and 200 mM NaCl. Fractions containing intact lipidated APOE were pooled, concentrated, flash-frozen and stored at −80 °C.

## APOE–mCherry purification and lipidation

The *APOE3* gene not containing a signal peptide was subcloned into a custom pcDx vector for mammalian expression as an N-terminal mCherry fusion protein. The vector contained a streptavidin-binding protein tag, a MYC tag, the mCherry sequence and a 3C cleavage site. HEK293 GNTI⁻ cells were transfected at a density of 1.2 × 10$^6$ per ml with PEI MAX, and valproic acid was added to stop cell division. After 60 h of expression, cells were collected and washed in PBS, and cell pellets were used directly for APOE–mCherry purification per the procedure described for APOE purification from *E. coli* cells. After purification by SEC, fractions containing APOE were pooled and concentrated. Lipidation was carried out using a lipid mix containing POPE, POPG, egg PC and cholesterol at a 3:1:1:0.5 w/w ratio with the cholate dialysis method as described above. After dialysis, lipidated APOE–mCherry was directly used for incubation with cells.

## Immunocytochemistry and confocal microscopy

Adherent HEK293 cells were seeded in a 12-well plate with round cover slips placed inside each well to a confluency of 50–80%. The cells were washed in PBS and fixed by incubation in 4% paraformaldehyde in PBS for 15 min. The reaction was quenched by adding 10 mM glycine in PBS for 10 min. Cells were permeabilized with 0.1% Triton X-100 in PBS for 10 min. The excess detergent was washed off and the cells were blocked in 2% BSA–PBS for 15 min. The cells were incubated with the first antibody diluted in 2% BSA–PBS for 2 h. Excess antibody was washed off and a secondary antibody diluted in 2% BSA–PBS was added and incubated for 1 h. The cells were washed with 2% BSA–PBS and the cover slips were mounted on microscope slides using Vectashield antifade mounting medium with DAPI (AdipoGen). For double-antibody staining, the cells were first stained with the anti-TTYH2 and the Alexa-488-conjugated anti-rabbit antibodies and then with the anti-RAB9 and the Alexa-594-conjugated anti-mouse antibodies.

For detection of colocalization of TTYH2 and APOE in HEK293 cells, in vitro lipidated APOE–mCherry was added at 0.9 µM to cells seeded in a 12-well plate and incubated for 20 min in an incubator at 37 °C and 5% CO$_2$. After incubation, cells were fixed and stained with anti-TTYH2 and Alexa-488-conjugated anti-rabbit antibodies as described above. For detection of colocalization of endocytosed lipids and TTYH2 in HEK293 cells, APOE lipidated in vitro using a lipid mix containing 15% rhodamine–PE was added at 2 µM to cells seeded in a 12-well plate and incubated for 30 min in an incubator at 37 °C and 5% CO$_2$. After the incubation, cells were fixed and stained with anti-TTYH2 and Alexa-488-conjugated anti-rabbit antibodies as described above. All samples were analysed using a Zeiss LSM 980 Airyscan inverted confocal laser scanning microscope at the Center for Microscopy and Image Analysis (ZMB) of the University of Zurich (UZH). z-stacks of images were acquired from multiple locations and processed in Fiji[47].

## Preparation of cell-derived vesicles containing TTYH2

For the preparation of cell-derived vesicles for structural studies, a construct of the human *TTYH2* gene in a pcDXC3MS vector not containing a Strep-Tactin-binding protein sequence was expressed in HEK293 GNTI⁻ cells for 60 h. A typical sample was obtained from 4 l of culture. Cells were initially collected and washed in PBS. Vesicles from total cell membranes were prepared as previously described[48] but with minor adjustments. Cells were resuspended in 100 ml buffer containing 20 mM HEPES, pH 7.4, 300 mM KCl, 1 mM MgCl$_2$, DNAse and protease inhibitors. The resuspended cells were dounce homogenized on ice with 30 strokes and then sonicated (on ice at 60% power with 4 × 30 s pulses interrupted by 30-s intervals). The sonicated lysate was centrifuged twice at 12,000$g$ for 10 min at 4 °C. After the first spin, 5 mM EDTA was added to the supernatant to prevent vesicle aggregation. The supernatant was loaded onto Q Sepharose resin with 20 ml bed volume pre-equilibrated with 20 mM HEPES, pH 7.4, 300 mM KCl, 1 mM MgCl$_2$ and 2 mM EDTA buffer to remove nucleic acids. The flow-through was collected and the resin was washed with 20 ml equilibration buffer. The wash and flow-through were pooled. Next, 2.5 mg Sb1 was immobilized on a 1.5 ml bed of Strep-Tactin resin by batch binding for 1 h at 4 °C while rotating. The excess sybody was washed away and the resin loaded with sybody was used to capture vesicles containing TTYH2 in the outside-out orientation. Pooled wash and flow-through fractions were mixed with the Sb1–Strep-Tactin resin and incubated for 1 h at 4 °C while rotating. Flow-through was discarded and the resin was washed with 100 ml of 20 mM HEPES, pH 7.4, 300 mM KCl and 2 mM EDTA buffer and vesicles were eluted with 15 ml wash buffer supplemented with 5 mM desthiobiotin. The vesicles were concentrated using a centrifugal filter with 100 kDa MW cut-off and used for the freezing of cryo-EM grids. For the sample containing lipidated APOE, the expression time was reduced to 40 h and vesicles were prepared as described above.

## Sybody-displacement assay

As classical binding experiments such as microscale thermophoresis turned out to be unsuitable (Supplementary Fig. 7), we probed the site specificity of APOE binding to TTYH2 using Sb2, which occupies a similar epitope. Sb2 was expressed as a fusion with Venus fluorescent protein on its C terminus in HEK293 GNTI⁻ cells and its displacement from the complex with TTYH2 was monitored by fluorescent SEC[49]. The Sb2–Venus construct was mixed with TTYH2 purified from 1.2 l HEK293 GNTI⁻ cells at a 3:1 molar ratio and incubated on ice for 30 min. The complex was subsequently purified by SEC on a Superose 6 10/300 GL column. Peak fractions at an appropriate elution volume corresponding to the complex (detected by the measurement of the absorption 280 nm and confirmed by SDS–PAGE) were pooled, kept on ice overnight and used for displacement assays the next day. Sb2 displacement was analysed using unlipidated APOE, its N-terminal domain and lipidated APOE. Displacement with unlabelled Sb2 served as the positive control. Every reaction contained 40 µl of the TTYH2–Sb2–Venus complex and

one of the three competitors at various concentrations. After 15 min of incubation on ice, each sample was injected onto a Superose 6 Increase 5/150 GL column equilibrated with SEC buffer and the fluorescence intensity of Sb2–Venus was recorded for 20 min. The displacement was quantified by the decrease in fluorescence of the complex and the concomitant increase in fluorescence of the free sybody fusion. The averaged values were compared with the displacement of Sb2–Venus by the unlabelled Sb2, which was assigned as 100%.

### Reconstitution of TTYHs into 85% DPPC liposomes

For the lipid-transfer assays, TTYH2 was reconstituted into liposomes with an 85% (w/w) DPPC content to reduce their nonspecific interaction with APOE. A lipid mix containing 85% DPPC, 14% POPC and 1% rhodamine–PE (head labelled, Avanti, 810150) (w/w) was prepared as described above and the lipid film was solubilized in 20 mM HEPES pH 7.4, 100 mM KCl, 2 mM $CaCl_2$ and 35 mM CHAPS by sonication at a lipid concentration of 10 mg ml$^{-1}$. For the experiments with APOE–His$_{10}$ tethering, the lipid mix contained 85% DPPC, 13.5% POPC, 1% rhodamine–PE and 0.5% DGS-NTA(Ni) (Avanti, 790404) (w/w). Lipids were flash-frozen and stored at −80 °C.

Before mixing with either purified TTYH2 or TTYH3, or buffer in case of mock liposomes, the lipids were diluted to 4 mg ml$^{-1}$ in the same buffer. Constructs of TTYH2 and TTYH3 in a pcDXC3MS vector were expressed in HEK293 GNTI$^-$ cells and purified as described above. The purified protein was concentrated after SEC to 2–4 mg ml$^{-1}$ and mixed with the CHAPS-solubilized lipids at a lipid to protein ratio of 50 (w/w). The mix was incubated for 15 min while rotating. To remove the detergent, 100 mg biobeads per 1 ml lipids was added and the mix was incubated for 30 min under rotation. Six additional aliquots of biobeads were added over the course of 3 days to ensure complete removal of the detergent. The entire reconstitution process was carried out at room temperature. Liposomes were collected by centrifugation at 200,000$g$ for 30 min at 21 °C and resuspended to 10 mg ml$^{-1}$ lipid in 20 mM HEPES pH 7.4, 100 mM KCl and 2 mM $CaCl_2$ buffer. Liposomes were flash-frozen and stored at −80 °C. The reconstitution efficiency was tested by SDS–PAGE and amounted to 20–30% for both TTYH2 and TTYH3 liposomes. Protein integrity was confirmed by analytical SEC of samples re-extracted from liposomes in detergent GDN.

### Lipid-transfer assay with fluorescent lipids

To detect lipid transfer between APOE and TTYH2, we used a liposome-based system using lipids containing complementary fluorophores that form a FRET pair. TTYH2 and TTYH3 were reconstituted into liposomes containing 1% rhodamine–PE, 85% DPPC and 14% POPC (w/w) as described above. APOE was lipidated in vitro using lipids from soy polar extract supplemented with 5% (w/w) NBD–PE as described above. The fluorescence of NBD was monitored using a Fluoromax Horiba spectrofluorometer with excitation at 460 nm, emission at 535 nm and a bandwidth of 5 nm. The signal was recorded every 0.1 s. A quartz cuvette was loaded with 2 ml liposomes extruded through a 400 nm membrane and diluted to 0.2 mg ml$^{-1}$ with 20 mM HEPES, pH 7.4, 100 mM KCl and 2 mM $CaCl_2$ buffer. After recording of a baseline for 50 s, lipidated APOE containing NBD–PE was added to 50 nM and NBD fluorescence was recorded for 250 s. After this time period, Triton X-100 was added to a concentration of 0.1% to completely solubilize liposomes and the NBD signal increase was recorded for another 100 s. The data were normalized using the following formula: $(F - F_{10})/(F_{400} - F_{100})$, where $F$ is the fluorescence intensity at every time point, $F_{10}$ is the fluorescence intensity at 10 s and $F_{400}$ is the fluorescence intensity at 400 s. To estimate the effect of APOE tethering on the observed lipid-transfer acceleration, we spiked the liposomes with the DGS-NTA(Ni) lipid and used APOE with a hexahistidine-tag, therefore forcing its binding to the liposome surface. The His-tagged APOE was lipidated in vitro using lipids from soy polar extract supplemented with 5% (w/w) NBD–PE as described above and was added to the liposomes containing 85% DPPC, 13.5% POPC, 1% rhodamine–PE and 0.5% DGS-NTA(Ni) lipids (w/w). NBD fluorescence was recorded and the data were analysed as described above.

### Scrambling assays

To probe whether TTYH2 catalyses lipid movement between the two bilayer leaflets, we used a liposome-based assay and a cellular assay. For the liposome-based assay, TTYH2 was reconstituted into liposomes containing lipids from soy polar extract, 20% cholesterol and 0.5% NBD–PE (head-labelled, Avanti, 810145). The liposomes were prepared either by solubilizing lipids in buffer containing 35 mM CHAPS and used for TTYH2 reconstitution as described above or by resuspending lipids in detergent-free buffer. The latter lipid batch was used for liposome reconstitution of TTYH2 by gradually destabilizing liposomes with small amounts of Triton-X 100 as previously described[1]. In brief, the lipids were extruded using a 400 nm membrane and diluted to 4 mg ml$^{-1}$ in the liposome buffer containing 20 mM HEPES pH 7.4, 100 mM KCl and 2 mM $CaCl_2$. The liposomes were titrated with 10% Triton-X 100 and destabilization was monitored by measuring absorbance at 540 nm. TTYH2 purified in detergent was added to the destabilized liposomes at a lipid to protein ratio of 100 (w/w) and detergent was gradually removed using biobeads. The liposomes were collected by centrifugation at 200,000$g$ and the liposome pellet was resuspended in the liposome buffer to 20 mg ml$^{-1}$ lipid, aliquoted and flash-frozen in liquid nitrogen for storage at −80 °C. The reconstitution efficiency was tested by SDS–PAGE and amounted to 30–40% for both solubilized and destabilized preparations. Protein integrity was confirmed by analytical SEC of samples re-extracted from liposomes in detergent GDN (Supplementary Fig. 8).

The liposome-based scrambling assay was performed as previously described[1]. NBD fluorescence was monitored using a Fluoromax Horiba spectrofluorometer with excitation at 460 nm, emission at 535 nm and a bandwidth of 2 nm. The signal was recorded every 0.1 s. A quartz cuvette was loaded with 2 ml of liposomes extruded using 400 nm membrane and diluted to 0.2 mg ml$^{-1}$ with the liposome buffer. After recording of a baseline for 50 s, freshly prepared sodium dithionite was added to 30 µM and NBD fluorescence was recorded for 350 s. The data were normalized using the formula: $F/F_{50}$, where $F_{50}$ is the fluorescence intensity at 50 s (Supplementary Fig. 8).

Cell-based scrambling assays[50] were performed using a HEK293 TMEM16F knockout cell line. Cells grown to 80–90% confluency were added to a 96-well polylysine-coated plate at a seeding density of 10%. The seeded cells were transfected with 100 ng DNA per well using Lipofectamine 3000 (ThermoFisher). The cells were transfected with pcDXC3MSV plasmid containing TTYH2, TMEM16F or the TMEM16F(F518H) constitutively active mutant, or an empty vector. The constructs contained Venus as a C-terminal tag for detection of transfected cells. At 48 h after transfection, the medium was replaced with imaging buffer containing 10 mM HEPES, pH 7.4, 25 mM glucose, 2 mM glutamax, 1.5 mM sodium pyruvate, 140 mM NaCl, 2.5 mM $CaCl_2$, 5% Annexin V Alexa Fluor 594 conjugate and 5 nM Sytox red. Data were acquired using a GE InCell analyzer 2500 HS microscope at ZMB, UZH as a time series at ×10 magnification in the green (Venus, transfection control), red (Annexin V, exposure of PS on the cell surface) and far red (Sytox, cell death) channels with images acquired every 10 s. Data were analysed in Fiji[47]. Cells displaying a Venus signal were selected as regions of interest and used for quantifying the fluorescence intensity in the Annexin V channel for all frames. Dead cells were excluded. The fluorescence intensity was quantified over multiple cells and normalized by the number of cells. The normalized values at 500 s after start of the recording were plotted.

### Dynamic force spectroscopy

MLCT cantilevers (Bruker AFM Probes) were functionalized through a gas-phase protocol. For initial cleaning, cantilevers were immersed in 2 ml CHCl$_3$ in a PTFE vessel for 5 min. The procedure was repeated

three times. After immersion, cantilevers were dried under a nitrogen stream. For functionalization, each cantilever was placed on a piece of Parafilm inside a polystyrene Petri dish, which was placed inside a desiccator along with 3× 30 µl of (3-aminopropyl)-triethoxysilane (APTES) and 3× 10 µl triethylamine in separate polystyrene caps. The cantilevers were incubated in the sealed desiccator for 2 h, flushed with argon and sealed for an additional 2 days to cure the amino functionalization. For functionalization of cantilevers with an aldehyde linker and TTYH2, 3.3 mg aldehyde-Ph-PEG24-NHS linker (BroadPharm) was dissolved in 500 µl DMSO in a PTFE vessel. Next, 30 µl triethylamine was added and mixed. The cantilevers were immersed in this solution for 2 h and subsequently cleaned in 2 ml CHCl₃, similar to the initial cleaning step. Cantilevers were then placed on a piece of Parafilm in a polystyrene Petri dish. Next, 100 µl TTYH2 (at a concentration of 3.3 µM) in SEC buffer and 2 µl of 1 M sodium cyanoborohydride stock solution (prepared by dissolving 13 mg NaCNBH₃ in 20 µl of 100 mM NaOH and 180 µl H₂O) was applied to each cantilever and incubated for 2 h followed by the addition of 5 µl of 1 M ethanolamine and a further 15 min of incubation. A final cleaning step involved immersing the cantilevers in SEC buffer, similar to the initial cleaning step. Cantilevers were stored in SEC buffer at 4 °C until use in measurements. The preparation of APOE surfaces followed a similar protocol to the preparation of the cantilevers, with specific modifications. As a linker, 1 mg acetal-PEG-NHS linker (Creative PEGWorks) in 0.5 ml DMSO was used. Following linker incubation, glass slides were immersed in 1% citric acid for 10 min and afterwards cleaned in H₂O, similar to the initial cleaning step. After the citric acid wash, low-height measurement chambers were mounted. All subsequent steps were performed in these chambers. For immobilization, 2 µM APOE solution in SEC buffer was applied followed by a final cleaning step involving three times exchange of the buffer solution as described for the initial cleaning step. Slides with mounted chambers were filled with SEC buffer and stored in sealed Petri dishes at 4 °C until used in measurements. An equivalent protocol was used for immobilization of Sb2.

## Dynamic force spectroscopy measurement protocol

Measurements were conducted in a sound-isolated and vibration-isolated chamber at room temperature using a JPK NanoWizard 4 atomic force microscope (Bruker) and JPK-SPM software (v.6.4.22). TTYH2–PEG-functionalized cantilevers were calibrated for spring constant in contact mode and thermal noise measurements[51–53]. Experiments were carried out in SEC buffer at neutral pH and at acidic pH in buffer containing 10 mM MES pH 5.5, 200 mM NaCl and 50 µM GDN using 3 different cantilevers and 3 APOE surfaces across 8 pulling velocities (0.1, 0.2, 0.5, 1, 2, 5, 10 and 20 µm s⁻¹). For each velocity, 1,000 force-displacement measurements were performed on a 10 × 10 grid (1 × 1 µm), with 10 measurements per grid point. Cantilever-bending corrections and analyses were conducted using JPK-Data processing software (DP-v.6.4). Further analysis, including Bell–Evans fitting for $k_{off}$ and $X_\beta$ values, was performed using in-house software developed in Python. During the approach and subsequent retraction of the cantilever, stochastic binding events were observed for both immobilized APOE (at pH 7.4 and 5.5) and Sb1 (at pH 7.4), which allowed the determination of the unbinding force $F_u$ (Fig. 2e). $F_u$ is directly correlated with the dissociation kinetics of the complex under an applied force and therefore depends on the loading rate (that is, the rate of force increase before unbinding)[27–29]. By varying the retraction speed while keeping the contact time constant, statistical analyses of $F_u$ as a function of the loading rate (defined as the product of retraction velocity and the effective spring constant of the cantilever, Fig. 2f) were performed. This resulted in a linear dependence of $F_u$ on the logarithm of the loading rate, which can be interpreted as a linear decrease in the free energy for dissociation, consistent with expectations for a single sharp energy barrier along the dissociation path[28] (Fig. 2f). Thus, the separation of the energy barrier from the

equilibrium position $X_\beta$ and $k_{off}$ were determined[30–32]. For competition experiments, 1,000 force displacement measurements were performed in the presence of 20–50 µM soluble Sb2 in the measurement buffer, and the reduction in binding events was analysed.

## Cryo-EM grid preparation and data collection

For the preparation of cryo-EM samples of TTYH2 in complex with sybodies, TTYH2 was concentrated to 2 mg ml⁻¹ and the sybodies were added at a 1:6 molar ratio of TTYH2 dimer to sybody and the samples were incubated for 30 min on ice before application on grids. For the preparation of a cryo-EM sample of APOE labelled with nanogold in complex with TTYH2, TTYH3 or the TTYH2(G165P/D166E/Q169R/F173R) mutant, the TTYH–APOE complexes were subjected to SEC and concentrated to 0.4 mg ml⁻¹. For the preparation of cryo-EM samples from the expression of both TTYH2 and APOE, the complex was concentrated to 0.7–1.5 mg ml⁻¹. For the sample of TTYH2 in complex with unlipidated APOE, the proteins were purified separately. TTYH2 was concentrated to 1.4 mg ml⁻¹ and mixed with APOE at 1:2.3 molar ratio. The sample was incubated for 30 min on ice before application on grids. The sample of TTYH3 containing delipidated APOE was prepared equivalently. For TTYH2 in complex with lipidated APOE, TTYH2 was concentrated to 1 mg ml⁻¹ and mixed with lipidated APOE at a 1:2.5 molar ratio. The sample was applied on grids immediately after mixing. For TTYH2 in cell-derived vesicles, the sample was concentrated to $A_{280}$ = 2.7. TTYH2 vesicles prepared for interaction studies were concentrated to $A_{280}$ = 9.1 and mixed with lipidated APOE immediately before application on grids. The final concentration of APOE was 45 µM and the vesicles were diluted twice by the addition of APOE.

Holey carbon grids Au 200 mesh R1.2/1.3 (Quantifoil) were used for all samples except the cell-derived vesicles, which were frozen on Au 300 mesh R1.2/1.3 (Quantifoil). Grids were freshly glow-discharged for 30 s. A volume of 2.5 µl sample was applied per grid. The blotting times varied between 2 and 4 s. Grids were plunge-frozen in liquid ethane–propane mix using Vitrobot Mark IV (Thermo Fisher Scientific) set to 4 °C and 100% humidity. After vitrification, grids were stored in liquid nitrogen. For samples with cell-derived vesicles, the Vitrobot was set to 20 °C and 100% humidity. A volume of 3.5 µl vesicles was applied and incubated on the grid for 1.5–2 min before the liquid was manually removed with filter paper. Subsequently, another 3.5 µl aliquot of vesicles was applied and incubated for 20 s, with excess liquid removed by blotting for 3 s using a Vitrobot before grids were plunge-frozen and stored in liquid nitrogen.

The grids were imaged on a Titan Krios G3i (Thermo Fisher Scientific) with a 100 µm objective aperture at the ZMB of UZH. All data were acquired using a post-column energy filter (Gatan) with a 20 eV slit and a K3 direct electron detector (Gatan) in super-resolution mode. All micrographs were recorded with a defocus range from −1 to −2.4 µm using EPU 2.9 + AFIS faster acquisition (Thermo Fisher Scientific) at a nominal magnification of ×130,000 corresponding to a pixel size of 0.651 Å pixel⁻¹ (0.3255 Å pixel⁻¹ in super-resolution) with a total exposure time of 1.26 s (47 individual frames). The total electron dose on the specimen level varied between 60 and 69 e⁻ Å⁻² for different datasets.

## Cryo-EM data processing

All cryo-EM datasets were processed in cryoSPARC[54], except for the datasets of detergent-purified TTYH2 in complex with lipidated APOE, which were processed in Relion[55]. A box size of 440 pixels was used for processing throughout unless specified otherwise. Datasets of TTYH2 expressed with APOE and TTYH2 in complex with delipidated APOE, as well as TTYH3 supplemented with delipidated APOE, were processed following a similar scheme. The micrographs were motion and CTF corrected and the particles were picked using the Template picker with the TTYH2 map obtained in our previous study[1] for template generation. Initially picked particles were 2D classified and particles with TTYH2 features were used to generate an ab initio volume.

The map was improved by homogenous and nonuniform refinement to high resolution (2.8–3.5 Å). At this stage, only density of TTYH2 but not of APOE was visible. To isolate the particles that contained the TTYH2–APOE complex, we used 3D variability analysis with a loose mask covering only the top of TTYH2, where the interaction with APOE was expected. This enabled the isolation of particle subsets that contained the TTYH2–APOE complex, which were used for training of the Topaz neural network for more precise particle picking. Particles picked using Topaz[56] were analysed using the same pipeline starting from 2D classification. After a final 3D variability analysis, particles that contained the TTYH2–APOE complex were used to create an ab initio map, which was refined by homogenous and nonuniform refinement.

Datasets of TTYH2 in complex with sybodies were processed following the same strategy as described above. For this dataset, the sybody density was already observed in early-stage ab initio maps. However, 3D variability analysis was still necessary for filtering out misaligned particles and to improve the quality of the map.

The datasets with TTYH2 purified in detergent in complex with lipidated APOE were processed in Relion[57]. Motion-corrected and CTF-corrected micrographs were used for particle picking using the TTYH2 map as a 3D reference. Particles were extracted with a box size of 440 pixels with 4× binning, which turned out to be optimal for this case (Supplementary Fig. 9). After several rounds of 2D classification, a subset of particles with visible APOE density was used to generate an ab initio map. This map was refined and used as a reference for a 3D classification with 10 classes and a regularization parameter of $T = 20$ on a large subset of particles. Particles from promising classes were selected for further 3D classification with similar parameters. At each step, the maps for 3D references were generated by ab initio reconstruction and refinement. Final particle subsets were used for generating maps by ab initio reconstruction and refinement and the final maps were sharpened.

For the dataset of TTYH2, TTYH3 and the TTYH2(G165P/D166E/Q169R/F173R) mutant with nanogold-labelled APOE, we collected 707, 3,000 and 2,486 micrographs, respectively, which produced particle sets of sufficient size for obtaining low-resolution reconstructions. Motion-corrected and CTF-corrected micrographs were used for template particle picking using the TTYH2 or TTYH3 map for the generation of templates. Particles picked on carbon areas were excluded. The particles were 2D-classified to remove junk and the final set of particles was refined by homogeneous refinement using the TTYH2 or TTYH3 map as input volume.

For processing of the datasets with TTYH2-containing cell-derived vesicles, the 're-center 2D classes' option was switched off in all 2D classification jobs. Motion-corrected and CTF-corrected micrographs were used for manual particle picking. About 500 manually selected particles with side and top views of TTYH2 were used for initial Topaz-based particle picking. These particles were 2D classified, and the best classes were used for a second round of Topaz-based picking. The iterative process was repeated until a homogenous subset of particles was obtained with clear TTYH2 features visible on 2D class images. This subset was used for ab initio reconstruction. Initially, the particles were extracted in a comparatively small box size of 280 pixels, which was necessary to assist the alignment process. The 'center structures in real space' option was switched off in all ab initio jobs. The initial map was refined using the homogenous refinement option. The refined particles were re-extracted in a box size of 440 pixels and refined again in the new box size. The obtained map was locally refined focusing on the protein and excluding the membrane. The final map was sharpened.

TTYH2 in cell-derived vesicles in complex with lipidated APOE was processed in a similar manner as for the dataset containing TTYH2 vesicles alone. Particle picking was initially carried out manually and then interactively with Topaz. Particles were first extracted in a box size of 280 pixels. Particles from the best-looking 2D classes with TTYH2

features were used in an ab initio job with the 'center structures in real space' option switched on. When this option was switched off, as described for the previous dataset, the APOE density vanished as a result of averaging. The ab initio map was refined and the refined particles were used in a 3D classification job with the protein density masked. The 3D classification helped to separate the particle subsets with the APOE density bound to TTYH2. The new set of particles containing APOE was refined again and the refined particles were re-extracted in a box size of 440 pixels. The re-extracted particles were then used as input for homogenous refinement and the refined volume was sharpened.

## Model building and refinement

The cryo-EM structure of TTYH2 in detergent (Protein Data Bank (PDB) accession 7P54) was used to build the TTYH2 models in complex with Sb1 or lipids. TTYH2 was initially placed into the cryo-EM density by rigid body fitting in Chimera[58,59]. The structure of the GFP-binding nanobody (PDB 3K1K) with variable regions removed was used as an initial scaffold for the modelling of Sb1. The nanobody was placed into the density in Chimera and the CDR loops were manually edited to match the Sb1 sequence in Coot. The structure of the complex was refined in Phenix[60]. For the high-resolution structure of TTYH3, the cryo-EM structure of TTYH3 in detergent (PDB 7P5C) was fitted into the cryo-EM density in Chimera and the structure was refined in Phenix. For the high-resolution structure of TTYH2 with bound lipids, TTYH2 was fitted into the cryo-EM density in Chimera and lipids were manually placed in Coot[61]. The structure was refined in Phenix. For all cryo-EM maps of TTYH2 in complex with APOE, the TTYH2 model was fitted into the density in Chimera. Figures containing molecular structures and densities were prepared using DINO (http://www.dino3d.org) and ChimeraX[59].

## Reporting summary

Further information on research design is available in the Nature Portfolio Reporting Summary linked to this article.

## Data availability

The 3D cryo-EM density maps have been deposited into the Electron Microscopy Data Bank (EMDB) with the following accession numbers: EMD-51106 for TTYH2 in complex with Sb1; EMD-53290 for TTYH2 in complex with Sb2; EMD-53291 for TTYH2 in complex with Sb1 in cell-derived vesicles; EMD-53289 for TTYH2 in complex with delipidated APOE; EMD-53249 and EMD-53263 for high and low resolution classes of TTYH3 after incubation with delipidated APOE; EMD-53301 and EMD-53297 for TTYH2 complexes with expressed APOE from dataset 1; EMD-53303 and EMD-53302 for TTYH2 complexes with expressed APOE from dataset 2; EMD-53304 for TTYH2 complex with expressed APOE from dataset 3; EMD-51108 for TTYH2 in complex with lipids; EMD-53293 for TTYH2 in complex with the C-terminal APOE fragment; EMD-53251 and EMD-53269 for TTYH2 complexes with lipidated APOE from dataset 1; EMD-53280 for TTYH2 in complex with lipidated APOE from dataset 2; EMD-53272, EMD-53273 and EMD-53271 for APOE-containing lipoprotein discs from dataset 1; and EMD-53292 for TTYH2 in complex with lipidated APOE in cell-derived vesicles. Relevant coordinates were deposited into the Protein Data Bank (PDB) with the following accession numbers: 9G6X for TTYH2 in complex with Sb1; 9G71 for TTYH2 in complex with lipids; and 9QNR for TTYH3. The cryo-EM structure of TTYH2 (PDB: 7P54) was used for model building of TTYH2 in complex with Sb1 or lipids and was fitted into cryo-EM maps of TTYH2 in complex with APOE. The structures of the GFP-binding nanobody (PDB: 3K1K) and of TTYH3 (PDB: 7P5C) were used as initial models for Sb1 and TTYH3, respectively. The structure of the N-terminal domain of APOE (PDB: 1B68) and the structure of TTYH2 (PDB: 7P54) were used to create illustrations in BioRender. Source data are provided with this paper.

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

**Acknowledgements** This research was supported by grants from the Swiss National Science Foundation (number 310030_215236 to R.D.) and the Austrian Science Fund (projects PIN2697624 and P33481-B to B.P.). A.S. was supported by a Candoc grant of the University of Zurich (project number K-41106-04). Cryo-EM and confocal light microscopy data were collected at the Center for Microscopy and Image Analysis (ZMB) of the University of Zurich. Mass spectrometry data were collected at the Functional Genomics Center Zurich. We thank M. Seeger for providing the sybody libraries and all members of the Dutzler Laboratory for their help at various stages of the project. The diagrams in Figs. 2d,e, 5a and 6 and Extended Data Figs. 2a, 3d and 9e,i,k were created using BioRender (https://www.biorender.com).

**Author contributions** A.S. expressed and purified proteins, prepared mass spectrometry samples, performed sybody selections, confocal light microscopy and cryo-EM data collection and analyses, cell-fractionation experiments and lipid-transport assays. A.P. expressed and purified proteins, performed confocal light microscopy and cryo-EM data analyses and performed scramblase assays. A.K., F.W. and B.P. carried out atomic force microscopy experiments. A.S., A.P., A.K., F.W., B.P. and R.D. jointly planned experiments, analysed the data and wrote the manuscript.

**Funding** Open access funding provided by University of Zurich.

**Competing interests** The authors declare no competing interests.

**Additional information**
**Correspondence and requests for materials** should be addressed to Raimund Dutzler.

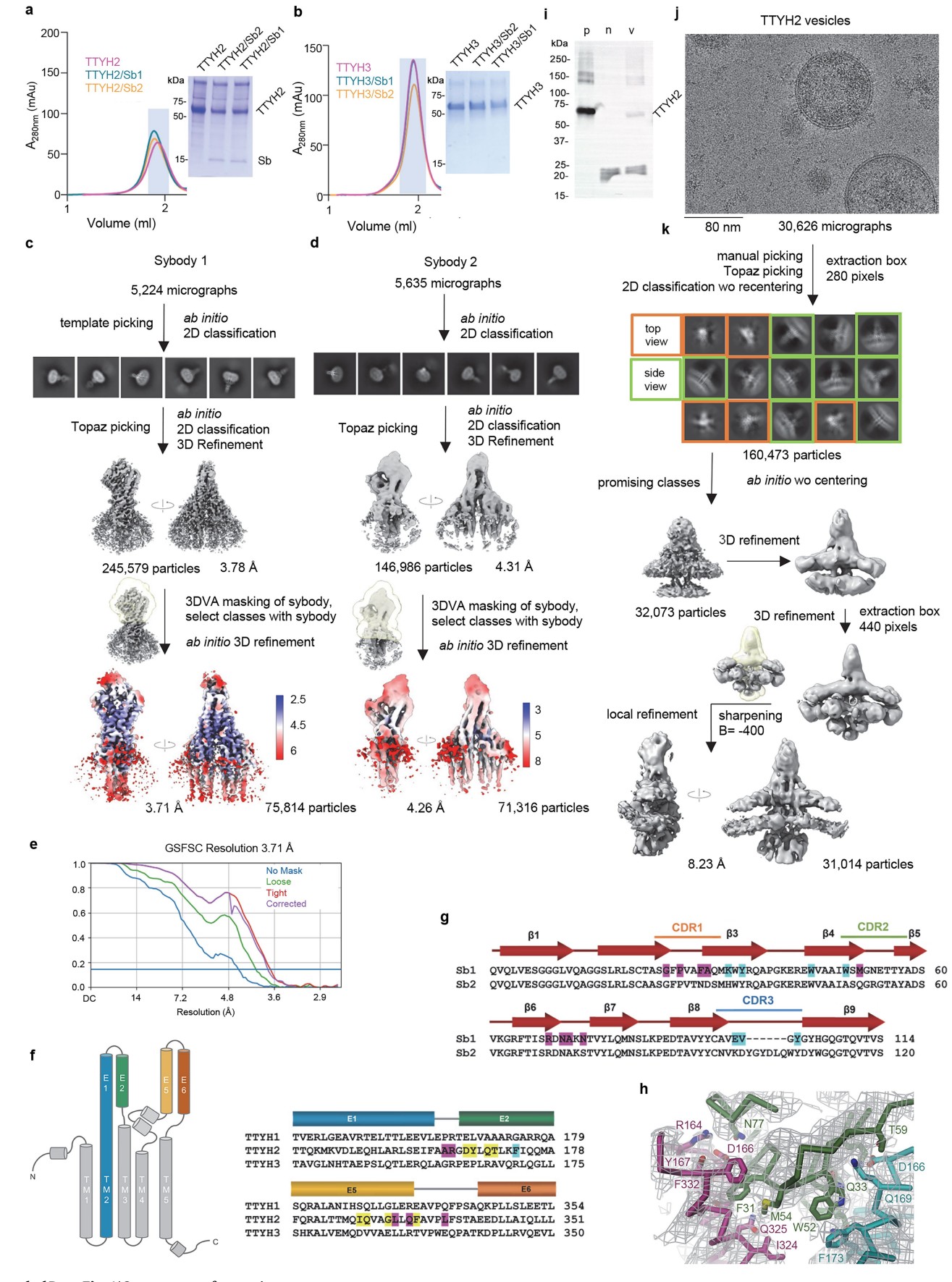

**Extended Data Fig. 1** | See next page for caption.

**Extended Data Fig. 1 | TTYH2-sybody interactions. a, b**, Size exclusion chromatography (SEC) traces of samples containing TTYH2 (**a**) or TTYH3 (**b**) and either Sb1 or Sb2. The peak corresponds to TTYH fractions. Insets show SDS-PAGE gels of peak fractions revealing complex formation in case of TTYH2 but not TTYH3. TTYH2 not containing sybody is shown as control. The measurements were carried out once per condition. **c, d**, Data processing strategy for TTYH2 in complex with Sb1 (**c**) and Sb2 (**d**). Particles containing sybodies in complex with TTYH2 were first isolated via 2D classifications and further sorted using 3D variability analysis in cryoSPARC to obtain high resolution reconstructions. **e**, Fourier shell correlation (FSC) plot of the final refined cryo-EM density map of the TTYH2/Sb1 complex. **f**, Topology of the TTYH subunit (left) and sequence alignment of the sybody-binding region of the three human paralogs (with secondary structure elements displayed on top). Residues buried in the interaction interface are highlighted (cyan and magenta indicate interacting residues in one of the two subunits, yellow indicates residues involved in sybody interactions in both subunits).

**g**, Sequence alignment of the TTYH2 binders Sb1 and Sb2 (with secondary structure elements displayed on top). Residues of Sb1 buried in the interface with TTYH2 subunits are highlighted (colored as in **f**). **h**, Cryo-EM density in the interaction region of the TTYH2/Sb2 complex superimposed on a Cα-trace of the proteins with sidechains of interacting residues shown as sticks and labeled. TTYH2 subunits are colored in cyan and magenta, Sb1 in green. **i**, Western blot of cell-derived vesicles overexpressing TTYH2 that were isolated by affinity purification with immobilized Sb1. TTYH2 is detected in purified protein (p) and purified vesicles (v) but not in the negative control only containing Sb1 (n). Western blot analysis was carried out once for this preparation. **j, k** Representative micrograph (total micrographs=30,626) from cell-derived vesicles containing TTYH2 (**j**), and data processing workflow (**k**). Initially manually picked particles were used for training a Topaz model. Topaz-picked particles were first extracted in a small box size of 280 pixels to aid the alignment of TTYH2 in vesicle membranes and the final particle set was re-extracted in a bigger box size of 440 pixels to obtain a reconstruction at 8.23 Å.

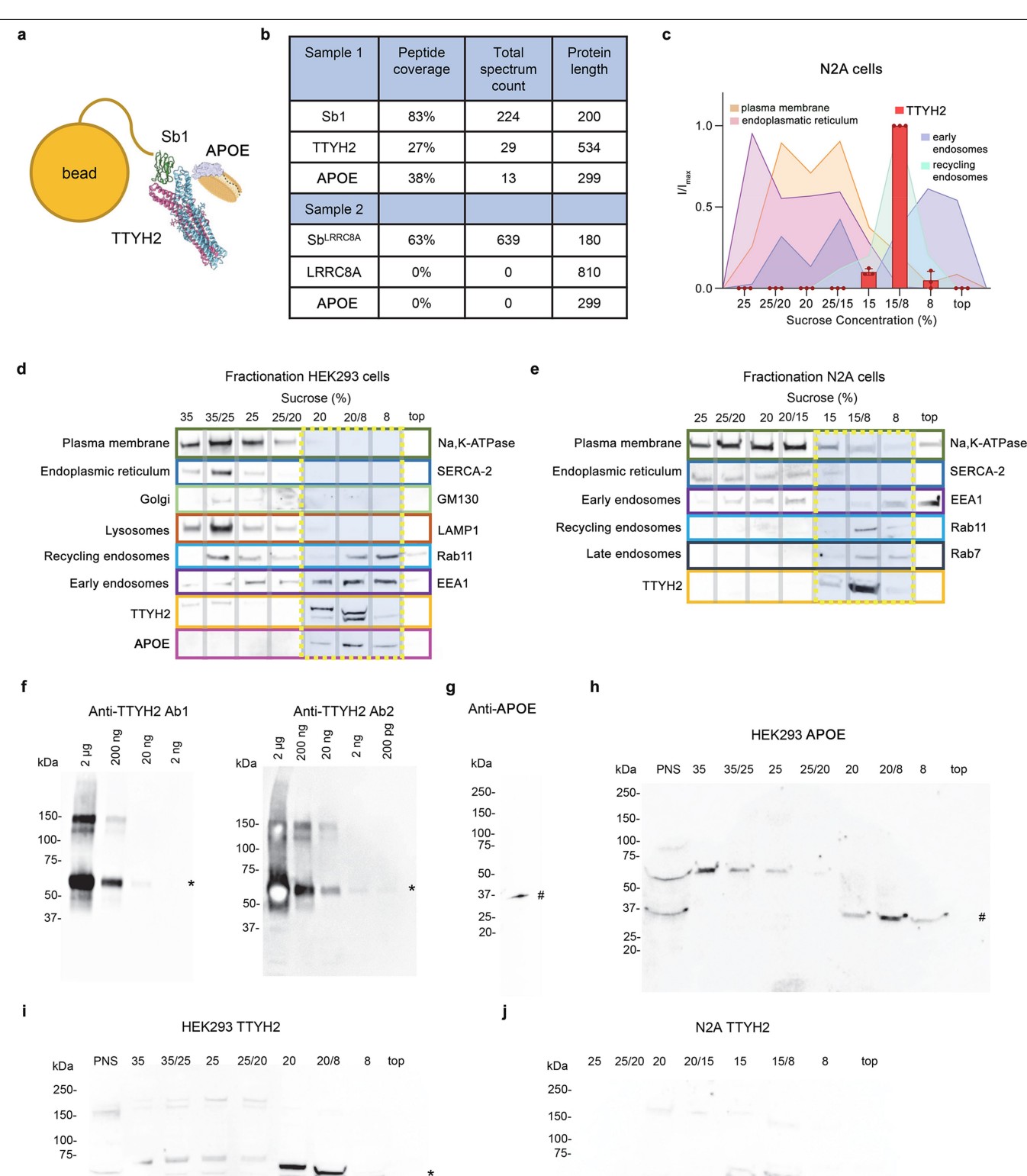

**Extended Data Fig. 2** | See next page for caption.

**Extended Data Fig. 2 | Proteomics and subcellular fractionation.**
**a**, Scheme of the pull-down experiment from HEK293 extracts with Sb1 immobilized on streptactin resin. The eluted sample was analyzed by LC/MS for presence of TTYH2 and its interaction partners. **b**, Table summarizing the results of the pulldown LC/MS analysis with TTYH2 and APOE identified as major co-purifying components. A sybody binding LRRC8A[62] was used in a negative control experiment where neither APOE nor TTYH2 were detected. **c**, Subcellular fractionation of the post nuclear supernatant of mouse neuroblastoma cells (N2A) on a sucrose gradient. TTYH2 (red) and specific marker proteins defining the cellular origin of the fractions (illustrated by shaded background) were identified by western blot. Plotted are averaged intensities from n = 3 independent fractionation experiments. that are normalized to the highest value. Errors are s.e.m. **d**, **e**, Sections of representative western blots from a fractionation of post-nuclear supernatant from HEK293 (**d**) or N2A cells (**e**) on a sucrose gradient.

TTYH2 and APOE are co-localizing in the same fractions overlapping with endosomal markers in HEK293 cells (**d**). A similar colocalization of TTYH2 with endosomal markers is found in N2A cells (**e**). **f**, western blots showing recognition of purified TTYH2 by two anti-TTYH2 antibodies used in western blots (Ab1) and immunocytochemistry (Ab2). Each experiment was performed once. **g**, Western blot showing recognition of purified APOE by an antibody used in western blots and immunocytochemistry, the experiment was carried out once. **h-i** Uncropped western blots of APOE (**h**) and TTYH2 (**i**) in subcellular fractions of HEK293 cells shown in **d**. **j**, Uncropped Western blot of TTYH2 in subcellular fractions of N2A cells shown in **e**. **f-j**, TTYH2 location is indicated by '*', APOE location by '#'. The western blots shown in **h-j** were repeated 3 times in separate fractionation experiments. The diagram in **a** was created using BioRender (https://www.biorender.com).

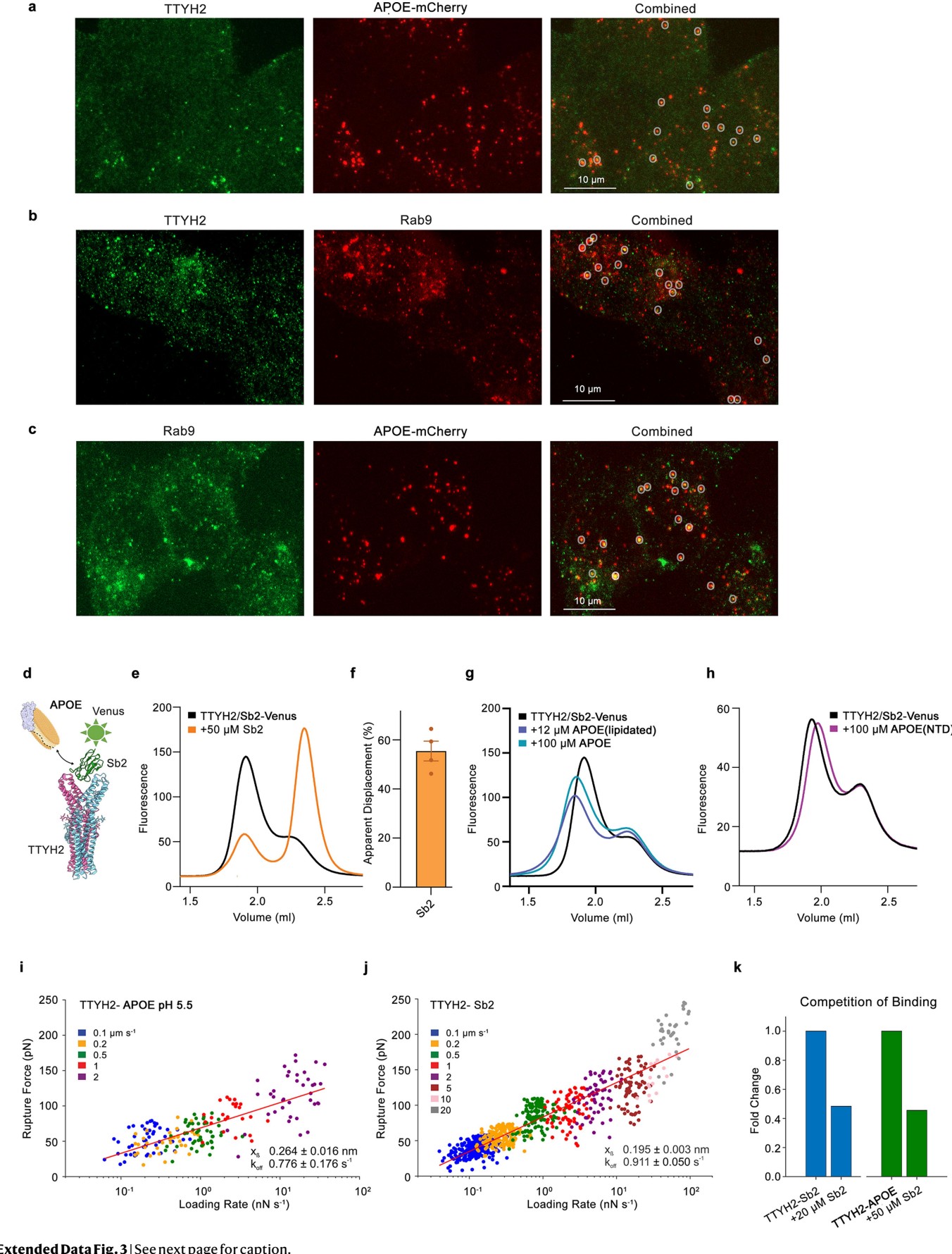

**Extended Data Fig. 3** | See next page for caption.

**Extended Data Fig. 3 | Confocal microscopy and sybody displacement to probe the colocalization and interaction of APOE and TTYH2.**
**a-c**, Confocal fluorescent microscopy images of HEK293 cells showing the pairwise colocalization of TTYH2, APOE and the endosomal marker Rab9. **a**, Image showing the intracellular localization of TTYH2 (green, stained with a polyclonal antibody targeting the protein) and its colocalization with endocytosed APOE-lipoprotein particles that were reconstituted from a purified APOE-mCherry fusion protein. **b**, Image showing the colocalization of TTYH2 (green) with Rab9 (red). **c**, Image showing the colocalization of Rab9 (green) with endocytosed APOE-mCherry lipoprotein particles. Shown are images of the individual channels and their combination (right). Locations with overlap of both fluorescent markers are indicated by circles. The samples were imaged in multiple locations producing similar results and the displayed images were chosen as representative (**a-c**). **d**, Schematic representation of the sybody displacement assay where APOE is incubated with a pre-formed complex of TTYH2 with Sb2-Venus followed by the separation on a Superose 6 5/150 GL column. **e**, Representative size-exclusion profiles of TTYH2 in complex with Sb2-Venus and the displacement of Sb2-Venus by non-fluorescent Sb2. **f**, Competition of Venus-labeled Sb2 bound to TTYH2 by non-fluorescent Sb2 quantified by the altered fluorescence ratio of the first peak corresponding to the complex of TTYH2 with Sb2-Venus and the second peak corresponding to isolated Sb2-Venus. Shown are averages of n = 4 independent experiments (circles), error is s.e.m. **g**, **h**, Representative size-exclusion profiles of the TTYH2 in complex with Sb2-Venus and the displacement of Sb2-Venus by indicated concentrations of APOE-lipoproteins (APOE(lipidated)) and delipidated APOE (APOE) (**g**), and the N-terminal domain of APOE (APOE(NTD)) (**h**). APOE and APOE(lipidated), but not APOE(NTD), displace Sb2-Venus to some extent. **e**, **g**, h Chromatograms show Venus fluorescence. **i**, **j** Analysis of the unbinding force $F_u$ as function of the loading rate measured at different retraction velocities of the tip containing immobilized TTYH2 and, **i**, delipidated APOE at pH 5.5 or, **j**, Sb2 immobilized on a solid support. Shown are individual measurements. The line displays a fit to a Bell-Evans model with the fitted values of $x_\beta$ and $k_{off}$ displayed. Errors are given as standard deviations. **k**, Competition of soluble Sb2 with TTYH2-Sb2 and TTYH2-APOE interactions, respectively. Shown is the reduction in binding events in force spectroscopy experiments upon addition of 20–50 μM Sb2 to the measurement buffer. The diagram in **d** was created using BioRender (https://www.biorender.com).

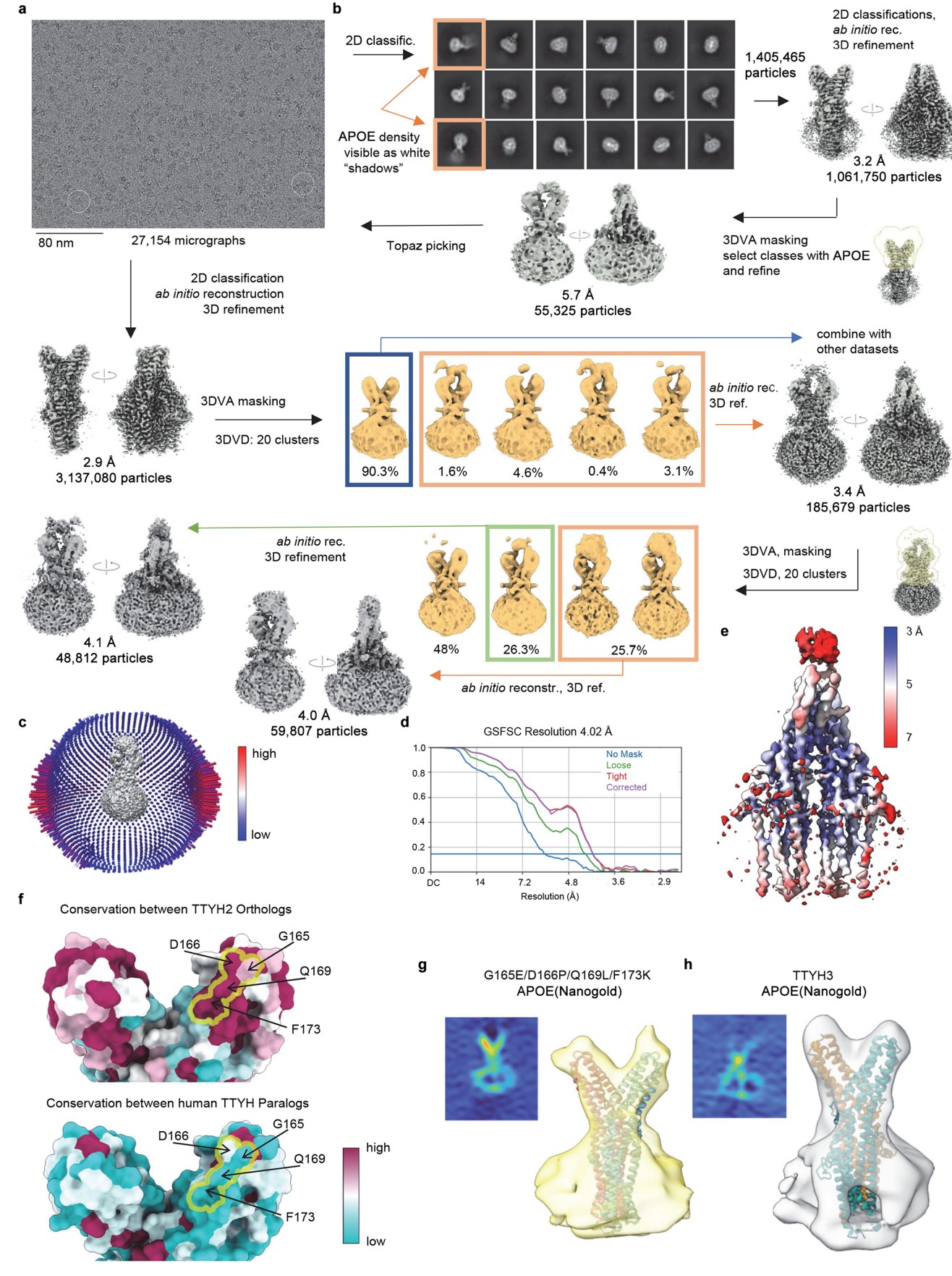

**a**

80 nm

27,154 micrographs

2D classification
*ab initio* reconstruction
3D refinement

2.9 Å
3,137,080 particles

**b**

2D classific.

APOE density
visible as white
"shadows"

2D classifications,
*ab initio* rec.
3D refinement

1,405,465
particles

3.2 Å
1,061,750 particles

3DVA masking
select classes with APOE
and refine

Topaz picking

5.7 Å
55,325 particles

3DVA masking

3DVD: 20 clusters

combine with
other datasets

90.3%    1.6%    4.6%    0.4%    3.1%

*ab initio* rec.
3D ref.

3.4 Å
185,679 particles

3DVA, masking
3DVD, 20 clusters

*ab initio* rec.
3D refinement

4.1 Å
48,812 particles

48%    26.3%    25.7%

*ab initio* reconstr., 3D ref.

4.0 Å
59,807 particles

**c**

high

low

**d**

GSFSC Resolution 4.02 Å

No Mask
Loose
Tight
Corrected

DC    14    7.2    4.8    3.6    2.9
Resolution (Å)

**e**

3 Å

**f**

Conservation between TTYH2 Orthologs

D166    G165

Q169

F173

Conservation between human TTYH Paralogs

D166    G165

Q169

F173

high

low

**g**

G165E/D166P/Q169L/F173K
APOE(Nanogold)

**h**

TTYH3
APOE(Nanogold)

**Extended Data Fig. 4** | See next page for caption.

**Extended Data Fig. 4 | Cryo-EM data processing of TTYH2 in detergent in complex with delipidated APOE and conservation of the APOE interaction interface. a**, Representative motion and CTF-corrected micrograph (total micrographs=27,154). **b**, Data processing workflow. Particles were picked using a previously obtained TTYH2 map for template generation. After several rounds of 2D classification, particles of APOE in complex with TTYH2 were isolated by 3D variability analysis in cryoSPARC and used for the training of a Topaz model for more accurate particle picking. Topaz-picked particles were sorted by 2D classifications and 3D variability analysis, which yielded two similar maps of the TTYH2-APOE complex at 4.1 and 4.0 Å. **c**, Angular distribution of particle orientations and, **d**, Fourier shell correlation (FSC) plot of the final refined cryo-EM density map of the TTYH2 in complex with APOE at 4.0 Å with pronounced density of APOE. **e**, Final 3D reconstruction of the TTYH2 in complex with APOE colored according to its local resolution. **f**, Sequence conservation of residues from an alignment of 26 mammalian TTYH2 orthologs (left) and the three human paralogs TTYH1-3 (right). Shown is the tip of the extracellular domain with the approximate binding epitope for APOE indicated by a yellow line and residues located in this epitope labeled. **g**, **h**, Low resolution 3D reconstructions of a sample containing, **g**, a TTYH2 quadruple mutant involving residues of the putative APOE binding site (G165P/D166E/Q169R/F173R) and, **h**, TTYH3 in presence of APOE containing a nanogold label. A slice through the density is shown left. In contrast to the data of TTYH2 prepared under equivalent conditions (Fig. 3a) no evidence for bound APOE is detected.

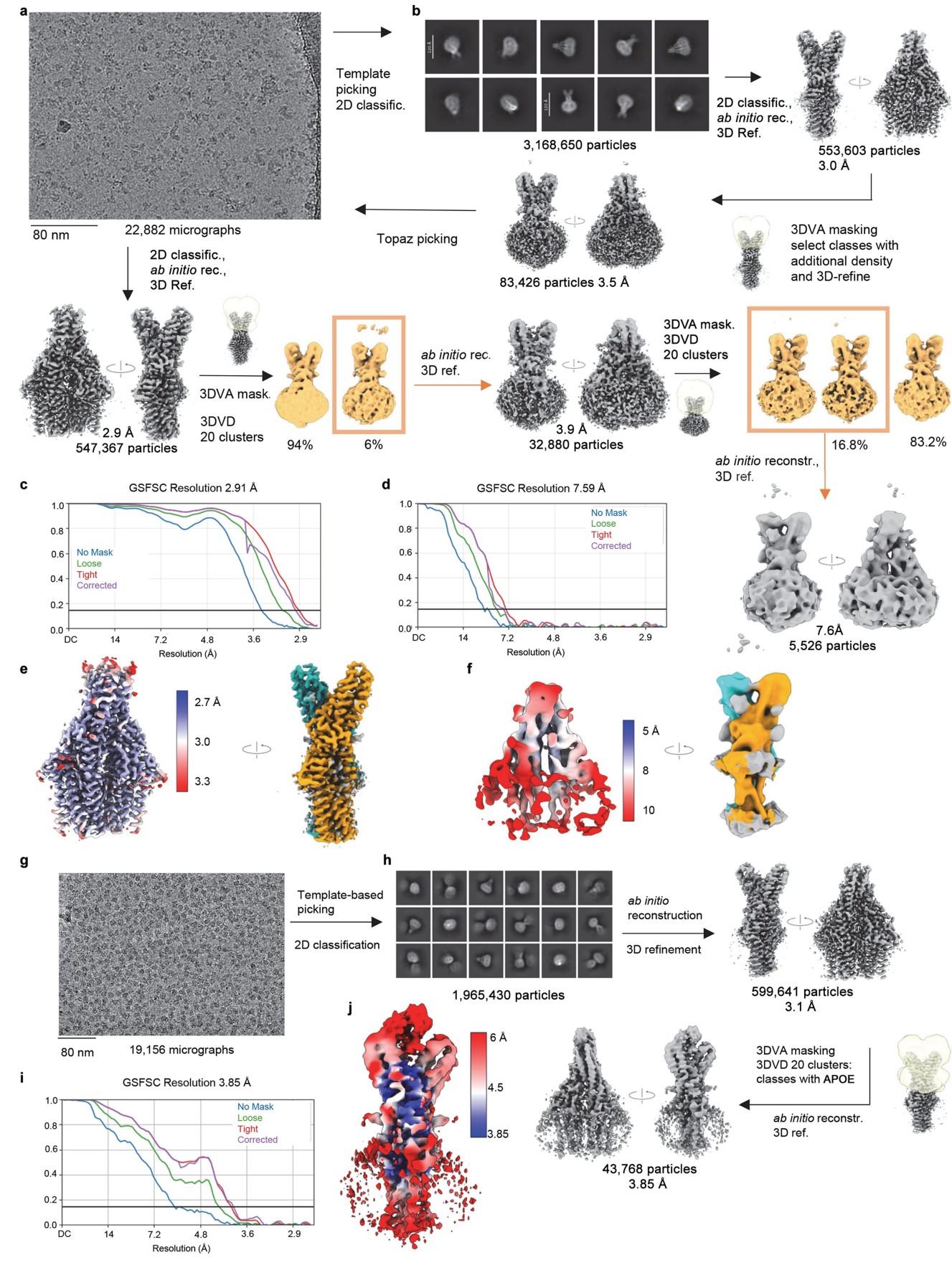

**Extended Data Fig. 5** | See next page for caption.

**Extended Data Fig. 5 | Cryo-EM data processing of TTYH3 in presence of delipidated APOE and TTYH2 in complex with the APOE C-terminal domain. a-f,** Structure of TTYH3 in presence of delipidated ApoE. **a,** Representative motion and CTF-corrected micrograph (total micrographs=22,882). **b,** Data processing workflow. The same protocol as for the reconstruction of the TTYH2 in complex with APOE was used. Particles were picked using a previously obtained TTYH3 map for template generation. After several rounds of 2D classification, particles of TTYH3 showing some additional density surrounding the extracellular domains were isolated by 3D variability analysis in cryoSPARC and used for the training of a Topaz model for more accurate particle picking. Topaz-picked particles were sorted by 2D classifications and *ab initio* reconstructions, which yielded a map of TTYH3 at 2.9 Å, which is very similar to the previously determined TTYH3 structure[1]. This particle set was further analyzed by 3D variability analysis aiming to separate particles containing any additional density, which yielded a low-resolution reconstruction of a small set of particles at 7.6 Å. This reconstruction did not display pronounced additional density in the equivalent region corresponding to the ApoE binding site of TTYH2. **c, d,** Fourier shell correlation (FSC) plots of the final refined cryo-EM density map of TTYH3 map at high resolution and of the low-resolution class of selected particles. **e, f,** Final 3D reconstructions of, **e,** TTYH3 at 2.9 Å and, **f,** the class at 7.6 Å colored according to their local resolution with 90° rotated view with subunits shown in unique colors. **g-j,** Cryo-EM structure of TTYH2 in complex with the C-terminal domain of APOE (APOE(CTD)). **g,** Representative motion and CTF-corrected micrograph of the TTYH2 in complex with APOE(CTD) (total micrographs=19,156). **h,** Data processing workflow. Particles were picked using a previously obtained TTYH2 map for template generation. Template-picked particles were initially sorted by 2D classifications. A homogenous particle set yielding a high-resolution TTYH2 reconstruction was further sorted using 3D variability analysis in cryoSPARC to isolate particles with APOE(CTD) bound on top of TTYH2. **i,** Fourier shell correlation (FSC) plot of the final refined cryo-EM density map of the TTYH2 in complex with APOE(CTD) at 3.8 Å with pronounced density of APOE(CTD). **j,** Final 3D reconstruction of the TTYH2 in complex with APOE(CTD) colored according to its local resolution.

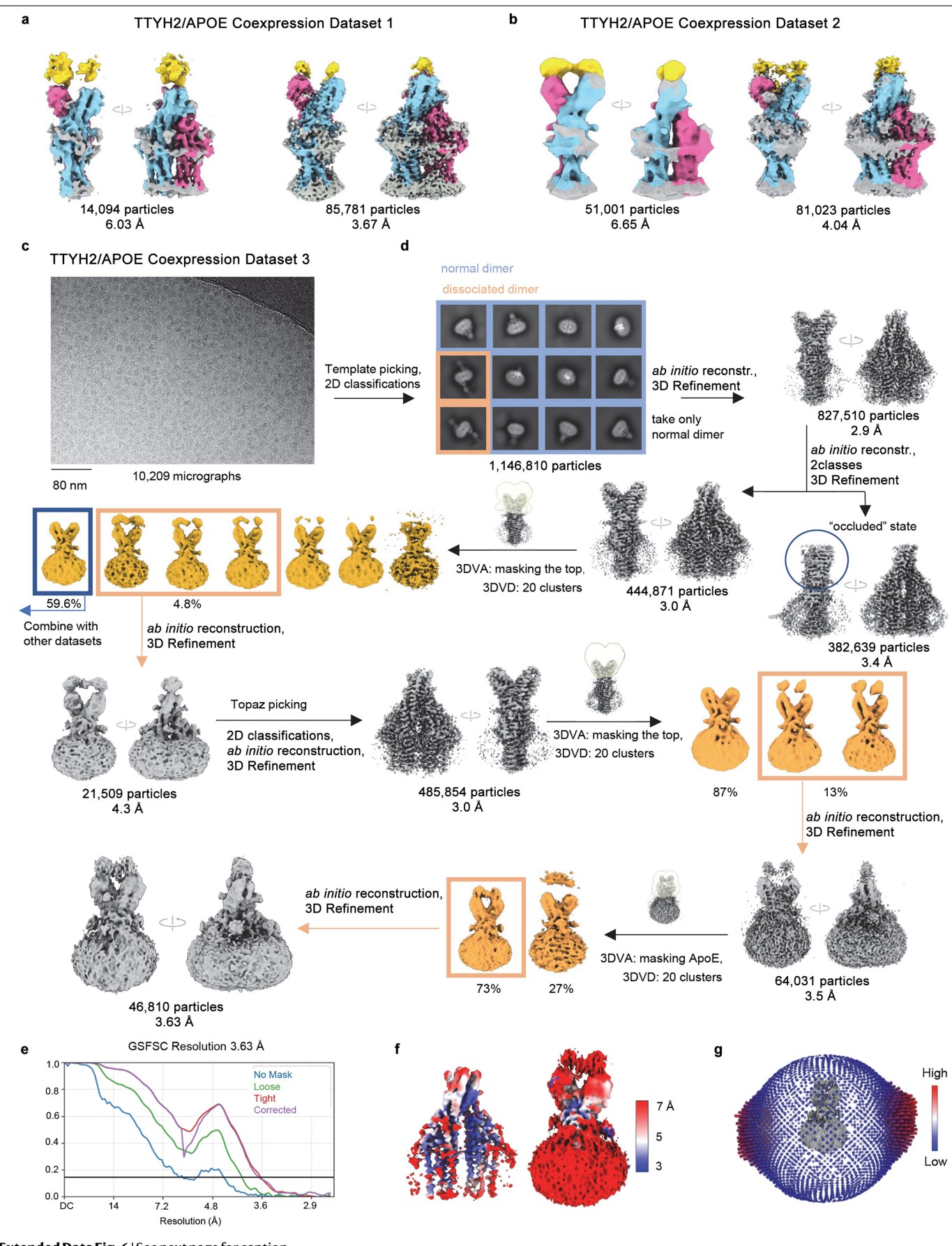

**a**  TTYH2/APOE Coexpression Dataset 1

14,094 particles
6.03 Å

85,781 particles
3.67 Å

**b**  TTYH2/APOE Coexpression Dataset 2

51,001 particles
6.65 Å

81,023 particles
4.04 Å

**c**  TTYH2/APOE Coexpression Dataset 3

80 nm

10,209 micrographs

**d**

normal dimer
dissociated dimer

Template picking,
2D classifications

1,146,810 particles

*ab initio* reconstr.,
3D Refinement

take only
normal dimer

827,510 particles
2.9 Å

*ab initio* reconstr.,
2classes
3D Refinement

"occluded" state

444,871 particles
3.0 Å

3DVA: masking the top,
3DVD: 20 clusters

382,639 particles
3.4 Å

59.6%

4.8%

Combine with
other datasets

*ab initio* reconstruction,
3D Refinement

21,509 particles
4.3 Å

Topaz picking
2D classifications,
*ab initio* reconstruction,
3D Refinement

485,854 particles
3.0 Å

3DVA: masking the top,
3DVD: 20 clusters

87%

13%

*ab initio* reconstruction,
3D Refinement

*ab initio* reconstruction,
3D Refinement

73%

27%

3DVA: masking ApoE,
3DVD: 20 clusters

64,031 particles
3.5 Å

46,810 particles
3.63 Å

**e**  GSFSC Resolution 3.63 Å

No Mask
Loose
Tight
Corrected

Resolution (Å)

**f**

7 Å
5
3

**g**

High

Low

**Extended Data Fig. 6** | See next page for caption.

**Extended Data Fig. 6 | Cryo-EM reconstructions from TTYH2 in complex with coexpressed APOE. a**, **b**, Final cryo-EM reconstructions from Dataset 1 (**a**) and Dataset 2 (**b**) collected from two independent samples obtained from the coexpression of TTYH2 and ApoE in HEK293 cells. Total number of micrographs for dataset 1 is 8,954 and 19,964 for dataset 2. Both samples yielded a high-resolution map of TTYH2 with traces of APOE density on top (right) and a second map at lower resolution displaying a more extended APOE density (left). **c-g**, Structural features of Dataset 3 of the TTYH2 in complex with coexpressed APOE (ApoE(CE)). **c**, Representative motion and CTF-corrected micrograph (total micrographs=10,209). **d**, Data processing workflow. Particles were picked using a previously obtained TTYH2 map for template generation. Template-picked particles were initially sorted by 2D classifications. A homogenous particle set yielding a high-resolution TTYH2 reconstruction was further sorted using 3D variability analysis in cryoSPARC to isolate particles of TTYH2 in complex with APOE(CE). These particles were used for the training of a Topaz model for more accurate particle picking. Topaz-picked particles were sorted by subsequent rounds of 2D classifications and 3D variability analysis, which yielded the final map of the TTYH2-APOE complex at 3.6 Å. **e**, Fourier shell correlation (FSC) plot of the final refined cryo-EM density map of the TTYH2 in complex with APOE(CE) at 3.6 Å with pronounced density of APOE(CE). **f**, Final 3D reconstruction of the TTYH2 in complex with APOE(CE) colored according to its local resolution at high and low contour. **g**, Angular distribution of particle orientations.

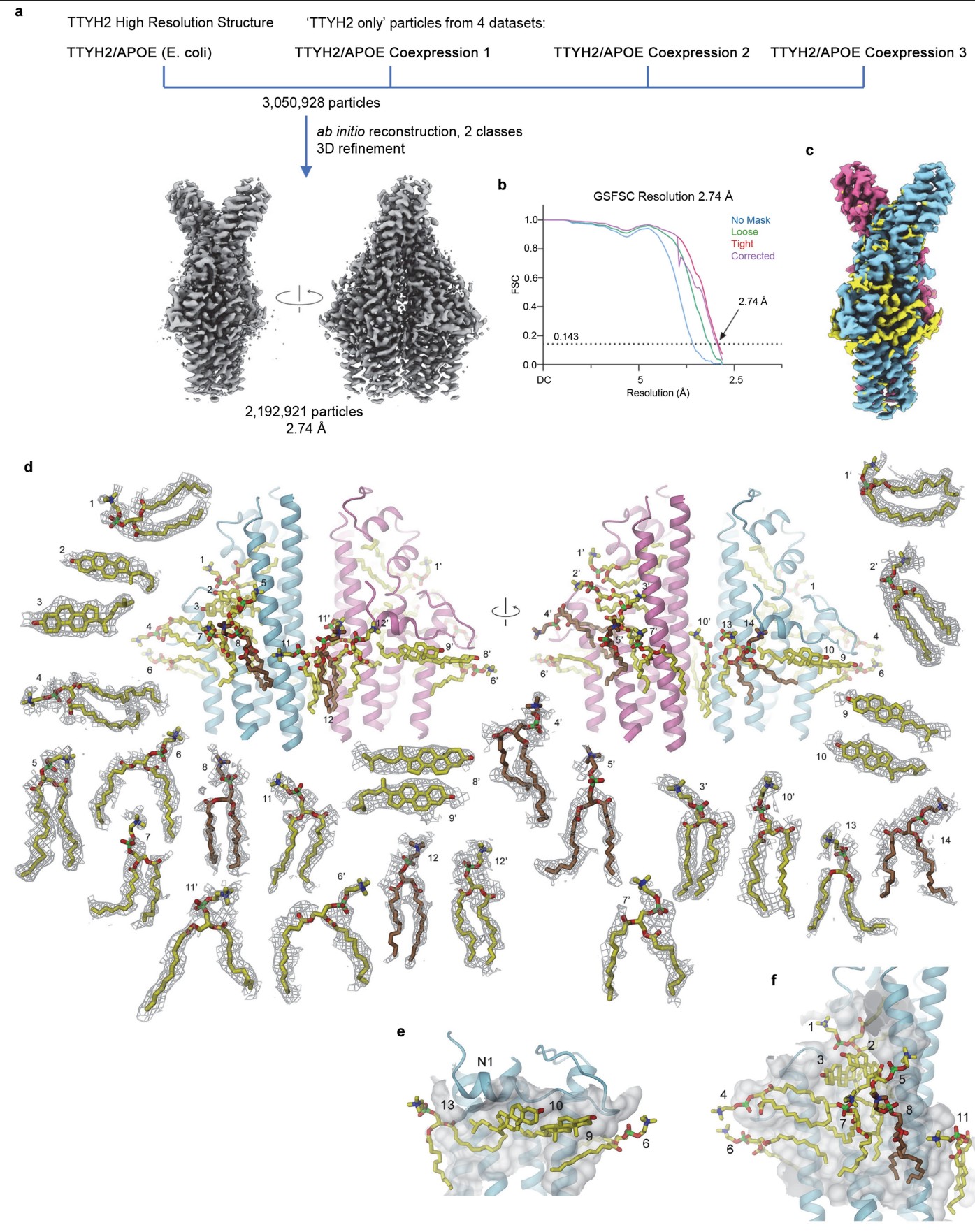

**a**

TTYH2 High Resolution Structure    'TTYH2 only' particles from 4 datasets:

TTYH2/APOE (E. coli)    TTYH2/APOE Coexpression 1    TTYH2/APOE Coexpression 2    TTYH2/APOE Coexpression 3

3,050,928 particles

*ab initio* reconstruction, 2 classes
3D refinement

2,192,921 particles
2.74 Å

**b**

GSFSC Resolution 2.74 Å

No Mask
Loose
Tight
Corrected

2.74 Å

0.143

FSC

DC    5    2.5
Resolution (Å)

**c**

**d**

**e**    N1

**f**

**Extended Data Fig. 7** | See next page for caption.

**Extended Data Fig. 7 | Lipid interaction in a high-resolution structure of TTYH2. a**, TTYH2 particles not containing density corresponding to APOE were selected from the dataset of TTYH2 in complex with delipidated APOE and the three datasets of TTYH2 in complex with coexpressed APOE. The resulting particle set was used for *ab initio* reconstruction and 3D refinement yielding a map at 2.7 Å resolution. **b**, Fourier shell correlation (FSC) plot of the final refined cryo-EM density map of the TTYH2 structure at 2.7 Å with pronounced density of bound lipids. **c**, Density of TTYH2 viewed along the long dimensions of the dimeric molecule (colored in cyan and magenta) with lipid density colored in yellow. **d**, Lipid interaction region in the TTYH2 dimer viewed from opposite sides. The protein is shown as ribbon (colored in cyan and magenta), refined positions of lipid and cholesterol molecules as sticks. 15 well defined lipids and 6 cholesterol molecules are colored in yellow, 5 putative lipids with weaker density in brown. Lipids are labeled corresponding to their interacting subunit and shown as blow-up with their respective cryo-EM density superimposed (map contour: 2′, 6 σ; 3′, 5.5 σ; 1, 1′, 2, 3, 5, 6′, 12′, 5 σ; 6, 7′, 9, 10′, 11, 11′, 13, 14, 4.5 σ; 4′, 5′, 7, 8′, 9′, 4.0 σ; 4, 8, 10, 3.5 σ; 12, 3.5 σ). **e**, Region surrounding the N-terminal helix N1 peripherally attached to the membrane and, **f**, region surrounding the hydrophobic cavity emerging from the membrane with interacting lipids (colors and numbering is as defined in **d**). The molecular surface is shown in white.

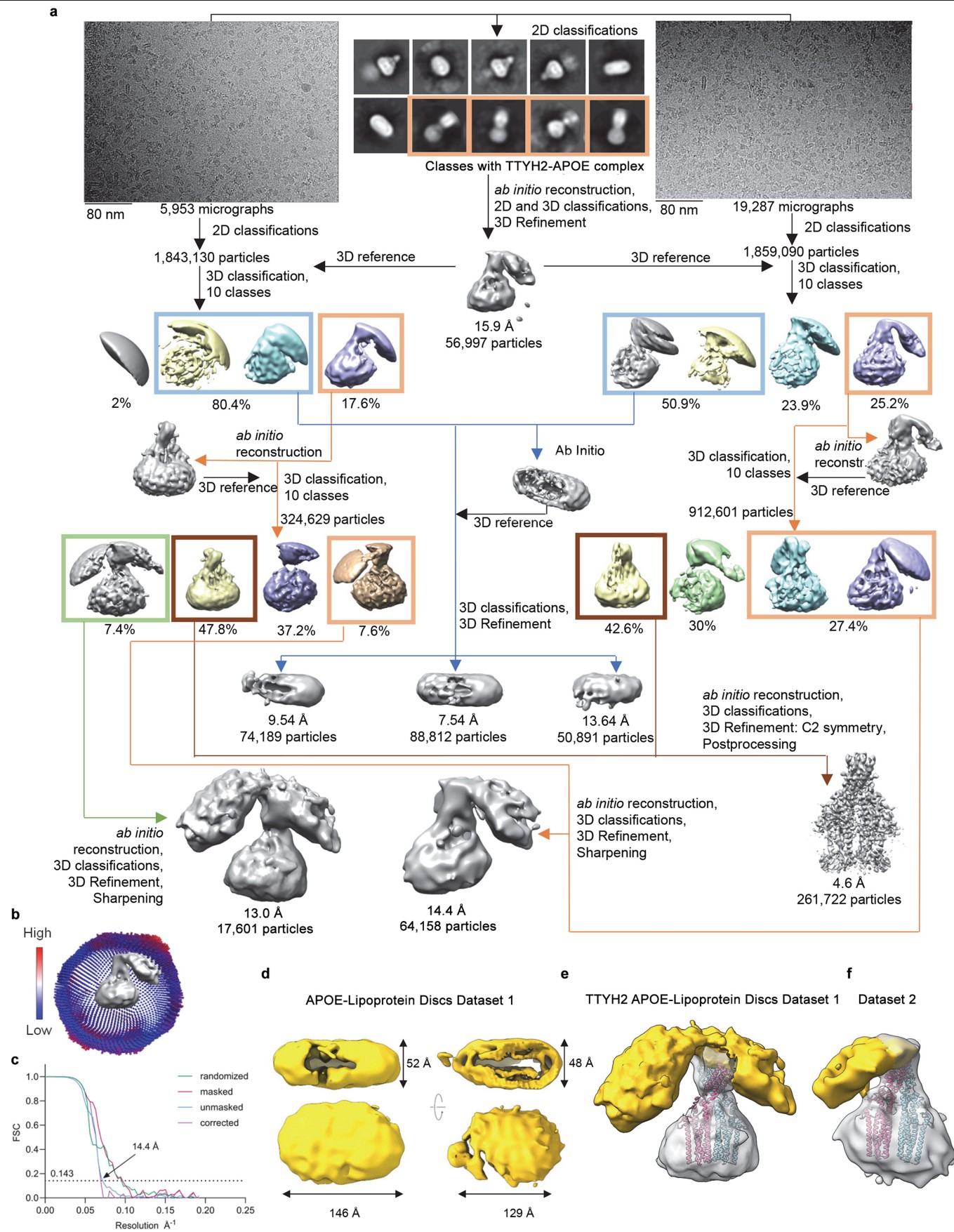

**Extended Data Fig. 8** | See next page for caption.

**Extended Data Fig. 8 | Cryo-EM data processing of TTYH2 in detergent in complex with APOE-lipoproteins. a**, Representative motion and CTF-corrected micrographs (total micrographs=25,220, combined from 2 data collection sessions) and data processing workflow. Particles were picked using previously obtained TTYH2 map for template generation. Initial 2D classification revealed classes of TTYH2 with bound APOE lipoproteins, which were used for further 2D and 3D classifications and 3D refinement to obtain an initial map of the complex. This map was used as a 3D reference in 3D classifications of large particle sets, typically with 10 classes and regularization parameter T = 20, which resulted in three types of classes: TTYH2 in complex with APOE-lipoprotein discs (APOE(lipidated)), isolated ApoE-lipoprotein discs and uncomplexed TTYH2. All classes were refined and the maps postprocessed. **b**, Angular distribution of particle orientations from a class of the TTYH2 in complex with APOE(lipidated) complex with pronounced density of an ApoE lipoprotein disc. **c**, Fourier shell correlation (FSC) plot of the TTYH2 in complex with APOE(lipidated) complex at 14 Å (left). **d**, Cryo-EM maps of disc-shaped APOE-lipoprotein particles in two different sizes with dimensions indicated. Top view is along the disc indicating the high-density region of the polar headgroup regions of the bilayer, bottom view is towards the plane of the bilayer. **e**, **f**, Low resolution reconstructions of TTYH2 in complex with lipoprotein-discs. **e**, Reconstruction at 13 Å containing two bound APOE-lipoprotein discs obtained from the same dataset. **f**, Reconstruction at 15 Å containing a single bound APOE-lipoprotein disc from a second dataset of similar quality. In all cases, the discs bind to the extracellular domain and are oriented towards the membrane.

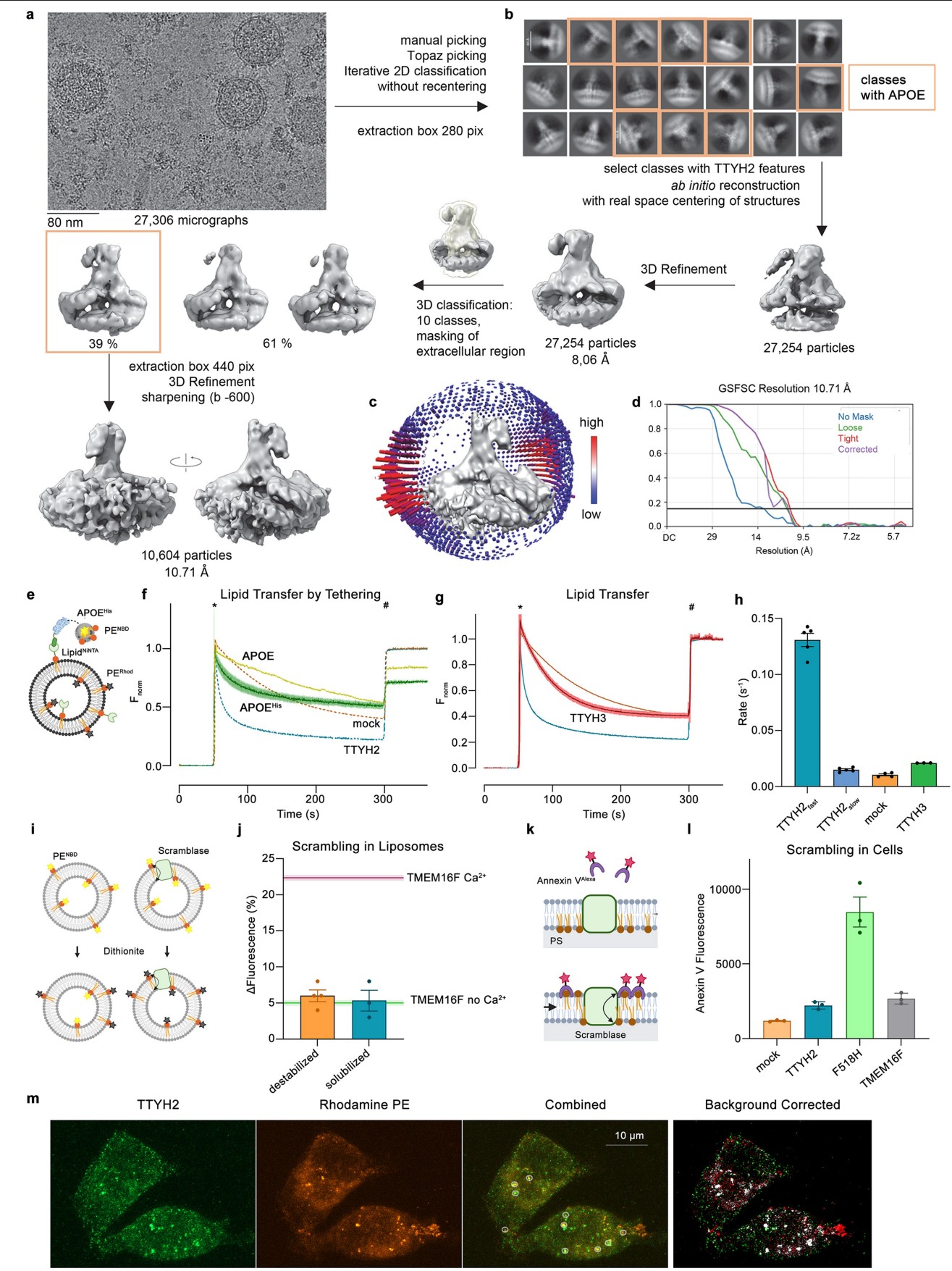

**Extended Data Fig. 9** | See next page for caption.

**Extended Data Fig. 9 | TTYH2 in complex with APOE-lipoprotein in cell-derived vesicles and functional investigations of lipid transport.**
**a**, Representative motion and CTF-corrected micrograph (total micrographs= 27,306) of TTYH2 in cell-derived vesicles with APOE-lipoprotein discs added at 45 μM. **b**, Data processing workflow. An initial set of manually picked particles were used for the training of a Topaz model. Topaz-picked particles were first extracted in a small box size of 280 pixels to aid the alignment of TTYH2 in vesicle membranes. 2D classification revealed classes of TTYH2 with additional density corresponding to ApoE. Classes with APOE were selected for an *ab-initio* reconstruction, which was 3D refined and yielded a map of TTYH2 residing in the vesicle membrane with extra density at the top of TTYH2. The particles were further sorted using 3D classification in cryoSPARC and classes with extended APOE density were picked for particle re-extraction in a bigger box of 440 pixels. The particles were refined to yield a final reconstruction at 10.7 Å. **c**, Angular distribution of particle orientations. **d**, Fourier shell correlation (FSC) plot of the final refined cryo-EM density map of the TTYH2 in complex with APOE(lipidated) in cell-derived vesicles at 10.7 Å with pronounced density of APOE-lipoprotein discs. **e**, Scheme of a lipid transfer assay monitoring the decay of NBD-fluorescence from labeled donor lipids reconstituted into APOE-containing lipoprotein particles quenched by Rhodamine-labeled acceptor lipids in liposomes containing 85% DPPC and DGS-NTA(Ni) lipids. **f**, Traces of the NBD fluorescence from lipoproteins assembled with a His-tagged construct of APOE leading to the tethering of the lipoprotein particle to the surface of the liposome is shown (as average and s.e.m. of three technical replicates) in comparison to a single trace of lipoprotein particles assembled from untagged APOE. The mean fluorescence decay recorded from proteoliposomes containing TTYH2 and its protein-free controls displayed in Fig. 5b are shown as dashed lines for reference. '*' indicates addition of NBD-labeled lipoprotein discs, '#' addition of Triton X-100 leading to the dissolution of liposomes and lipoproteins. **g**, Lipid transfer into TTYH3-containing proteoliposomes is only moderately enhanced compared to mock liposomes. Data show mean of three experiments from a single reconstitution. Mean values of TTYH2 and mock (displayed in Fig. 5b) are shown for comparison. '*' indicates addition of NBD-labeled lipoprotein discs, '#' addition of Triton X-100 leading to the dissolution of liposomes and lipoproteins. **h**, Rate constants from a fit of the fluorescence decay of mock liposomes (n = 4 samples from 2 independent reconstitutions) and TTYH3-containing proteoliposomes (n = 3 samples from 1 reconstitution) to a single exponential function defining the rate of non-specific transfer. The two rate constants obtained for the fit of the TTYH2-mediated fluorescence decay correspond to nonspecific (slow rate, n = 5 samples from 2 independent reconstitutions) and TTYH2-enhanced transfer (fast rate, n = 5 samples from 2 independent reconstitutions). **g, h**, Errors are s.e.m. Differences between the TTYH2-enhanced transfer rate constant (fast) and nonspecific lipid transfer rate constants were analyzed in a two-sample two-sided *t* test and found to be significant (TTYH2 (slow) $p = 0.00004$, mock $p = 0.00003$, TTYH3 $p = 0.00005$). **i-l**, Proteoliposome-based and cellular assays testing the ability of TTYH2 to scramble lipids. **i**, Scheme of a proteoliposome-based lipid scrambling assay monitoring the irreversible bleaching of NBD-labeled lipids located on the outside by the membrane-impermeable reducing agent dithionite. **j**, Results from proteoliposome-based lipid scrambling assays. Shown are the averaged differences in the fluorescence compared to mock reconstituted liposomes from the same lipid batch 200 s after the addition of dithionite. TTYH2 liposomes were reconstituted by two different methods into either preformed and destabilized liposomes (n = 4 from 2 independent reconstitutions) or liposomes assembled from solubilized lipids (n = 3 from 1 reconstitution). The data for the $Ca^{2+}$-activated scramblase TMEM16F in absence and presence of $Ca^{2+}$, as reported previously[50], are shown for comparison (line shows the average of n = 3 repeats from 1 reconstitution). Values of TTYH2-containing liposomes are similar to inactive TMEM16F in its $Ca^{2+}$ free state[63] and thus does not show any indication of scrambling. Errors are s.e.m. Differences in the liposome-based scrambling assay data of solubilized and destabilized liposome preparations containing TTYH2 compared to previously obtained data from TMEM16F in absence and presence of $Ca^{2+}$ were analyzed in a two-sample two-sided *t* test. Differences were found to be non-significant compared to inactive TMEM16F in absence of $Ca^{2+}$ (*p* values are 0.32 and 0.85 for destabilized and solubilized preparations, respectively), and significant compared to active TMEM16F in presence of $Ca^{2+}$ (*p* values are 0.0003 and 0.007 for destabilized and solubilized preparations, respectively). **k**, Scheme of a cellular scrambling assay monitoring the exposure of phosphatidyl serine (PS) to the cell surface by fluorescently labeled Annexin V. **l**, Results from cellular scrambling assays where protein constructs are overexpressed in a HEK293 TMEM16F knockout cell line by transient transfection. Surface expression of TTYH2 under such conditions was confirmed previously[1]. Scrambling is assayed under resting $Ca^{2+}$ concentrations where the activating TMEM16F mutant F518H but not WT TMEM16F mediates basal scrambling[50]. No activity was detected for TTYH2 under such conditions. For each datapoint, Annexin V fluorescence data was averaged from multiple cells and examined over 3 independent experiments, errors are s.e.m. Differences in the cellular scrambling assay data to the constitutively active TMEM16F mutant F518H were analyzed in a two-sample two-sided *t* test and found to be significant for mock, TTYH2 and TMEM16F (TTYH2 $p = 0.03$, mock $p = 0.02$, TMEM16F $p = 0.03$). Differences between data from TTYH2 and TMEM16F samples were found to be non-significant with the *p* value of 0.17. **m**, Confocal fluorescent microscopy images of HEK293 cells showing the colocalization of TTYH2, (green, stained with a polyclonal antibody targeting the protein) and fluorescently labeled PE delivered via APOE-lipoprotein particles applied to the cell media. Shown are images of the individual channels and their combination (center, right). Locations with overlap of both fluorescent markers are indicated by circles. The illustration on the right shows co-localization (white signals) after correcting for the mean intensity of extracellular regions and applying an adjusted background subtraction to eliminate false or random co-localizations. The analysis was performed using the open-source software ImageJ_Coloc2 (https://imagej.net/plugins/coloc-2). The diagrams in **e**, **i** and **k** were created using BioRender (https://www.biorender.com).

# Extended Data Table 1 | Cryo-EM Data and refinement statistics

| | TTYH2 Sb1 | TTYH2 Sb2 | TTYH2 vesicles | TTYH2 APOE delipid. | TTYH3 APOE delipid. | TTYH2 APOE coex.1 | TTYH2 APOE coex.2 | TTYH2 APOE coex.3 | TTYH2 high res. combined data | TTYH2 APOE Cterm. | TTYH2 APOE lipid.1 | TTYH2 APOE lipid.2 | TTYH2 vesicles APOE |
|---|---|---|---|---|---|---|---|---|---|---|---|---|---|
| **Data collection and processing** | | | | | | | | | | | | | |
| Magnification | 130,000 | 130,000 | 130,000 | 130,000 | 130,000 | 130,000 | 130,000 | 130,000 | | 130,000 | 130,000 | 130,000 | 130,000 |
| Voltage (kV) | 300 | 300 | 300 | 300 | 300 | 300 | 300 | 300 | | 300 | 300 | 300 | 300 |
| Electron exposure (e–/Å$^2$) | 68.48 | 69.33 | 62.43 | 61.80 | 68 | 61.68 | 64.88 | 65.16 | | 60 | 61.50 | 59.60 | 62.43 |
| Defocus range (μm) | -1 to -2.4 | -1 to -2.4 | -1 to -2.4 | -1 to -2.4 | -1 to -2.4 | -1 to -2.4 | -1 to -2.4 | -1 to -2.4 | | -1 to -2.4 | -1 to -2.4 | -1 to -2.4 | -1 to -2.4 |
| Pixel size (Å) | 0.3255 | 0.3255 | 0.651 | 0.651 | 0.651 | 0.651 | 0.651 | 0.651 | 0.651 | 0.651 | 0.651 | 0.651 | 0.651 |
| Symmetry imposed | C1 | C1 | C1 | C1 | C1 | C1 | C1 | C1 | C1 | C1 | C1 | C1 | C1 |
| Final particle images (no.) | 75,814 | 71,316 | 31,014 | 59,807 | 5,526[map1] 547,367[map2] | 14,094[map1] 85,781[map2] | 51,001[map1] 81,023[map2] | 46,810 | 2,192,921 | 43,768 | 64,158[map1] 17,601[map2] 88,812[disc1] 74,189[disc2] 50,891[disc3] | 31,510 | 10,604 |
| Map resolution (Å) | 3.7 | 4.26 | 8.23 | 4.02 | 7.59[map1] 2.91[map2] | 6.03[map1] 3.67[map2] | 6.65[map1] 4.04[map2] | 3.63 | 2.74 | 3.85 | 14.4[map1] 13[map2] 7.54[disc1] 9.54[disc2] 13.64[disc3] | 15 | 10.71 |
| FSC threshold | 0.143 | 0.143 | 0.143 | 0.143 | 0.143 | 0.143 | 0.143 | 0.143 | 0.143 | 0.143 | 0.143 | 0.143 | 0.143 |
| Map resolution range (Å) | 3.3-9.2 | 3.9-9.5 | 7.6-10.4 | 3.5-9.0 | 6.8-18.3[map1] 2.91-6.87[map2] | 5.4-12.0[map1] 3.3-9.3[map2] | 6.5-11.2[map1] 3.5-8.9[map2] | 4.0-9.7 | 3.0-5.7 | 3.5-9.4 | 9.9-22[map1] | 11.0-22.8 | 9.7-14.4 |
| **Refinement** | | | | | | | | | | | | | |
| Initial model used (PDB code) | 7p54; 3k1k | 7p54 | 7p54 | 7p54 | 7p5c | 7p54 | 7p54 | 7p54 | 7p54 | 7p54 | 7p54 | 7p54 | 7p54 |
| Model resolution (Å) | 3.9 | | | | 3.0 | | | | 2.8 | | | | |
| FSC threshold | 0.5 | | | | 0.5 | | | | 0.5 | | | | |
| Map sharpening *B* factor (Å$^2$) | -121.2 | | | | -124.9 | | | | -139.6 | | | | |
| Model composition | | | | | | | | | | | | | |
| Non-hydrogen atoms | 7,121 | | | | 6,222 | | | | 7,318 | | | | |
| Protein residues | 908 | | | | 784 | | | | 798 | | | | |
| Ligands | NAG:12 | | | | NAG:8 | | | | NAG:12 PLC:17 CPL:3 CLR:6 | | | | |
| *B* factors (Å$^2$) | | | | | | | | | | | | | |
| Protein | 89.87 | | | | 63.06 | | | | 76.84 | | | | |
| Ligand | 106.98 | | | | 82.49 | | | | 95.64 | | | | |
| R.m.s. deviations | | | | | | | | | | | | | |
| Bond lengths (Å) | 0.004 | | | | 0.003 | | | | 0.013 | | | | |
| Bond angles (°) | 0.721 | | | | 0.535 | | | | 1.782 | | | | |
| Validation | | | | | | | | | | | | | |
| MolProbity score | 2.86 | | | | 1.72 | | | | 2.27 | | | | |
| Clashscore | 18.9 | | | | 4.02 | | | | 8.11 | | | | |
| Poor rotamers (%) | 12.27 | | | | 5.03 | | | | 5.74 | | | | |
| Ramachandran plot | | | | | | | | | | | | | |
| Favored (%) | 96.21 | | | | 98.07 | | | | 96.2 | | | | |
| Allowed (%) | 3.67 | | | | 1.93 | | | | 3.8 | | | | |
| Disallowed (%) | 0.11 | | | | 0 | | | | 0 | | | | |

# Reporting Summary

Please do not complete any field with "not applicable" or n/a. Refer to the help text for what text to use if an item is not relevant to your study.
For final submission: please carefully check your responses for accuracy; you will not be able to make changes later.

## Statistics

For all statistical analyses, confirm that the following items are present in the figure legend, table legend, main text, or Methods section.

| n/a | Confirmed | |
|---|---|---|
| ☐ | ☒ | The exact sample size (*n*) for each experimental group/condition, given as a discrete number and unit of measurement |
| ☐ | ☒ | A statement on whether measurements were taken from distinct samples or whether the same sample was measured repeatedly |
| ☐ | ☒ | The statistical test(s) used AND whether they are one- or two-sided *Only common tests should be described solely by name; describe more complex techniques in the Methods section.* |
| ☐ | ☒ | A description of all covariates tested |
| ☐ | ☒ | A description of any assumptions or corrections, such as tests of normality and adjustment for multiple comparisons |
| ☐ | ☒ | A full description of the statistical parameters including central tendency (e.g. means) or other basic estimates (e.g. regression coefficient) AND variation (e.g. standard deviation) or associated estimates of uncertainty (e.g. confidence intervals) |
| ☐ | ☒ | For null hypothesis testing, the test statistic (e.g. *F*, *t*, *r*) with confidence intervals, effect sizes, degrees of freedom and *P* value noted *Give P values as exact values whenever suitable.* |
| ☒ | ☐ | For Bayesian analysis, information on the choice of priors and Markov chain Monte Carlo settings |
| ☒ | ☐ | For hierarchical and complex designs, identification of the appropriate level for tests and full reporting of outcomes |
| ☒ | ☐ | Estimates of effect sizes (e.g. Cohen's *d*, Pearson's *r*), indicating how they were calculated |

*Our web collection on statistics for biologists contains articles on many of the points above.*

## Software and code

Policy information about availability of computer code

| Data collection | Cryo-EM images were collected using EPU 2.9 with AFIS for faster acuisition, confocal microscopy images were acquired using ZEN 2. AFM force curves were acquired using the JPK SPM 6.4.22 software. |
|---|---|
| Data analysis | Relion 4.0.4, Cryosparc 4.3 - 4.5, Chimera 1.17, ChimeraX 1.3, Phenix 1.21, Coot 0.9.8, GraphPad Prism 10, Fiji 2.14.0/1.54f, JPK-Data 6.4, JPK Data Processing 6.4, Python 3.12. |

For manuscripts utilizing custom algorithms or software that are central to the research but not yet described in published literature, software must be made available to editors and reviewers. We strongly encourage code deposition in a community repository (e.g. GitHub). See the Nature Portfolio guidelines for submitting code & software for further information.

## Data

Policy information about availability of data

All manuscripts must include a data availability statement. This statement should provide the following information, where applicable:
- Accession codes, unique identifiers, or web links for publicly available datasets
- A description of any restrictions on data availability
- For clinical datasets or third party data, please ensure that the statement adheres to our policy

The three-dimensional cryo-EM density maps were deposited in the Electron Microscopy Data Bank under the following accession numbers: EMD-51106 for TTYH2 in complex with Sb1, EMD-53290 for TTYH2 in complex with Sb2, EMD-53291 for TTYH2 in complex with Sb1 in cell-derived vesicles, EMD-53289 for TTYH2 in

complex with delipidated ApoE, EMD-53301 and EMD-53297 for TTYH2 complexes with coexpressed ApoE from dataset 1, EMD-53303 and EMD-53302 for TTYH2 complexes with coexpressed ApoE from dataset 2, EMD-53304 for TTYH2 complex with coexpressed ApoE from dataset 3, EMD-51108 for TTYH2 in complex with lipids, EMD-53293 for TTYH2 in complex with C-terminal ApoE fragment, EMD-53251 and EMD-53269 for TTYH2 complexes with lipidated ApoE from dataset 1, EMD-53280 for TTYH2 in complex with lipidated ApoE from dataset 2, EMD-53272, EMD-53273 and EMD-53271 for ApoE lipoprotein discs from dataset 1, and EMD-53292 for TTYH2 in complex with lipidated ApoE in cell-derived vesicles. Relevant coordinates were deposited in the Protein Data Bank under the following accession numbers: 9G6X for TTYH2 in complex with Sb1, 9G71 for TTYH2 in complex with lipids, and 9QNR for TTYH3. The cryo-EM structure of TTYH2 (PDB: 7P54) was used for building initial models of TTYH2 in complex with Sb1 and in complex with lipids and was fitted into all cryo-EM maps of TTYH2 in complex with ApoE. The structures of the GFP-binding nanobody and of of TTYH3 (PDB: 7P5C) were used as initial models for building of the Sb1 and TTYH3 models, respectively. The structure of N-terminal domain of ApoE (PDB: 1B68) and the structure of TTYH2 (PDB: 7P54) were used to create illustrations in Biorender.  A source data file is provided with this paper.

## Research involving human participants, their data, or biological material

Policy information about studies with human participants or human data. See also policy information about sex, gender (identity/presentation), and sexual orientation and race, ethnicity and racism.

| | |
|---|---|
| Reporting on sex and gender | N/A |
| Reporting on race, ethnicity, or other socially relevant groupings | N/A |
| Population characteristics | N/A |
| Recruitment | N/A |
| Ethics oversight | N/A |

Note that full information on the approval of the study protocol must also be provided in the manuscript.

# Field-specific reporting

Please select the one below that is the best fit for your research. If you are not sure, read the appropriate sections before making your selection.

☒ Life sciences ☐ Behavioural & social sciences ☐ Ecological, evolutionary & environmental sciences

For a reference copy of the document with all sections, see [nature.com/documents/nr-reporting-summary-flat.pdf](http://nature.com/documents/nr-reporting-summary-flat.pdf)

# Life sciences study design

All studies must disclose on these points even when the disclosure is negative.

| | |
|---|---|
| Sample size | Sample sizes were based on standards commonly used in the field. Experiments with liposomes and binding assays were performed in multiple biological and technical replicates with similar results and further inclusion of data did not change the results. Confocal microscopy samples were prepared multiple times and similar images were obtained. For Cryo-EM datasets sufficient number of micrographs were collected to obtain best possible 3D reconstructions for each preparation type. Complete Cryo-EM statistics are provided in Extended Data Table 1. |
| Data exclusions | Data selection for Cryo-EM is illustrated in Extended Data Table 1 and  Extended Data figures. Representative confocal microscopy images were chosen to be displayed in the manuscript. Scrambling assay data from unsuccessful reconstitutions with aggregated protein were excluded. Otherwise, no data were excluded. |
| Replication | Cryo-EM datasets of TTYH2 coexpressed with ApoE were acquired from three different samples resulting in similar cryo-EM maps. Cryo-EM datasets of TTYH2 in complex with lipidated ApoE were acquired from two different samples resulting in similar cryo-EM maps. Subcellular fractionation of HEK293 and N2A cells was repeated three times each with similar results. Lipid transfer assay was performed in two biological and multiple technical (TTYH2) or three technical (TTYH3) replicates and all replications were successful. Lipid transfer assay with ApoE tethering was performed in three technical replicates. Sybody displacement assay was performed in four biological and multiple technical replicates and all replications were successful. Liposomal scrambling assay was performed in two biological and four technical replicates for the destabilized preparations and in three biological and seven technical replicates for the solubilized preparation. Two of three preparations using solubilized lipids were unsuccessful as the protein aggregated during the process, the two destabilized preparations were successful. Cellular scrambling assay was performed in three biological replicates. All replications were successful for the cell-based scrambling assay. |
| Randomization | Randomization was used during cryo-EM data processing for the quality assessment of obtained reconstructions by randomly splitting the dataset in half, each used for obtaining a separate 3D reconstruction and measuring the correlation between the two. |
| Blinding | Not applicable, as we did not perform experiments requiring subjective interpretation. |

# Reporting for specific materials, systems and methods

We require information from authors about some types of materials, experimental systems and methods used in many studies. Here, indicate whether each material, system or method listed is relevant to your study. If you are not sure if a list item applies to your research, read the appropriate section before selecting a response.

## Materials & experimental systems

| n/a | Involved in the study |
|-----|----------------------|
| ☐ | ☒ Antibodies |
| ☐ | ☒ Eukaryotic cell lines |
| ☒ | ☐ Palaeontology and archaeology |
| ☒ | ☐ Animals and other organisms |
| ☒ | ☐ Clinical data |
| ☒ | ☐ Dual use research of concern |
| ☒ | ☐ Plants |

## Methods

| n/a | Involved in the study |
|-----|----------------------|
| ☒ | ☐ ChIP-seq |
| ☒ | ☐ Flow cytometry |
| ☒ | ☐ MRI-based neuroimaging |

## Antibodies

| | |
|---|---|
| Antibodies used | Antibodies purchased from ThermoFisher Scientific: rabbit anti-TTYH2 (PA5-34395), rabbit anti-ApoE (16H22L18), rabbit anti-EEA1 (F.43.1), rabbit anti-LAMP1 (107), rabbit anti-Na,K-ATPase (ST0533), rabbit anti-SERCA (JM10-20), rabbit anti-Rab7 (PA5-23138), rabbit anti-GM130 (ARC0589), mouse anti-Rab9 (Mab9), goat anti-rabbit HRP-conjugated (31460), goat anti-rabbit Alexa 488-conjugated (A-11008), goat anti-mouse Alexa 594-conjugated (A-11032). Anti-Rab11 (20229-1-AP) was obtained from Proteintech. Anti-TTYH2 antibody used for immunocytochemistry was purchased from antibodies online: https://www.antibodies-online.com/antibody/7124004/anti-Tweety+Homolog+2+TTYH2+AA+455-534+antibody/. |
| Validation | All primary antibodies were used for western blotting or immunocytochemistry of samples from human-derived or mouse-derived cell lines. All primary antibodies were validated by the supplier for recognition of their human-specific antigens. Rabbit anti-ApoE (16H22L18), anti-EEA1 (F.43.1), rabbit anti-Rab11 (20229-1-AP; Proteintech), rabbit anti-Na,K-ATPase (ST0533), rabbit anti-SERCA (JM10-20), rabbit anti-Rab7 (PA5-23138) and rabbit anti-GM130 (ARC0589) antibodies were also validated by the supplier for recognition of their mouse-specific antigens. Primary antibodies used for western blotting including rabbit anti-TTYH2 (PA5-34395), rabbit anti-ApoE (16H22L18), rabbit anti-EEA1 (F.43.1), rabbit anti-LAMP1 (107), rabbit anti-Rab11 (20229-1-AP; Proteintech), rabbit anti-Na,K-ATPase (ST0533), rabbit anti-SERCA (JM10-20), rabbit anti-Rab7 (PA5-23138), rabbit anti-GM130 (ARC0589) were all validated for usage in western blot analysis by the supplier and in multiple publications. The mouse anti-Rab9 (Mab9) antibody used in immunocytochemistry was validated by the supplier and in multiple publications for this application. The anti-TTYH2 antibody obtained from antibodies online that was used in immunocytochemistry was validated by western blotting by the supplier. In addition, both anti-TTYH2 antibodies and the anti-ApoE antibody were validated by us for recognition of purified recombinant proteins using western blotting. |

## Eukaryotic cell lines

Policy information about cell lines and Sex and Gender in Research

| | |
|---|---|
| Cell line source(s) | HEK293T (ATCC, CLR-1573), HEK293S GnTI– (ATCC, CLR-3022), Neuro-2A (ATCC, CCL-131), HEK293 TMEM16F knockout provided by Huanghe Yang. |
| Authentication | No further authentication was performed. |
| Mycoplasma contamination | The cell lines were tested and are free from mycoplasma contamination |
| Commonly misidentified lines (See ICLAC register) | No commonly misidentified cell lines were used in this study. |

## Plants

| | |
|---|---|
| Seed stocks | N/A |
| Novel plant genotypes | N/A |
| Authentication | N/A |

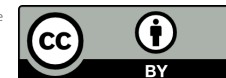

