## [Peer Review File · Nature]

Interactions between TTYH2 and ApoE facilitate endosomal lipid transfer

Corresponding Author: Professor Raimund Dutzler

Version 0:

Reviewer comments:

Referee #1

(Remarks to the Author)

The manuscript titled "Interactions between TTYH2 and ApoE facilitate endosomal lipid transfer" reports identification of antibodies against TTYH2, discovery of physical interactions and formation of a stable complex between TTYH2 and ApoE, cellular localization of the TTYH2-ApoE complex, structure determination of TTYH2 and TTYH2 in complex with ApoE by cryoEM, and demonstration of lipid transfer function of the TTYH2-ApoE complex. The study is rigorous and comprehensive and is a brilliant tour de force that shows a mastery of an array of cellular and molecular level experimental approaches. The experiments are well-designed and expertly executed, and the interpretations are accurate and often self-restrained. The main conclusions are truly ground-breaking and seem to support several previous speculations. The study also establishes a new avenue of research that will lead to deeper understanding of the complex nature of lipid transfer and homeostasis in a cell.

I have no major concerns or criticisms. I do find the choice of single molecule force spectroscopy somewhat odd, and would prefer to see a more traditional binding assay, however, this is a minor point and not a call for new experiments. I also feel that the resolutions of the TTYH2-ApoE complex in either detergent micelle or vesicle/lipidic environment may have room for improvement, however, the authors are cautious and the conclusions are adequate.

Referee #2

(Remarks to the Author)

Sukalskaia et al. have reported a novel interaction between TTYH2 and APOE which may contribute to the endosomal lipid transfer. Using synthetic nanobody bound TTYH2 structure, they have reconstructed several CryoEM density maps of different forms of human APOE3 including nonlipidated APOE3, APOE3 C-terminal domain (APOECTD) and lipidated APOE particles binding to full length TTYH2 on the lipid nanodisc. Additionally, they also provided supportive data including cellular staining of colocalized TTYH2 and APOE, binding between non-lipidated APOE3 and TTYH2 measured by single molecule force spectroscopy (SMFS) and lipid transfer assays.

In summary, the potential discovery of APOE-TTYH2 interaction and its impact on endosomal lipid transfer is both interesting and novel. The structural investigation on the APOE-TTYH2 complex with a variety of APOE forms by CryoEM is also impressive. However, there are some important concerns rise from the uncertainty of current CryoEM density maps, the lack of alternative methods to cross-verify, and the unclear nature of physiological relevance of the interaction (at least from current data). Concerns to address are listed below.

Major:

1. It is unclear how the authors managed to use Sb1 to pulldown the APOE with TTYH2 because later data showed that the binding interface of Sb1-TTYH2 complex significantly overlaps with APOE-TTYH2 binding interface. Do the authors have an explanation?
2. The APOE-TTYH2 structure is not yet convincing though this be able to be clarified with some additional results.
 - a. The resolution if all complex density maps are misleading because all the near-atom resolutions resulted from a highly impacted region inside TTYH2, where the resolutions of actual binding interface were all about 7Å (Fig. S4-7). This may be correct, but the authors should clearly report the details of this result.

- b. Following the point 2a, the resolution of APOE density in the complex of delipidated APOE and TTYH2 was about 7Å (Fig. S4E). At this resolution, could the authors ensure that this is the APOE density? Could the authors apply further 3DVA masking on that APOE density to improve the regional refinement or just obtain more particles with additional imaging to improve the resolution? At least to the extent where some of the APOE density could be modeled? The same issues can be found in other complex maps.
- c. If one compares the final 3D reconstructions between APOE-TTYH2 and APOECTD-TTYH2 (Fig. 3B&C, S4&S5), both APOE density maps suggest a very compacted structure of APOE C-terminal domain instead of a major α -helix showed by previous literature [1-3]. Certainly, the APOECTD might undergo a conformational change upon binding to TTYH2. However, the authors need to either show high resolution modeling or apply mutagenesis to verify their structure results.
- d. Related to previous point 2c, the APOE-TTYH2 complex density map from the overexpression system raises more concerns of the CryoEM data (Fig. 3D). Regardless of the heterogeneity that the authors mentioned, the real elusive observation is the change of stoichiometry and binding interface between APOE and TTYH2. Now the binding stoichiometry becomes 2 vs. 2 instead of 1 APOE to two TTYH2 molecules. The binding domain of TTYH2 slides to the side compared to the inner face side (Fig. 3D vs. 3B&C). Considering the relative low resolution of APOE in all the maps, the authors need to be able to explain such a difference and also provide more binding results to reconcile this inconsistency.
- e. The reconstruction of lipidated APOE complex with TTYH2 in Fig. 4 was not completely satisfying. First, the low resolution of all domains make this map not completely determined. Second, the box size was too small so that it cut off the edge of lipidated APOE particles where a spherical boundary could even be observed in the final reconstruction (Fig. 4B&C).
- f. As partially mentioned in point 2e, there are concerns on the stringent box size chosen by the authors in all the complex reconstructions. First, have the authors tried some larger box sizes to pick particles and align them to ensure the extra density was not from another TTYH2 nanodisc or some other background average? Because TTYH2 has been shown with trans-interactions by itself [4] at the exact binding interface between APOE and TTYH2 reported in this study. Second, could the authors also provide 2D class images of the final 3D reconstructions in the supplementary figure?
- g. Maybe it's an orientation issue, but it's hard to understand the conformation of TTYH2 in the nano-gold labeled APOE complex with TTYH2 (Fig. 3a). Could the authors perhaps just fit TTYH2 model into that low resolution map and indicate where the labeled APOE could be? That would be informative.

3. The interactions between APOE-TTYH2 were only determined by single molecule force spectroscopy (SMFS) in addition to CryoEM analysis (Fig. 2E&F). It directly measures the molecular binding force instead of more popular biochemical assays such as ITC and SPR. Moreover, the measurement in Fig. 2E&F was between APOE3 and TTYH2. Therefore, it raises three questions:

- a. In the same assay, the authors should provide a negative control such as APOE3 NTD and a positive control such as Sb1.
 - b. The authors should verify the binding affinity with ITC or SPR to provide true KD values, which will be more informative for the readers to comprehend the binding affinity of APOE3-TTYH2 with cross-reference from other interactions.
 - c. Have the authors assessed the binding affinity changes among APOE2/3/4? This will be very informative and improve the significance of determining the APOE-TTYH2 interactions.
4. The authors should include mutagenesis verifications to demonstrate the binding interface observed in the CryoEM density maps, at least mutating the key residues in the APOE C terminal region.
5. The functional investigation of TTYH2-APOE mediated endosomal lipid transfer was not strong.
- a. The authors should include truncated TTYH2 as a negative control in their liposome assay in Fig. 5. It's hard to tell whether it should be E1/2 or E5/6 based on current maps but the authors might know better.
 - b. The authors should show the difference of lipid transfer from exogenous APOE to HEK cells with or without TTYH2 expression.
 - c. The authors also should determine the changes of APOE-TTYH2 binding at neutral pH vs. acidic pH to demonstrate that the binding occurs in endosome/lysosome. With the current buffer system (Fig. 5B-D, pH 7.4), why would the lipid transfer occur post endocytosis? It should be processed on the cell membrane.

6. The experiments show interactions of apoE from HEK cells with TTYH2 or exogenous apoE artificially lipidated particles with TTYH2. HEK cells produce a very poorly lipidated form of apoE that differs very much for cells that are the main producers of apoE in vivo which would include hepatocytes peripherally and astrocytes in the CNS. It would be important to determine physiological relevance and potential relevance to disease to study whether similar interactions of lipidated apoE produced by the main cells that produce apoE in the body (hepatocytes and astrocytes) occur with TTYH2. Also, would similar results of interactions between TTYH2 be seen with other similar apolipoproteins, specifically the most similar to apoE, apoA1? This would help to understanding the specificity of the TTYH2-apoE vs. interactions with other apolipoproteins.

Minor points:

1. In fig. 2C, it would be better to include endosomal marker such as EEA1 or Rab5/7 to distinguish intracellular localizations of the TTYH2.
2. The authors should provide at least NS-TEM images to demonstrate the disc-shapes of the APOE particles with their detergent-solubilized lipidation methods. Because this method may lead to a spherical shape [5] instead of disc-shapes, which might impact the interpretation of the final cryoEM map in fig. 4.
3. Scale bars are all missing in the panels. Please add them accordingly.

References:

- [1] Wang, G et al. "Conformations of human apolipoprotein E(263-286) and E(267-289) in aqueous solutions of sodium

- dodecyl sulfate by CD and ¹H NMR." *Biochemistry* vol. 35,32 (1996): 10358-66. doi:10.1021/bi960934t
- [2] Chen, Jianglei et al. "Topology of human apolipoprotein E3 uniquely regulates its diverse biological functions." *Proceedings of the National Academy of Sciences of the United States of America* vol. 108,36 (2011): 14813-8. doi:10.1073/pnas.1106420108
- [3] Stuchell-Brereton, Melissa D et al. "Apolipoprotein E4 has extensive conformational heterogeneity in lipid-free and lipid-bound forms." *Proceedings of the National Academy of Sciences of the United States of America* vol. 120,7 (2023): e2215371120. doi:10.1073/pnas.2215371120
- [4] Li, Baobin et al. "Structures of tweety homolog proteins TTYH2 and TTYH3 reveal a Ca²⁺-dependent switch from intra- to intermembrane dimerization." *Nature communications* vol. 12,1 6913. 25 Nov. 2021, doi:10.1038/s41467-021-27283-8
- [5] Peters-Libeu, Clare A et al. "Model of biologically active apolipoprotein E bound to dipalmitoylphosphatidylcholine." *The Journal of biological chemistry* vol. 281,2 (2006): 1073-9. doi:10.1074/jbc.M510851200

Referee #3

(Remarks to the Author)

Integral membrane proteins in the Tweety family were previously hypothesized to participate in the transfer of lipids from soluble lipid carriers to the cell. Here the authors have identified ApoE as a lipid carrier that interacts directly with the Tweety protein TTYH2. They used sub cellular fractionation and imaging studies to demonstrate that both proteins co-localize and to endosomal compartments (but see point 3 below), and they demonstrate that ApoE both in its lipid bound and apo state stably interacts with TTYH2, not least of all by obtaining structures of the complexes by cryo-EM. They propose a model in which ApoE first binds to its receptor on the cell surface, is internalized into endosomes, where it dissociates from its receptor due to pH changes and is therefore able to associate with TTYH2. They propose that TTYH2 helps in the transfer of lipids from the lipoprotein particle and into cellular membranes--via a cavity/channel starting in the soluble part of TTH2 that extends from the membrane and into the membrane. They have developed an assay that shows that lipid transfer from lipoprotein particle to liposomes is (somewhat) faster in the presence of TTH2 than not, although this may just be due to the fact that TTH2 tethers the lipoprotein particle to the liposome (they did not control for that possibility in Fig 5, but they need to; see #1 below).

I am very intrigued by this manuscript because of its timeliness. Their story critically supports a fundamental emerging concept in protein-mediated lipid transfer, namely that integral membrane proteins are critical in transferring lipids out of and into membranes, between soluble carriers and the membrane, by lowering the energy barrier of this transfer and thus affecting rates. There are some examples of this for the transport of cholesterol out of lysosomes by the Niemann-Pick protein complex and also in sterol transfer in Wnt signaling. In the NPC there is a soluble portion and an integral membrane portion, featuring a contiguous channel serving for sterol insertion into and through the membrane. There is also a recent paper on BioRxiv for a complex featuring a BLTP family lipid transport protein, which also has an integral membrane portion (Kang Y, Lehmann KS, Vanegas J, Long H, Jefferson A, et al. 2024. bioRxiv), presumably to facilitate transfer of glycerophospholipids into the membrane and probably scrambling (although the authors do not address the latter possibility). There is also a low resolution structure of the ATG2 (lipid transport tube)-ATG9 (integral membrane protein) complex, which supports the direct transfer of lipids between soluble carrier and integral membrane protein (Wang, Y., et al. (2024). "Structural basis for lipid transfer by the ATG2A-ATG9A complex." NSMB) Note that the lipid transport field, at least for transport at contact sites, has until very recently been oblivious to intramembrane dynamics. They thought soluble proteins do everything!

Here are issues that I feel the authors must address:

1. They need to establish in their in vitro lipid transfer assay that the reason TTYH2 facilitates transfer is in fact because it helps with lipid insertion, not just that it is tethering the ApoE lipoprotein particle near the liposome membrane so that lipid transfer is actually even feasible. (For an example of the power of tethering, please see PMID 36282247 vs 35764626). They could probably dissect out the contribution of tethering versus membrane insertion by assessing the effect of tethering ApoE via a hexahistidine tag to liposomes versus TTHY-containing liposomes. Is there any difference at all?
2. As said, I love their model, but I would suggest that it might be energetically difficult to keep inserting lipids into one leaflet of a membrane bilayer, as suggested in their Fig. 6 (but not 5), thus creating leaflet asymmetry, and I would propose that a scramblase activity is also involved in this process to distribute incoming lipids between membrane leaflets. I suspect very strongly, as based on their very, very exciting higher resolution structure from this manuscript, showing lipids bound to TTH2 (Fig. 3f), that TTH2 can also transfer lipids between membrane bilayers—ie, it has scramblase activity—so that the leaflets stay the same size. This structure showing lipids appearing to be scrambled is super-exciting in my mind and would be an excellent rationale for publishing this work in this versus a more plebian journal.

The author's lab has worked on scramblases before, and so they know how to do in vitro scrambling assays to test this possibility. They should do these.

3. The microscopy should be improved for Fig 2c, which show TTYH2 localizing on punctae. The authors should demonstrate that TTYH2 co-localizes with an endosome marker to definitively show that the punctae are, in fact, endosomes.
4. The authors admit in the last paragraph of the manuscript's discussion section that functional/cellular studies" are required

to confirm the importance of TTYH2 in lipid transfer. They should do these. Ideally, they could dissect the role of TTYH2 in tethering ApoE lipoprotein particles to membranes and actual membrane insertion and maybe scrambling. The reviewer has experienced that KOing scrambling activity can be very difficult (due to limitations in the in vitro assays), so this dissection may/or may not be possible, but an attempt should be made as success would strengthen the manuscript. A lack of functional data should not be acceptable for a high-tier journal like this one.

5. If they can demonstrate a role for TTYH2 in intramembrane lipid dynamics, this role should be emphasized more in the discussion as the general importance of integral membrane proteins for lipid transfer is not yet firmly established.

The manuscript is well written and very clear.

Once the conclusions are strengthened as described above, this manuscript represents a significant advance in our understanding of the mechanisms underlying lipid trafficking.

Version 1:

Reviewer comments:

Referee #1

(Remarks to the Author)

I am satisfied with authors' response, and have no further comments.

Referee #2

(Remarks to the Author)

In this revision and rebuttal, Sukalskaia et al. have made considerable efforts to address this reviewer's original comments and have provided additional data to clarify their results. Although several critical points remain only partially resolved due to technical limitations, the reviewer acknowledges the authors' efforts and recognizes that three key aspects have been addressed using alternative approaches:

1. Validation of the cryoEM density map, supported by the absence of APOE density in the TTYH2 mutant or TTYH3 mixed with APOE;
2. Assessment of relative binding affinity using Microscale Thermophoresis;
3. Liposome-based assays demonstrating lipid transport activity.

While the reviewer has some disappointment with the overall data quality – especially given the group's expertise in structural biology and the manuscript's submission to this journal – the reviewer is OK with this revision, if Figs R1, R3, and R4 are included in the main paper. These figures are crucial for readers to understand the data processing and the relative binding strengths involved in this interaction, which will be important for future studies. The reviewer also recommends that the authors include a discussion of the study's limitations and the technical challenges encountered.

Referee #3

(Remarks to the Author)

Integral membrane proteins in the Tweety family were previously hypothesized to participate in the transfer of lipids from soluble lipid carriers to the cell. Here the authors have identified ApoE as a lipid carrier that interacts directly with the Tweety protein TTYH2. They used subcellular fractionation and imaging studies to demonstrate that both proteins co-localize to endosomal compartments, and they demonstrate that ApoE both in its lipid-bound and apo states stably interacts with TTYH2 (and not other TTYHs), not least of all by obtaining structures of the complexes by cryo-EM. They propose a model in which ApoE first binds to its receptor on the cell surface, is internalized into endosomes, where it dissociates from its receptor due to pH changes and is therefore able to associate with TTYH2. They propose that TTYH2 helps in the transfer of lipids from the lipoprotein particle and into cellular membranes-- via a cavity/channel starting in the soluble part of TTYH2 that extends from the membrane and into the membrane. They have demonstrated via an in vitro assay that TTYH2 facilitates the transfer of lipids between ApoE and membrane.

I am very intrigued by this manuscript because of its timeliness. Their story critically supports a fundamental emerging concept in protein-mediated lipid transfer, namely that integral membrane proteins are critical in transferring lipids out of and into membranes, between soluble carriers and the membrane, by lowering the energy barrier of this transfer and thus affecting rates. The authors have responded thoughtfully and to my satisfaction to any outstanding concerns I had regarding the previous version, and I strongly support publication of this manuscript in Nature.

We thank the reviewers for their generally constructive comments. For our revision, we have carefully addressed all remarks by either including novel data that strengthen our original claims, or by stating where we think that requests go beyond the scope of this work, which presents first investigations of a novel lipid transport mechanism. Due to the specific properties of ApoE and the complexity of its interactions with lipids and TTYH2, the seemingly straightforward structural characterization of the TTYH2-ApoE complexes, the quantification of binding affinities or the assay of lipid transfer are severely complicated by the conformational heterogeneity of the sample and by non-specific interactions that hamper *in vitro* experiments. We thus hope to convince the reviewers that while our current results have reached the limit of what is achievable at this stage, they establish a compelling novel mechanism of lipid transport that provides a solid foundation for follow-up studies that will be supported by a recently secured grant.

For our revision, we have carried out the following experiments and analyses:

To address queries related to the processing and characterization of our structural data:

- We have re-analyzed our data to refute the possibility that the observed structural features are artifacts of data processing. Particularly, we were able to show that the density assigned to the discoidal ApoE-lipoprotein discs does not originate from neighboring particles and that the structure of the complex of ApoE that was co-expressed with TTYH2 bears very similar features to the equivalent complex assembled from separately purified components (shown as Fig. 3d and Extended Data Fig. 6).
- Additionally, we have provided novel structural data of samples containing TTYH3 and a TTYH2 mutant with altered residues of the presumed ApoE binding epitope in presence of nanogold-labeled ApoE. In both cases we did not detect density for the gold label that was observed under the same conditions with TTYH2, indicating that binding to these protein constructs is severely compromised (shown as Extended Data Fig. 4g, h).
- We also obtained a high-resolution structure of a sample containing TTYH3 mixed with ApoE, where we did not detect any features of the bound apolipoprotein found in case of TTYH2 (shown as Extended Data Fig. 5a-f). The following sentence was added to the manuscript on Page 10, lines 191-195: Consequently, we did not find similar density of the

apolipoprotein upon addition of gold-labeled ApoE to either TTYH3 or a mutant of TTYH2 where four residues of the presumed binding site were mutated to their equivalent positions in TTYH3 (Extended Data Fig. 4g, h) and in a high-resolution structure of TTYH3 that was incubated with unlabeled ApoE (Extended Data Fig. 5a-f).

- With respect to comments regarding the quality of our structural data, we want to emphasize that the low resolution of the ApoE density is likely a consequence of the conformational heterogeneity of the protein, where the N-terminus forms a tightly folded four-helix bundle, while the C-terminal domain, which carries the TTYH2 interaction region, was described to oligomerize and loosely interact with the N-terminal domain. Consequently, it is justified not to expect a well-defined unique binding mode leading to high resolution features, which would allow the unambiguous localization of the binding epitope on ApoE.

To address queries related to the binding of ApoE to TTYH2, we have carried out experiments to better characterize their interaction:

- To strengthen the conclusions of SMFS experiments, we have included novel data showing the interaction of TTYH2 with the immobilized sybody Sb2 used as positive control and data where we show competition of TTYH2 binding to immobilized Sb2 and ApoE by addition of soluble Sb2. To probe the effect of low pH, we have performed SMFS experiments at pH 5.5, which demonstrate a strong interaction between TTYH2 and delipidated ApoE (shown as Extended Data Fig. 3i-k). The following sentences were added to the manuscript on Page 8, lines 158-165: Its linear dependence on the logarithm of the loading rate is a hallmark of a specific interaction and can be used for the determination of the kinetic off-rate constant k_{off} , amounting to about 1 s^{-1} at neutral and acidic pH (Fig. 2f, Extended Data Fig. 3i). A similar binding behavior as for ApoE was observed for the immobilized Sb2 used as positive control (Extended Data Fig. 3j), while addition of soluble Sb2 added to the surrounding buffer has successfully competed the interaction with both, the immobilized Sb2 and ApoE (Extended Data Fig. 3k), further underlining the specificity of their binding and the overlap of their epitopes.

- We have also attempted the use of orthogonal approaches to characterize binding, such as Surface Plasmon Resonance Spectroscopy (SPR) and Microscale Thermophoresis (MST). In case of SPR, we were unable to record any signal even for the binding of Sb2 used as positive control, which made the method unsuitable for our purposes. In case of MST, we have detected saturable binding of delipidated ApoE to TTYH2. However, while the recorded affinity was consistently higher for TTYH2, we also found binding to the iron transporter ferroportin (FPN) used as negative control, which emphasizes the properties of ApoE to engage in non-specific interactions with detergent solubilized membrane proteins. These non-specific interactions complicate the interpretation of classical binding experiments and increase the importance of competition-based assays as provided in our study.
- We also want to point out that while we consider the remarks concerning the interaction of TTYH2 with other ApoE isoforms or other apolipoproteins as interesting, we believe that such investigations would be beyond the scope of the current study. For the same reason, we have not included experiments on spherical lipoprotein particles or lipoprotein particles obtained from blood samples at this stage. We do not think that such experiments would change the general conclusions drawn in our work, since the observed interactions are confined to the C-terminal domain of ApoE and are largely independent its lipidation state, while the residues that define the ApoE isoforms are located in the linker region immediately following the N-terminal domain.

Lipid transport:

- To address the reviewer questions concerning the specificity of TTYH2-mediated lipid transfer, we have investigated the effect of tethering of ApoE-lipoproteins to the surface of liposomes not containing any membrane protein. Although in this case we did observe increased fluorescence quenching compared to the background quenching obtained from non-tethered lipoproteins, this effect is much smaller than in proteoliposomes containing TTYH2, thus underlining that the observed lipid transfer cannot be explained by the mere

tethering of lipoproteins to the membrane surface (shown as Extended Data Fig. 8h, i). The following sentence was added to the manuscript on Page 14, lines 274-277: In proteoliposomes with a DPPC content of 85%, we were able to detect a pronounced (14-fold) acceleration of lipid transfer mediated by TTYH2 (Fig. 5b-d), which strongly exceeds the effect obtained from the mere tethering of ApoE-containing lipoproteins to the membrane (Extended Data Fig. 9h, i).

- In another set of experiments, we used liposome- and cell-based assays to investigate the ability of TTYH2 to scramble lipids. We have already investigated this property in our previous study on TTYH proteins, where we did not detect scrambling activity¹. We have now revisited these studies by using two different methods to reconstitute TTYH2 and again did not find convincing evidence for lipid scrambling. Similarly, we did not detect scrambling in cellular assays monitoring PS exposure on the cell surface by the binding of labeled Annexin V (shown as Extended Data Fig. 9j, k). The following sentences were added to the manuscript on Page 14, lines 281-284: We also revisited a potential function of TTYH2 as a scramblase that catalyzes the lipid transfer between membrane leaflets but did not find convincing evidence in either reconstituted systems or cellular assays (Extended Data Fig. 9j, k), in line with results obtained from earlier studies. Page 17, line 357-360: Although a net transfer of lipids to the close-by leaflet would lead to an imbalance that ultimately requires the relaxation by a lipid scramblase, we did not find evidence that TTYH2 itself would be the protein mediating this process (Extended Data Fig. 9j, k).

Cellular localization:

- We have engaged in immunocytochemistry experiments to better define the cellular localization of TTYH2 and found colocalization of TTYH2 with the endosomal marker Rab9 (shown as Extended Data Fig. 3b, c). The following modification was added to the manuscript on Page 7, lines 131-134: In parallel, we have studied the cellular localization of TTYH2 by confocal fluorescence microscopy and found a punctate pattern inside cells, which overlaps with ApoE and the endosomal marker Rab9, providing further evidence for its intracellular localization (Fig. 2c, d, Extended Data Fig. 3a-c).

Cellular experiments:

- We have provided experiments that show the intracellular colocalization of TTYH2 with fluorescently labeled lipids that were added to cells as part of ApoE lipoprotein particles, demonstrating that the lipid cargo of endocytosed lipoproteins would encounter TTYH2 in a cellular context. The data is displayed as Extended Data Fig. 9I. The following sentence was added to the manuscript on Page 14, lines 284-289: Finally, we studied HEK293 cells incubated with ApoE-lipoproteins containing fluorescently labeled PE, where we found a distinct intracellular distribution of the labeled lipids and a partial overlap with TTYH2, indicating endocytosed phospholipids to be retained in the same compartment (Extended Data Fig. 9I). These results provide first evidence for the colocalization of endocytosed lipids with TTYH2 as expected for its presumed role as lipid transfer catalyst.

With our revision, we hope to have clarified major questions of the reviewers and we are positive that the new data has strengthened the evidence concerning the role of the TTYH2/ApoE interactions. Naturally, a first study on this topic leaves important details open, which will be subject of future investigations.

Following, we address the comments of the three referees in detail.

Referees' comments:

Referee #1 (Remarks to the Author):

The manuscript titled “Interactions between TTYH2 and ApoE facilitate endosomal lipid transfer” reports identification of antibodies against TTYH2, discovery of physical interactions and formation of a stable complex between TTYH2 and ApoE, cellular localization of the TTYH2-ApoE complex, structure determination of TTYH2 and TTYH2 in complex with ApoE by cryoEM, and demonstration of lipid transfer function of the TTYH2-ApoE complex. The study is rigorous and comprehensive and is a brilliant tour de force that shows a mastery of an array of cellular and molecular level experimental approaches. The experiments are well-designed and expertly executed, and the interpretations are accurate and often self-restrained.

The main conclusions are truly ground-breaking and seem to support several previous speculations. The study also establishes a new avenue of research that will lead to deeper understanding of the complex nature of lipid transfer and homeostasis in a cell.

We thank the reviewer for these highly supportive comments and want to confirm that we have attempted for a conservative interpretation of the results throughout.

I have no major concerns or criticisms. I do find the choice of single molecule force spectroscopy somewhat odd, and would prefer to see a more traditional binding assay, however, this is a minor point and not a call for new experiments. I also feel that the resolutions of the TTYH2-ApoE complex in either detergent micelle or vesicle/lipidic environment may have room for improvement, however, the authors are cautious and the conclusions are adequate.

As mentioned before, we consider single molecule force spectroscopy (SMSF) the most adequate method to characterize the described interaction for several reasons: While Surface Plasmon Resonance and Microscale Thermophoresis experiments turned out to be unsuitable, the results from the competition with the bound Sb2 as method to test the site specificity of the interaction are qualitative. Additionally, the latter method did not permit us to probe the interaction at low pH, where the binding affinity of Sb2 was strongly decreased. In contrast, SMSF allowed us to characterize a binding event that carries the hallmarks of a specific interaction on a single molecule level. This method is by now well established for the investigation of molecular interactions, as demonstrated in numerous studies in related systems. In our revision, we have extended our SMSF data by measurements at low pH and the introduction of important controls as detailed later.

Referee #2 (Remarks to the Author):

Sukalskaia et al. have reported a novel interaction between TTYH2 and APOE which may contribute to the endosomal lipid transfer. Using synthetic nanobody bound TTYH2 structure, they have reconstructed several CryoEM density maps of different forms of human APOE3 including nonlipidated APOE3, APOE3 C-terminal domain (APOECTD) and lipidated APOE particles binding to full length TTYH2 on the lipid nanodisc. Additionally, they also provided supportive data including cellular staining of colocalized TTYH2 and APOE, binding between

non-lipidated APOE3 and TTYH2 measured by single molecule force spectroscopy (SMFS) and lipid transfer assays.

In summary, the potential discovery of APOE-TTYH2 interaction and its impact on endosomal lipid transfer is both interesting and novel. The structural investigation on the APOE-TTYH2 complex with a variety of APOE forms by CryoEM is also impressive. However, there are some important concerns rise from the uncertainty of current CryoEM density maps, the lack of alternative methods to cross-verify, and the unclear nature of physiological relevance of the interaction (at least from current data). Concerns to address are listed below.

We thank the reviewer for the generally supportive comments and we are positive to be able to clarify most of the stated concerns in the response provided below.

Major:

1. It is unclear how the authors managed to use Sb1 to pulldown the APOE with TTYH2 because later data showed that the binding interface of Sb1-TTYH2 complex significantly overlaps with APOE-TTYH2 binding interface. Do the authors have an explanation?

While the binding interface of both proteins overlap, our competition assays also suggest a potential formation of a ternary complex, as shown in extended Dat Fig. 3g, which appears to be sufficient to have retained endogenous ApoE on the column. We also want to emphasize that we have carried out similar pulldowns with a sybody binding to the unrelated protein LRRC8A as control, where we have not detected ApoE in the retained protein fractions, excluding a non-specific effect (see Extended Data Fig 2b).

2. The APOE-TTYH2 structure is not yet convincing though this be able to be clarified with some additional results.

We would like to emphasize that the described structural features are the result of several large and independent datasets, which all display equivalent structural properties. We are thus positive that they provide the best view of the complex that is currently achievable. The high quality of each dataset is illustrated in the high-resolution reconstructions from subsets of particles not

containing bound ApoE. Together these particle populations have led to the TTYH2 reconstruction at 2.7 Å resolution shown in Fig. 3e. The comparably low resolution of bound ApoE is likely a consequence of the intrinsic structural heterogeneity of the protein, where the N-terminal domain forms a tight four-helix bundle, whereas the C-terminal domain, which is involved in the interaction with TTYH2 and lipids, is loosely packed. The C-terminal domain is responsible for the oligomerization of ApoE in the lipid-free state, which we could confirm for the construct of the isolated C-terminal domain, that was used for the structure shown in Figure 3c. Indications for the structural heterogeneity of the C-terminal domain is also found in an NMR structure of an ApoE construct carrying five point-mutations, that were introduced to prevent its oligomerization². In this structure, the C-terminal half of the protein is loosely wrapped around the N-terminal domain (Reference 13). In light of the observed mobility of the C-terminal domain, we expect its interaction with TTYH2 to be heterogeneous, which precludes a reconstruction of a single conformation of the protein at high resolution.

In contrast to the complexes with delipidated ApoE, the low resolution of the entire density of TTYH2 in complex with lipidated ApoE is presumably a consequence of the heterogeneity of the disc-shaped lipoprotein and its flexibility in a complex consisting of two structural assemblies of similar size, which further compromises particle alignment.

a. The resolution of all complex density maps is misleading because all the near-atom resolutions resulted from a highly impacted region inside TTYH2, where the resolutions of actual binding interface were all about 7Å (Fig. S4-7). This may be correct, but the authors should clearly report the details of this result.

The mentioned resolution refers to the entire map obtained from a fourier-shell correlation analysis. As frequently observed in cryo-EM structures, the resolution of the density is not isotropic, which is clearly documented in the local resolution mapped on the density shown in Extended Data Figs. 4e, 5j, 6f. In this case, the resolution of the ApoE portion is clearly much lower and we expect that it reflects the intrinsic mobility of the protein and the heterogeneity of its binding. We did not claim that the mentioned resolution would extend to the ApoE part and we have made that clear in our revised manuscript.

Page 10, line 184-187: While the heterogeneous binding of ApoE to TTYH2 compromises its resolution and consequently prohibits a detailed interpretation of its interaction, the maps confine the region in contact with the apolipoprotein to the upper part of the extracellular domain facing the gap between the two subunits.

b. Following the point 2a, the resolution of APOE density in the complex of delipidated APOE and TTYH2 was about 7Å (Fig. S4E). At this resolution, could the authors ensure that this is the APOE density? Could the authors apply further 3DVA masking on that APOE density to improve the regional refinement or just obtain more particles with additional imaging to improve the resolution? At least to the extent where some of the APOE density could be modeled? The same issues can be found in other complex maps.

There are several lines of evidence supporting that the observed density originates from ApoE:

1. It is not found in previous data sets of TTYH2 determined in absence of ApoE
2. It is at a similar location as the displaced sybody.
3. It is at the equivalent location as the strong density of the gold label in a complex of TTYH2 with labeled ApoE.
4. It is found at the same position in independent datasets of the complex.

For our revision, we have now recorded additional datasets of gold-labeled ApoE added to TTYH3 and a TTYH2 mutant containing four mutations in the presumed interaction site with ApoE. In both cases, no density was observed in the same region (Extended Data Fig. 4g, h). Similarly, we did not find equivalent density in a high-resolution dataset of a sample of TTYH3 that was incubated with delipidated ApoE (Extended Data Fig. 5 a-f). In all cases, further 3DVA and masking of the ApoE density did not improve the map in this region and we do not find sufficient features that would justify the modeling of the ApoE structure.

c. If one compares the final 3D reconstructions between APOE-TTYH2 and APOE-CTD-TTYH2 (Fig. 3B&C, S4&S5), both APOE density maps suggest a very compacted structure of APOE C-terminal domain instead of a major α -helix showed by previous literature [1-3]. Certainly, the

APOECTD might undergo a conformational change upon binding to TTYH2. However, the authors need to either show high resolution modeling or apply mutagenesis to verify their structure results.

We intentionally refrain from a detailed interpretation of the observed density with a model as we assume that it originates from the interaction with a comparably flexible molecule of unclear oligomeric state with considerable conformational heterogeneity. While this prevents the detailed assignment of residues involved in the interaction, our study has confined the interaction to be mediated by the C-terminal domain of the molecule. From the location of the ApoE density, we were able to spot the region on TTYH2, which is presumably buried in the interaction interface and which, in its sequence composition, deviates from the two other human paralogs (Extended Data Fig. 4f). We have thus constructed a mutant where four amino acids were replaced by their corresponding residues in TTYH3. A structure determined with gold labeled ApoE did not show any strong density of the label at the binding site, similarly to TTYH3, thus further emphasizing the relevance of the site for the interaction.

d. Related to previous point 2c, the APOE-TTYH2 complex density map from the overexpression system raises more concerns of the CryoEM data (Fig. 3D). Regardless of the heterogeneity that the authors mentioned, the real elusive observation is the change of stoichiometry and binding interface between APOE and TTYH2. Now the binding stoichiometry becomes 2 vs. 2 instead of 1 APOE to two TTYH2 molecules. The binding domain of TTYH2 slides to the side compared to the inner face side (Fig. 3D vs. 3B&C). Considering the relative low resolution of APOE in all the maps, the authors need to be able to explain such a difference and also provide more binding results to reconcile this inconsistency.

While the TTYH2/ApoE complex obtained by co-expression of both proteins in HEK293 cells contains ApoE in a potentially heterogeneous lipidation state, we find equivalent properties as for a sample assembled from separately purified components. Consequently, we do not interpret this structure as showing an altered interaction interface or changed binding stoichiometry. This is illustrated after introduction of an additional refinement step for the same set of selected particles (documented in Extended Data Fig. 6), where the obtained density shows very similar features to the complex assembled from purified components (shown in the revised Figure 3d).

e. The reconstruction of lipidated APOE complex with TTYH2 in Fig. 4 was not completely satisfying. First, the low resolution of all domains make this map not completely determined. Second, the box size was too small so that it cut off the edge of lipidated APOE particles where a spherical boundary could even be observed in the final reconstruction (Fig. 4B&C).

The density obtained for TTYH2 with bound ApoE-lipoprotein discs shown in Fig. 4b is the result of the interaction of two similarly sized components via a comparatively small interface. Together with the large compositional heterogeneity of the lipoprotein particle, the described properties compromise particle alignment and consequently the obtained resolution. Nevertheless, the displayed densities have resulted from two independent large datasets that allow for the unambiguous assignment of both components and the placement of dimeric TTYH2, which generally does not alter its conformation. A density with equivalent features was obtained from a structure of the same complex obtained from native vesicles, where the protein was never solubilized in detergents (Fig. 4c). We thus consider the reconstructions to be at the limit of what can currently be achieved for such complex.

We also disagree that the selected box size would be too small as illustrated in FigR. 1. We consider the box size as appropriate since it is almost double the particle diameter (FigR. 1a). The circular boundary seen in the map of TTYH2-ApoE-lipoprotein complexes (Fig. 4b, c) corresponds to the edge of a circular mask applied during refinement whose boundaries become visible at low map contour and which does not truncate parts of the structure (FigR. 1a). In case of the complex in detergent, the increase of the mask size does not alter the general features of the map, but compromises the quality of the reconstruction (FigR. 1b). The general shape of these particles is also directly observed in the images (FigR. 1c). In the case of TTYH2 in vesicles, the mask size only limits the lower membrane edge (Fig. 4c) but does not affect the ApoE density.

FigR. 1. Parameter dependence of the TTYH2-ApoE reconstruction quality. **a**, 3D reconstruction of a TTYH2/ApoE lipoprotein complex at different contour levels. At decreasing contour levels, the boundaries of the spherical mask applied during refinement become visible, and are outside of the TTYH2-ApoE complex structure. The size of the box is shown right. **b**, Density of the same reconstruction refined with a larger mask shows similar features but lower quality. **c**, Selected particle images of TTYH2 in complex with lipidated ApoE from motion and CTF-corrected micrographs. **d**, 2D classes of TTYH2 in complex with delipidated ApoE from particles extracted in a larger box size show similar arrangement as particles extracted in a normal box size and no indication of interfering particles.

f. As partially mentioned in point 2e, there are concerns on the stringent box size chosen by the authors in all the complex reconstructions. First, have the authors tried some larger box sizes to pick particles and align them to ensure the extra density was not from another TTYH2 nanodisc or some other background average? Because TTYH2 has been shown with trans-interactions by itself [4] at the exact binding interface between APOE and TTYH2 reported in this study. Second, could the authors also provide 2D class images of the final 3D reconstructions in the supplementary figure?

FigR. 2. 2D class images from particle sets used for the final 3D reconstructions of the indicated dataset. **a, b,** Particles were extracted in a box of 600 pixels. **c-f,** Particles were extracted in a box of 440 pixels.

As explained before, the choice of the box size was optimal and an increase did not affect the general properties of reconstructions. To demonstrate this point further, we have re-extracted the particles from the TTYH2/ApoE^{delipidated} map in a larger box size of 600 pixels and obtained 2D classes, clearly showing the ApoE density on top of TTYH2 dimer and no other molecules around the complex (FigR. 1d). The mentioned trans-interactions between TTYH2 molecules were only observed between monomeric TTYH2 molecules in the absence of Ca²⁺ in a single study³, and never in our data. The described interaction interface would be buried in dimeric proteins. 2D class images of the final 3D reconstructions have been prepared and are shown in FigR. 2. These would not fit into the current Extended Data Figures but could be provided as supplementary data if required.

g. Maybe it's an orientation issue, but it's hard to understand the conformation of TTYH2 in the nano-gold labeled APOE complex with TTYH2 (Fig. 3a). Could the authors perhaps just fit TTYH2 model into that low resolution map and indicate where the labeled APOE could be? That would be informative.

We now show a model fitted in the density of TTYH2 in complex with gold-labeled ApoE in Fig. 3a.

3. The interactions between APOE-TTYH2 were only determined by single molecule force spectroscopy (SMFS) in addition to CryoEM analysis (Fig. 2E&F). It directly measures the molecular binding force instead of more popular biochemical assays such as ITC and SPR.

We have described before, why we consider SMFS a superior method to probe the binding of ApoE to TTYH2, since it tends to engage in non-specific interactions with membrane proteins. As a complementary qualitative method, we use sybody displacement to probe the site-specificity of the interaction. While ITC is off-limits for a poorly expressed human membrane protein, we have attempted SPR measurements but did not observe a signal even in case of Sb2 used as positive control. Microscale Thermophoresis, as alternative method, has shown binding of TTYH2 but also non-specific interactions at lower affinity with the iron transport protein Ferroportin used as negative control FigR. 3.

FigR. 3. Titration of delipidated ApoE to TTY2 shows a saturable binding event with an IC_{50} of 303 nM. A similar binding with somewhat lower affinity was also observed for the Fe^{2+} -transporter Ferroportin (FPN), presumably reflecting of non-specific interactions.

Moreover, the measurement in Fig. 2E&F was between APOE3 and TTYH2. Therefore, it raises three questions:

a. In the same assay, the authors should provide a negative control such as APOE3 NTD and a positive control such as Sb1.

In our revised manuscript, we have included SMFS experiments with immobilized Sb2 as positive control, where we find a generally similar binding characteristic (shown in Extended Data Fig. 3k). To further illustrate the specificity of binding, we have added soluble Sb2 to the buffer surrounding the immobilized Sb2 or TTYH2 and found in both cases competition of the interaction, illustrating its specificity and the overlap of the epitopes (Extended Data Fig. 3l). Finally, we used the same method to probe the interaction between TTYH2 and ApoE at pH 5.5 and found similar properties.

b. The authors should verify the binding affinity with ITC or SPR to provide true K_D values, which will be more informative for the readers to comprehend the binding affinity of APOE3-TTYH2 with cross-reference from other interactions.

We would like to emphasize the challenges associated with the determination of the K_D for the TTYH2-ApoE interaction. As mentioned before, ITC is out of reach for this poorly expressed human membrane protein and SPR was attempted but did not yield any signal, even for the positive control (TTYH2-Sb2). Investigations using microscale thermophoresis (MST) allowed the measurement of an apparent K_D , although these experiments also revealed a non-specific interaction for the unrelated membrane protein FPN, which presumably reflects the binding of the apolipoprotein to the hydrophobic surfaces of membrane proteins (FigR. 1). In contrast to ensemble measurements, SMFS, as single molecule technique, allows a clear discrimination between specific and unspecific interactions and, due to the short contact times, would not capture any binding events associated with the unfolding of ApoE in order to interact with hydrophobic surfaces. The measurements have provided a quantitative estimation of the off-rate, which when paired with on-rates typically found for sybody interactions, that were previously determined in our group for other membrane proteins (10^5 - 10^6 mol⁻¹ sec⁻¹) would yield a K_D in the micromolar range.

c. Have the authors assessed the binding affinity changes among APOE2/3/4? This will be very informative and improve the significance of determining the APOE-TTYH2 interactions.

We have at this stage not investigated the interaction of TTYH2 with other ApoE isoforms. While generally interesting, we consider these experiments to be beyond the scope of the current investigations. In light of the location of the two residues distinguishing different ApoE isoforms at the end of the N-terminal domain, and the location of the TTYH2 interaction site at the C-terminal domain of the molecule, we expect a generally similar interaction for all three isoforms.

4. The authors should include mutagenesis verifications to demonstrate the binding interface observed in the CryoEM density maps, at least mutating the key residues in the APOE C terminal region.

As detailed before, the high intrinsic mobility of the ApoE C-terminal domain in conjunction with the heterogeneity of its interaction with TTYH2 has precluded a detailed interpretation of the binding mode. While our results have placed the interactions to the C-terminal domain, the assignment of the interaction region in ApoE requires a systematic approach that is beyond the scope of this first study and that will be part of follow-up investigations. For our study, we have instead prepared a construct of TTYH2 containing mutations of four residues in the presumed binding site to their equivalent residues found in TTYH3 and studied its interaction with ApoE by cryo-EM using a gold labeled lipoprotein. Neither for this mutant nor for TTYH3 did we find any evidence for binding. In case of TTYH3, this result was further confirmed in a high-resolution structure of the protein with unlabeled ApoE, where no density of the bound apolipoprotein could be identified either.

5. The functional investigation of TTYH2-APOE mediated endosomal lipid transfer was not strong.

We would like to emphasize the difficulties associated with the establishment of lipid transfer assays in reconstituted systems, due to the non-specific interaction of lipoproteins with liposomal membranes. We thus made large efforts to establish a system where this non-specific interaction

was minimized, while lipid transfer was still occurring. To our knowledge, our approach is novel and there have not been any other more convincing studies on related processes published to date. The obtained data is robust and it provides initial functional evidence for the facilitation of lipid transfer by TTYH2.

a. The authors should include truncated TTYH2 as a negative control in their liposome assay in Fig. 5. It's hard to tell whether it should be E1/2 or E5/6 based on current maps but the authors might know better.

It is not clear what the reviewer is exactly referring to, but a truncated form of TTYH2 would presumably not properly fold and oligomerize, since the extracellular domain is not a separate entity but formed by two regions that are contiguous with the membrane-inserted part of the protein. It is thus highly unlikely that such truncated protein could serve as negative control. We want to emphasize that the observed interactions require a properly folded and assembled dimer.

b. The authors should show the difference of lipid transfer from exogenous APOE to HEK cells with or without TTYH2 expression.

In our revised manuscript, we have now provided data where we show the colocalization of TTYH2 with fluorescent phospholipids that were reconstituted into ApoE-containing lipoprotein particles and taken up by cells. This assay provides initial evidence for the encounter of lipids transferred by ApoE and TTYH2 in endosomal compartments (Extended Data Fig. 9I). More detailed studies including genetic knockouts of the protein are beyond the scope of the present manuscript and will be subject of future investigations.

c. The authors also should determine the changes of APOE-TTYH2 binding at neutral pH vs. acidic pH to demonstrate that the binding occurs in endosome/lysosome. With the current buffer system (Fig. 5B-D, pH 7.4), why would the lipid transfer occur post endocytosis? It should be processed on the cell membrane.

We have studied the interaction between ApoE and TTYH2 at lower pH (i.e. pH 5.5) by SMFS and found similar binding properties (Extended Data Fig. 3i). The characterization by Sb2 competition, in contrast, did not work due to the decreased affinity of sybody binding to TTYH2 at low pH, which is manifested in its dissociation on size exclusion chromatography. The exact location of lipid transfer from lipoproteins is still not resolved and it is generally accepted that the endocytosis of ApoE proceeds in a state where it still contains a substantial part of its lipid load as also suggested from our studies with fluorescent lipids, which were found in intracellular organelles after ApoE was externally added to cells (Extended Data Fig. 9l).

6. The experiments show interactions of apoE from HEK cells with TTYH2 or exogenous apoE artificially lipidated particles with TTYH2. HEK cells produce a very poorly lipidated form of apoE that differs very much for cells that are the main producers of apoE in vivo which would include hepatocytes peripherally and astrocytes in the CNS. It would be important to determine physiological relevance and potential relevance to disease to study whether similar interactions of lipidated apoE produced by the main cells that produce apoE in the body (hepatocytes and astrocytes) occur with TTYH2. Also, would similar results of interactions between TTYH2 be seen with other similar apolipoproteins, specifically the most similar to apoE, apoA1? This would help to understanding the specificity of the TTYH2-apoE vs. interactions with other apolipoproteins.

Our study demonstrates the interaction between TTYH2 and ApoE that is mediated by the C-terminal domain of the lipoprotein and is largely independent on its lipidation state. The lipoprotein particles containing phospholipids and cholesterol used in this study were generated by reconstitution and their general size and structure was confirmed by cryo-EM (Fig. 4a, Extended Data Fig. 9a). Although the discoid lipoprotein particles might only represent a subset of the ApoE containing lipoproteins present in our body, they do nevertheless resemble a very similar population observed inside the brain as also shown in a recent publication on astrocyte-secreted lipoproteins⁴. Though interesting, the potential interaction of TTYH2 with other lipoproteins is beyond the scope of the current study and will be subject of future investigations.

Minor points:

1. In fig. 2C, it would be better to include endosomal marker such as EEA1 or Rab5/7 to distinguish intracellular localizations of the TTYH2.

We have extended our immunocytochemistry studies and found overlap between TTYH2 and the endosomal marker Rab9 (Extended Data Fig. 3b, c).

2. The authors should provide at least NS-TEM images to demonstrate the disc-shapes of the APOE particles with their detergent-solubilized lipidation methods. Because this method may lead to a spherical shape [5] instead of disc-shapes, which might impact the interpretation of the final cryoEM map in fig. 4.

While NS-TEM is a generally useful method to gain initial structural information at very low resolution if the sample is limiting, our reconstructions of disc-shaped lipoproteins, which were obtained with the cryo-EM data of a sample containing TTYH2 ApoE-lipoproteins, provides a much more detailed view of the lipoprotein particles in three dimensions and at higher resolution. These revealed reconstructions with distinct sizes that generally conform with the structural features observed in ApoE lipoprotein discs secreted from astrocytes that were reported in a recent study⁴ and that clearly showed the bimodal distribution of density associated with the lipid headgroup region (Fig. 4a, Extended Data Fig. 9a). Spherical apolipoproteins would require a different lipid composition containing triglycerides and cholesterol esters that would reside in the core of the spherical particle.

3. Scale bars are all missing in the panels. Please add them accordingly.

We have introduced a scale bar defining dimensions of TTYH2 and its density in Fig. 1 a. The dimensions of equivalent densities displayed in other figures are defined by the placed TTYH2 model, which is either visible as ribbon or indicated by the colored surface of the density.

References:

[1] Wang, G et al. "Conformations of human apolipoprotein E(263-286) and E(267-289) in

aqueous solutions of sodium dodecyl sulfate by CD and 1H NMR.” Biochemistry vol. 35,32 (1996): 10358-66. doi:10.1021/bi960934t

[2] Chen, Jianglei et al. “Topology of human apolipoprotein E3 uniquely regulates its diverse biological functions.” Proceedings of the National Academy of Sciences of the United States of America vol. 108,36 (2011): 14813-8. doi:10.1073/pnas.1106420108

[3] Stuchell-Brereton, Melissa D et al. “Apolipoprotein E4 has extensive conformational heterogeneity in lipid-free and lipid-bound forms.” Proceedings of the National Academy of Sciences of the United States of America vol. 120,7 (2023): e2215371120. doi:10.1073/pnas.2215371120

[4] Li, Baobin et al. “Structures of tweety homolog proteins TTYH2 and TTYH3 reveal a Ca²⁺-dependent switch from intra- to intermembrane dimerization.” Nature communications vol. 12,1 6913. 25 Nov. 2021, doi:10.1038/s41467-021-27283-8

[5] Peters-Libeu, Clare A et al. “Model of biologically active apolipoprotein E bound to dipalmitoylphosphatidylcholine.” The Journal of biological chemistry vol. 281,2 (2006): 1073-9. doi:10.1074/jbc.M510851200

Referee #3 (Remarks to the Author):

Integral membrane proteins in the Tweety family were previously hypothesized to participate in the transfer of lipids from soluble lipid carriers to the cell. Here the authors have identified ApoE as a lipid carrier that interacts directly with the Tweety protein TTYH2. They used subcellular fractionation and imaging studies to demonstrate that both proteins co-localize and to endosomal compartments (but see point 3 below), and they demonstrate that ApoE both in its lipid bound and apo state stably interacts with TTYH2, not least of all by obtaining structures of the complexes by cryo-EM. They propose a model in which ApoE first binds to its receptor on the cell surface, is internalized into endosomes, where it dissociates from its receptor due to pH changes and is therefore able to associate with TTYH2. They propose that TTYH2 helps in the transfer of lipids from the lipoprotein particle and into cellular membranes--via a cavity/channel starting in the soluble part of TTH2 that extends from the membrane and into the membrane. They have developed an assay that shows that lipid transfer from lipoprotein

particle to liposomes is (somewhat) faster in the presence of TTH2 than not, although this may just be due to the fact that TTH2 tethers the lipoprotein particle to the liposome (they did not control for that possibility in Fig 5, but they need to; see #1 below).

I am very intrigued by this manuscript because of its timeliness. Their story critically supports a fundamental emerging concept in protein-mediated lipid transfer, namely that integral membrane proteins are critical in transferring lipids out of and into membranes, between soluble carriers and the membrane, by lowering the energy barrier of this transfer and thus affecting rates. There are some examples of this for the transport of cholesterol out of lysosomes by the Niemann-Pick protein complex and also in sterol transfer in Wnt signaling. In the NPC there is a soluble portion and an integral membrane portion, featuring a contiguous channel serving for sterol insertion into and through the membrane. There is also a recent paper on BioRxiv for a complex featuring a BLTP family lipid transport protein, which also has an integral membrane portion (Kang Y, Lehmann KS, Vanegas J, Long H, Jefferson A, et al. 2024. bioRxiv), presumably to facilitate transfer of glycerophospholipids into the membrane and probably scrambling (although the authors do not address the latter possibility). There is also a low resolution structure of the ATG2 (lipid transport tube)-ATG9 (integral membrane protein) complex, which supports the direct transfer of lipids between soluble carrier and integral membrane protein (Wang, Y., et al. (2024). "Structural basis for lipid transfer by the ATG2A-ATG9A complex." NSMB) Note that the lipid transport field, at least for transport at contact sites, has until very recently been oblivious to intramembrane dynamics. They thought soluble proteins do everything!

We thank the reviewer for these supportive comments.

Here are issues that I feel the authors must address:

1. They need to establish in their in vitro lipid transfer assay that the reason TTYH2 facilitates transfer is in fact because it helps with lipid insertion, not just that it is tethering the ApoE lipoprotein particle near the liposome membrane so that lipid transfer is actually even feasible. (For an example of the power of tethering, please see PMID 36282247 vs 35764626). They could probably dissect out the contribution of tethering versus membrane insertion by assessing the

effect of tethering ApoE via a hexahistidine tag to liposomes versus TTHY-containing liposomes. Is there any difference at all?

We have investigated the effect of ApoE tethering to the surface of liposomes via a His₁₀ tag binding to lipids carrying a Ni-NTA group and found some increase in the kinetics of the fluorescence decay, although this effect is much smaller than the one observed in TTYH2 containing proteoliposomes (Extended Data Fig. S9i). While the moderate enhancement in the non-specific transfer can be expected as a consequence of the increase of the local concentration of ApoE-lipoproteins on the surface of liposomes, it does not account for the pronounced increase in lipid transfer observed in case of TTYH2.

2. As said, I love their model, but I would suggest that it might be energetically difficult to keep inserting lipids into one leaflet of a membrane bilayer, as suggested in their Fig. 6 (but not 5), thus creating leaflet asymmetry, and I would propose that a scramblase activity is also involved in this process to distribute incoming lipids between membrane leaflets. I suspect very strongly, as based on their very, very exciting higher resolution structure from this manuscript, showing lipids bound to TTH2 (Fig. 3f), that TTH2 can also transfer lipids between membrane bilayers—ie, it has scramblase activity—so that the leaflets stay the same size. This structure showing lipids appearing to be scrambled is super-exciting in my mind and would be an excellent rationale for publishing this work in this versus a more plebian journal. The author's lab has worked on scramblases before, and so they know how to do in vitro scrambling assays to test this possibility. They should do these.

We agree that the sided insertion of lipids into one membrane leaflet would require a parallel activity of a lipid scramblase that dissipates the resulting membrane tension. However, such process does not necessarily have to be conferred by the same protein. In our previous study, we have investigated a potential role of TTYH2 as a lipid scramblases by its reconstitution into detergent de-stabilized liposomes and did not find evidence for lipid scrambling¹.

Fig. R4. Investigation of lipid scrambling by TTYH2. Assay of lipid scrambling in proteoliposomes containing TTYH2 reconstituted from, **a**, preformed and destabilized liposomes and, **b**, solubilized lipids. **a, b**, Left, scrambling is followed by the bleaching of fluorescent lipids located in the outer membrane leaflet upon the addition of the membrane-impermeable reducing agent dithionite (*). Shown are representative traces from a reconstitution containing TTYH2 (cyan) in comparison to mock liposomes of the same lipid batch (orange). Right, size exclusion chromatography profile of solubilized proteoliposomes containing TTYH2 in comparison to a sample of the protein used for reconstitution. The peak of dimeric TTYH2 is indicated by an asterisk. **c**, Summary of results from *in vitro* scrambling experiments. Shown is the difference in fluorescence intensity between TTYH2-containing liposomes and mock liposomes 200 s after addition of dithionite in comparison to equivalent experiments with the Ca^{2+} -activated lipid scramblase TMEM16F, which is inactive in absence of Ca^{2+} . **d**, Fluorescence intensity of fluorescent Annexin V bound to the surface of TMEM16F knockout cells transfected with the indicated constructs at resting Ca^{2+} concentrations. F518H corresponds to a TMEM16F mutant with pronounced constitutive activity.

In response to the comments of the reviewer, we have revisited these studies and also investigated whether scrambling would be enhanced by low pH or ApoE binding. To this end, we have performed *in vitro* studies using two different methods for the reconstitution of TTYH2 with either detergent-destabilized liposomes, or, alternatively, by reconstitution with detergent-solubilized

lipids with equivalent composition, and compared the fluorescence decay to mock-reconstituted liposomes from the same batch. The detergent was in all cases removed by the addition of biobeads and the incorporation of TTYH2 was assayed by SDS-page and size exclusion chromatography of the protein re-extracted from liposomes (FigR. 4a, b). In both cases, we found a similar low scrambling activity as observed previously¹ in liposome preparations that show a peak of the re-extracted protein at the expected elution volume (FigR. 4a-c). This activity was neither enhanced by the addition of ApoE nor the lowering of the pH (FigR. 4c). We occasionally observed a detectable decrease of the fluorescence below the 50% level in proteoliposomes reconstituted from detergent-solubilized lipids where the re-extracted protein was aggregated and we associate this behavior with a compromised reconstitution. To investigate lipid scrambling by a complementary approach, we have studied the exposure of PS in a cellular assay by the binding of fluorescently-labelled Annexin V to the cell surface. These experiments were carried out with a HEK293 TMEM16F-KO cell-line that was transfected with different constructs and assayed at resting Ca²⁺ concentrations. The fluorescence intensity of mock-transfected cells was used to define the baseline in absence of scrambling activity, the TMEM16F mutant F518H, which shows strong basal scrambling already at resting Ca²⁺ concentrations, as positive control. The comparison with TTYH2 transfected cells, where part of the protein was targeted to the plasma membrane as confirmed by surface biotinylation in previous studies¹, yielded a fluorescence level that was close to mock transfected cells or inactive TMEM16F at resting Ca²⁺ concentrations. Consequently, also the cellular assay did not indicate scrambling activity of TTYH2. In summary, we did not obtain convincing evidence that TTYH2 would facilitate lipid scrambling by two different assays, in agreement with our previous study¹.

In the revised manuscript, the data is shown as Extended Data Fig. 9j, k, and the following sentence was added:

Page 14, lines 281-284: We also revisited a potential function of TTYH2 as a scramblase that catalyzes the lipid transfer between membrane leaflets but did not find convincing evidence in either reconstituted systems or cellular assays (Extended Data Fig. 9j, k), in line with results obtained from earlier studies.

3. The microscopy should be improved for Fig 2c, which show TTYH2 localizing on punctae.

The authors should demonstrate that TTYH2 co-localizes with an endosome marker to definitively show that the punctae are, in fact, endosomes.

For our revision, we have extended our immunocytochemistry analysis and found colocalization of TTYH2 with the endosomal marker Rab9. We have independently also confirmed the colocalization of Rab9 with endocytosed fluorescently-labeled ApoE. The pictures are shown as Extended Data Fig 3b, c.

4. The authors admit in the last paragraph of the manuscript's discussion section that "functional/cellular studies" are required to confirm the importance of TTYH2 in lipid transfer. They should do these. Ideally, they could dissect the role of TTYH2 in tethering ApoE lipoprotein particles to membranes and actual membrane insertion and maybe scrambling. The reviewer has experienced that KOing scrambling activity can be very difficult (due to limitations in the in vitro assays), so this dissection may/or may not be possible, but an attempt should be made as success would strengthen the manuscript. A lack of functional data should not be acceptable for a high-tier journal like this one.

As described above, we have included experiments showing the effect of tethering of ApoE to the surface of liposomes, and *in vitro* and cellular experiments assaying lipid scrambling to our revised manuscript.

To probe the encounter of TTYH2 with endocytosed lipids in a cellular context, we have also performed experiments where we monitored the distribution of labeled lipids that were added to cells as part of ApoE-lipoprotein particles, and found the lipids to overlap with TTYH2 in intracellular compartments. The data shown as Extended Data Fig. 9l provides initial evidence for the colocalization of the endocytosed lipids with TTYH2. A more detailed study concerning the role of TTYH2 in cellular lipid distribution using genetic knockouts of the protein will be part of future investigations.

We have added the following sentences:

Page 14, lines 284-289:

Finally, we studied HEK293 cells incubated with ApoE-lipoproteins containing fluorescently labeled PE, where we found a distinct intracellular distribution of the labeled lipids and a partial overlap with TTYH2, indicating endocytosed phospholipids to be retained in the same compartment (Extended Data Fig. 9l). These results provide first evidence for the colocalization of endocytosed lipids with TTYH2 as expected for its presumed role as lipid transfer catalyst.

Page 17, Line 344-347:

Cellular studies demonstrating signal overlap in enriched TTYH2 environments with endocytosed PE lipids suggest a potential interaction in physiological conditions, as indicated by cluster-like structures that point to the co-localization of TTHY2 and PE lipids (Extended Data Fig. 9l).

5. If they can demonstrate a role for TTYH2 in intramembrane lipid dynamics, this role should be emphasized more in the discussion as the general importance of integral membrane proteins for lipid transfer is not yet firmly established.

Our additional data, have strengthened the proposed mechanism concerning the role of TTYH2 in endosomal lipid transfer and have provided a strong foundation for future studies. We have mentioned the general resemblance of the TTYH2-ApoE system to distantly related processes mediating lipid exchange between the bilayer and soluble proteins.

Page 18, line 369-373:

Although unique in its details, TTYH2-mediated lipid transfer resembles other systems where membrane proteins facilitate the exchange of lipids between bilayers and soluble lipid carriers such as the lysosomal NPC sterol transporters, or the ATG2-ATG9 system which channels lipids during autophagosome formation.

Since we did not find evidence for lipid scrambling by TTYH2 itself we have added the following sentence:

Page 17, line 557-360: Although a net transfer of lipids to the close-by leaflet would lead to an imbalance that ultimately requires the relaxation by a lipid scramblase, we did not find evidence that TTYH2 itself would be the protein mediating this process (Extended Data Fig. 9j, k).

The manuscript is well written and very clear.

Once the conclusions are strengthened as described above, this manuscript represents a significant advance in our understanding of the mechanisms underlying lipid trafficking

- 1 Sukalskaia, A., Straub, M. S., Deneka, D., Sawicka, M. & Dutzler, R. Cryo-EM structures of the TTYH family reveal a novel architecture for lipid interactions. *Nat Commun* **12**, 4893, doi:10.1038/s41467-021-25106-4 (2021).
- 2 Chen, J., Li, Q. & Wang, J. Topology of human apolipoprotein E3 uniquely regulates its diverse biological functions. *Proc Natl Acad Sci U S A* **108**, 14813-14818, doi:10.1073/pnas.1106420108 (2011).
- 3 Li, B., Hoel, C. M. & Brohawn, S. G. Structures of tweety homolog proteins TTYH2 and TTYH3 reveal a Ca(2+)-dependent switch from intra- to intermembrane dimerization. *Nat Commun* **12**, 6913, doi:10.1038/s41467-021-27283-8 (2021).
- 4 Strickland, M. R. *et al.* Apolipoprotein E secreted by astrocytes forms antiparallel dimers in discoidal lipoproteins. *Neuron* **112**, 1100-1109 e1105, doi:10.1016/j.neuron.2023.12.018 (2024).

We have made the following changes to our manuscript to meet the editorial requests and respond to final comments of reviewer 2.

Referee #2:

In this revision and rebuttal, Sukalskaia et al. have made considerable efforts to address this reviewer's original comments and have provided additional data to clarify their results. Although several critical points remain only partially resolved due to technical limitations, the reviewer acknowledges the authors' efforts and recognizes that three key aspects have been addressed using alternative approaches:

- 1. Validation of the cryoEM density map, supported by the absence of APOE density in the TTYH2 mutant or TTYH3 mixed with APOE;*
- 2. Assessment of relative binding affinity using Microscale Thermophoresis;*
- 3. Liposome-based assays demonstrating lipid transport activity.*

While the reviewer has some disappointment with the overall data quality – especially given the group's expertise in structural biology and the manuscript's submission to this journal – the reviewer is OK with this revision, if Figs R1, R3, and R4 are included in the main paper. These figures are crucial for readers to understand the data processing and the relative binding strengths involved in this interaction, which will be important for future studies.

Since we were unable to fit Figs. R1, R3 and R4 in the Extended Data figures at suitable positions due to lack of space we decided to provide them as Supplementary Figs. 7-9. Please note that the relevant scrambling data that was part of Fig. R4 is presented as Extended Data Fig. 9i-l. Example Traces of *in vitro* experiments and a validation of the reconstitution are now shown in Supplementary Fig. 8. The Microscale Thermophoresis experiments (Fig. R3) are now shown as Supplementary Fig. 7. The data did not contribute to the conclusions made in our manuscript and does not contain sufficient repeats to justify its inclusion in the Extended Data. Finally Supplementary Fig. 9 (Fig. R1) contains a documentation that demonstrates that the chosen processing strategy did not introduce artefacts.

The reviewer also recommends that the authors include a discussion of the study's limitations and the technical challenges encountered.

In response to the request of the reviewer we have included the following sentences:

Page 7 Line 125-129:

As classical methods such as surface plasmon resonance spectroscopy and microscale thermophoresis turned out to be not suitable to characterize ApoE binding to TTYH2, we have

subsequently used single molecule force spectroscopy (SMFS) to directly assay the interaction between both proteins by measuring the mechanical dissociation of single ApoE-TTYH2 complexes.

Page 14 line 281-286:

These data are limited by several technical challenges. Binding studies were complicated by the property of ApoE to interact with hydrophobic surfaces, structural studies by the conformational heterogeneity of the apolipoprotein and its lipid complexes, which has precluded a structural characterization of ApoE at high resolution, and lipid transfer studies by the non-specific interactions of lipoproteins with cellular membranes. However, despite these limitations, our work has provided robust insight that is supported by complementary observations.

Page 43 line 835-837:

Since classical binding experiments such as Microscale Thermophoresis turned out to be unsuitable (Supplementary Fig. 7), we probed the site specificity of ApoE binding to TTYH2 using Sb2, which occupies a similar epitope.

Response to editorial requests:

STATISTICS: When revising your manuscript, you should ensure that any statistical analysis used is sound and that it conforms to Nature's guidelines. A collection of articles explaining the basics of statistical analysis and advice on how to best present it can be found here.

Statistics conforms to Nature's guidelines

REPRODUCIBILITY: All of the checklists provided with the current submission (Reporting summary, Editorial policy checklist, and Code and software checklist (if applicable)) should be updated to reflect the revisions made and submitted with the revised manuscript.

Updated checklists are provided.

LENGTH: In print, biological sciences papers do not normally exceed 8 pages on average; the final print length, however, is at the editor's discretion. The typical length of an 8-page article with 5 modest (quarter-page) display items is 4300 words. If a composite figure (with multiple panels) must occupy at least half a page in order for all the elements to be visible, the text length may need to be reduced accordingly to accommodate such figures. Essential but technical details can be moved into the Methods or Supplementary Information (see below).

The total length of the revised manuscript was estimated as 7.68 pages according to an excel sheet provided by your journal for a previous publication.

TITLE: Titles cannot exceed 75 characters (including spaces); they must not contain punctuation.

The title contains 71 characters and does not contain punctuation.

SUMMARY PARAGRAPH: All Nature papers begin with a fully referenced paragraph, typically no longer than 200 words, aimed at readers in other disciplines. This paragraph starts with a 2- to 3-sentence, basic introduction to the field; continues with a 1-sentence statement of the main findings starting 'Here we show' or an equivalent phrase; and finally, concludes with 2 to 3 sentences putting the main findings into general context so it is clear how the results described in the paper have moved the field forward.

The summary paragraph is 194 words long and meets the indicated format.

MAIN TEXT: If further introductory material is necessary, the main text can begin with up to 500 words of introduction expanding on the background to the work (some overlap with the summary is acceptable), before proceeding to a concise, focused account of the findings, and ending with 1 or 2 short paragraphs of discussion. Sections are separated with subheadings (up to 40 characters including spaces) to aid navigation.

The introduction is 491 words long. Subheadings are below 40 characters.

REFERENCES: As a guideline, most papers should include no more than 50 main text references; additional references can be cited in (and listed after) the Methods section, as detailed below.

The main text contains 43 references, the methods additional 21 references.

FIGURE LEGENDS: These should be listed sequentially after the main text references and not in the figure files; they should not exceed 300 words each.

Each legend should begin with a brief title for the whole figure and continue with a short description of each panel and the symbols used. Each figure legend should contain, for each panel where relevant, the following information:

- * the exact sample size (n) for each experimental group/condition, given as a number, not a range;*
- * a description of the sample collection allowing the reader to understand whether the samples represent technical or biological replicates (including how many animals, litters, cultures, etc);*
- * a statement of how many times the experiment shown was replicated;*
- * definitions of statistical methods and measures:*
- * very common tests (e.g., t-test, simple Chi-square tests, Wilcoxon and Mann-Whitney tests) can be identified by name only, but more complex techniques should be described in the Methods;*
- * whether tests are one-sided or two-sided;*
- * whether there are adjustments for multiple comparisons;*
- * the statistical test results (e.g., P values);*
- * the definition of 'center values' as median or average;*
- * the definition of error bars as s.d. or s.e.m.*

Any descriptions too long for the figure legend should be included in the Methods section; see here for further explanation.

Legends are below 300 words and contain the requested information.

METHODS: After the main text figure legends there should be a section entitled "Methods", which provides the full, step-by-step instructions that would allow other researchers to replicate the results. The Methods section will not appear in print but will appear online in the full-text HTML and PDF versions. The Methods section should be written as concisely as possible but should contain all elements necessary to allow interpretation and reproduction of the results. If there are additional references in the Methods section, their numbering should continue from the last reference in the main text, and they should be listed following the Methods section. Specialized methods that require chemical structures, figures or tables, or methods requiring equations, cannot be accommodated in the Methods section of the main text file. If such information is part of the Methods, the entire Methods section must instead be included within a Supplementary Information text file.

OK

MAIN TEXT STATEMENTS: Several statements (which will not appear in print but will appear online in the full-text HTML and PDF) are required after the Methods, and before the Extended Data legends. First, there should be an Acknowledgements section, listing grant/financial support. Next, we require a detailed Author Contribution statement; the specific contributions of each author, particularly in terms of which authors performed which specific experiments, must be listed. This is followed by a Competing Interest statement. Financial or non-financial interests should be noted here, as well as any patents; patent information should include at a minimum what is covered by the patent and who submitted the patent application. Finally, an Additional Information statement should include information regarding reprints and permissions and name the author(s) to whom correspondence and requests for materials should be addressed. For details of "end note" style and an example see here.

OK

DATA AND CODE AVAILABILITY STATEMENTS: All original research manuscripts published in Nature Portfolio journals must include a Data availability statement (DAS). This statement must make the conditions of access to the "minimum dataset" that is necessary to interpret, verify and extend the research in the article, transparent to readers. This minimum dataset may be provided through deposition in public community/discipline-specific repositories, custom proprietary repositories for certain types of datasets, or general repositories like Figshare, Zenodo and Dryad. Providing large datasets in supplementary information is strongly discouraged and the preferred approach is to make data available in repositories. More information on Nature Portfolio's reporting standards and preparing your Data availability statement can be found here.

OK

For all studies using custom code or mathematical algorithms that are deemed central to the conclusions, a Code availability statement (CAS) must be included, indicating whether and how the code or algorithm can be accessed, including any restrictions to access. the CAS should be provided as a separate section after the DAS but before the references. Code should be deposited in a DOI-minting repository such as Zenodo, Gigantum or Code Ocean and cited in the reference list. We

encourage you to manage subsequent code versions and to use a license approved by the open source initiative. Additional details can be found here.

N/A

Wherever possible the data used in the paper should be placed into a public data repository or presented as Supplementary Information. If data can only be shared on request, please explain why in the DAS and in the cover letter. The DAS must list which data are included (e.g. by figure panels and data types) and mention any restrictions on availability. If a dataset generated or analysed during the study is publicly available and has a unique DOI, please include it in the reference list and cite the dataset in the Methods. Accession numbers for any newly determined sequences, structures, microarray or zoobank data; project IDs for MG-RAST data; accession numbers for X-ray crystallographic coordinates and structure factor files; or comparable NMR or cryoEM data, should be included only in the DAS.

OK

*DISPLAY ITEMS: We ask that you take stock of all the data that have been generated throughout the review process and ensure that only the data most central to the conclusions are presented in the main text figures. Although figures can be presented within the text file during the review process, they must be removed from the final text file and uploaded as separate individual files. Figures should be comprehensible to readers in other or related disciplines and assist their understanding of the paper. Figures should be as small and simple as is compatible with clarity. Main text figures (but **not** Extended Data) must be provided in an editable format. Acceptable formats include .ai, .cmx, .cdr, .doc, .eps, .pdf, .ppt, .ps, .psd, .svg and .xls; unacceptable formats include .cvs, .gif, .jpg, .png and .tif. All panels of a figure should be logically connected; each panel of a multipart figure should be sized so that the whole figure can be reduced by the same amount and reproduced on the printed page at the smallest size at which essential details are visible. For guidance, Nature's standard figure sizes are 9, 12, or 18 cm wide; the maximum permitted height of a figure is 17 cm. All panels of figures must be on a single page and assembled into a rectangular shape for publication; any essential alignments (parts horizontal, vertical, spacings of stereo pairs, etc) should be indicated.*

Nature requests that authors of accepted manuscripts contribute towards the total cost of reproduction of colour figures in print. You will be charged a fee for the print reproduction of colour figures (see here for cost information). Tables should be prepared using the Table menu in Microsoft Word.

FIGURE FORMATTING: Lettering in all figures (e.g., labelling of axes) should be in uniform, sans-serif font, in lower-case type, and large enough to permit substantial reduction for publication (the minimum font size permitted is 5 pt). Separate parts of a figure are labelled a, b, etc. Units have a single space between the number and the unit, and follow SI nomenclature or the nomenclature common to a particular field. Thousands are separated by commas (1,000). Unusual units or abbreviations are defined in the legend.

IMAGE PRESENTATION: Data figures are integrated into the main paper online. An overview of the key features of this presentation may be found here. We strongly encourage you to consult the guidelines. A discussion of our standards regarding how images should be prepared and presented can be found

here.

All figures meet the requested criteria and were submitted as editable .psd files at high resolution.

EXTENDED DATA: Extended Data do not appear in print but are included online within the full-text HTML and integrated in the downloadable PDF. Extended Data are an integral part of the paper, and only data that directly contribute to the main message should be included. All Extended Data must be referred to in the main text, and their legends should be listed sequentially at the end of the main text, not in the Extended Data files. The Extended Data should be assembled into a maximum of 10 A4 size, multi-panelled display items, submitted as individual files in .jpg, .tif or .eps format only. They should be of the same quality as print figures, but there are important differences in their formatting. More specific instructions can be found here. We encourage authors who are describing complex processes to include a schematic of the main finding as part of the Extended Data to aid readers unfamiliar with the immediate discipline.

Nine Extended Data Figures were submitted as flattened .tif files at a resolution of 300 dpi meeting the indicated requirements. One Extended Data Table was submitted as word file.

SUPPLEMENTARY INFORMATION: Supplementary Information (SI) is online-only, peer-reviewed material that is essential background to the study (e.g., large data sets, more complex methods, and calculations), but which is too large or impractical, or of interest only to a few specialists, to justify inclusion in the print version of the paper (see here for further details). While SI should not typically contain data figures (any figures additional to those appearing in the main text should be formatted as Extended Data), we require that the raw, uncropped data for gels be presented as an SI figure (see below). Tables may be included in SI, but only if they are unsuitable for formatting as Extended Data (e.g., tables containing large data sets or raw data tables that are best suited to Excel files). If a manuscript has SI, each discrete item of the SI (e.g., videos, tables) must be referred to at an appropriate point in the main manuscript. We ask that you provide a word file entitled "SI Guide", containing a cover page with manuscript title and author information; a table of contents (preferably with page numbers); and then any SI text, notes, figures, and titles and legends for any separate SI files. Additional information can be found here.

Please pay careful attention to the formatting of the SI because it is not subedited. After the paper has been accepted, SI files can only be amended for critical changes to the scientific content, not for style.

The supplementary Information was submitted as single PDF document containing six figures (Supplementary Fig. 1-6) of uncropped gels and three Figures that were part of our response to reviewer document and that were requested to be included by reviewer 2 (Supplementary Fig. 7-9).

SOURCE DATA (GRAPHS): To increase transparency, we strongly encourage you to provide, in spreadsheet form, the data underlying the graphical representations used in the figures. In the case

of all experiments presenting data from animal models, this is a requirement and is not optional. This is in addition to our well-established data-deposition policy for specific types of experiments and large datasets. Online readers of the manuscript will be able to access the graphical source data directly from the figure legend. Spreadsheets can only be submitted in .xls, .xlsx or .csv formats. One file per figure is permitted. If there is a multi-panelled figure, the source data for each panel should be clearly labeled in the file; alternatively the source data for a figure can be included in multiple, clearly labeled sheets within an Excel file. File sizes of up to 30 MB are permitted, but it is expected that the vast majority of graphical source data files will be considerably smaller than this. When submitting these files with your manuscript, you should select the "Source Data" file type and use the title field in the file description tab to indicate the figure(s) to which the source data pertain.

We have submitted the source data as single layered .xls file.

RAW DATA (GELS): You must provide the original source images for all data obtained by electrophoretic separation (e.g., EMSA, northern/Southern/western blots, etc). The raw images must be assembled into a single .pdf or .tif file (multiple gels on a single page is encouraged). The file should be uploaded as Supplementary Figure 1. The full scanned images must be in uncropped form and contain size/molecular weight markers and loading controls. There should be an accurate indication of how the gels were cropped for the final figure. Please clarify in the figure legends and raw data files whether controls (such as beta-actin) were run on the same gel as loading controls, or on separate gels as sample processing controls (see here). While the data can be displayed in a relatively informal style, there must be a correspondence between each source data image and a specific main text or Extended Data figure. The main text or Extended Data figure legends should refer to the uncropped scans explicitly (e.g., "For gel source data, see Supplementary Figure 1."). For examples, see here or here.

As mentioned above the uncropped gels were included in the Supplementary Data document as Supplementary Figures 1-6.

THIRD PARTY RIGHTS: Please identify any content used in your article, whether in the main text, extended data or supplementary information, that comes from a third party. This could include figures, tables, images, videos or text boxes that are reproductions or adaptations of items that have previously been published elsewhere and/or are owned by a third party. It also encompasses pictures taken by professional photographers, maps and images downloaded from the internet. You must obtain the right to use each of these items and provide evidence that you have these rights for your paper. You will also need to give proper attribution to the copyright holders in your paper. We ask that you fill out a Third party rights table and upload this with your manuscript. You can find more information about third party rights here.

We have provided licenses of several figures and figure panels that were prepared with the BioRender software (as mentioned in the Acknowledgments).

ORCID: As part of our efforts in improving transparency in authorship, we now request that all authors

identified as ‘corresponding author’ create and link their Open Researcher and Contributor Identifier (ORCID) with their account on Nature’s manuscript tracking system prior to acceptance. ORCID helps the scientific community achieve unambiguous attribution of all scholarly contributions. More information and instructions on how to associate your ORCID and MTS accounts can be found here. If you have any issues attaching an ORCID identifier to your MTS account, please contact the Platform Support Helpdesk.

OK

* Include a point-by-point response to our referees and to any editorial suggestions;

OK

* Complete the Reporting summary;

OK

* Complete the Editorial policy checklist;

OK

* Complete the Code and software checklist (if applicable);

N/A

* Ensure it complies with our format requirements.

OK